# Cold-induced *Arabidopsis* FRIGIDA nuclear condensates for *FLC* repression

Pan Zhu[1], Clare Lister[1] & Caroline Dean[1✉]

Plants use seasonal temperature cues to time the transition to reproduction. In *Arabidopsis thaliana*, winter cold epigenetically silences the floral repressor locus *FLOWERING LOCUS C* (*FLC*) through POLYCOMB REPRESSIVE COMPLEX 2 (PRC2)[1]. This vernalization process aligns flowering with spring. A prerequisite for silencing is transcriptional downregulation of *FLC*, but how this occurs in the fluctuating temperature regimes of autumn is unknown[2–4]. Transcriptional repression correlates with decreased local levels of histone H3 trimethylation at K36 (H3K36me3) and H3 trimethylation at K4 (H3K4me3)[5,6], which are deposited during FRIGIDA (FRI)-dependent activation of *FLC*[7–10]. Here we show that cold rapidly promotes the formation of FRI nuclear condensates that do not colocalize with an active *FLC* locus. This correlates with reduced FRI occupancy at the *FLC* promoter and *FLC* repression. Warm temperature spikes reverse this process, buffering *FLC* shutdown to prevent premature flowering. The accumulation of condensates in the cold is affected by specific co-transcriptional regulators and cold induction of a specific isoform of the antisense RNA *COOLAIR*[5,11]. Our work describes the dynamic partitioning of a transcriptional activator conferring plasticity in response to natural temperature fluctuations, thus enabling plants to effectively monitor seasonal progression.

Plants overwinter before flowering in *Arabidopsis* through FRI-dependent upregulation of *FLC*. This mechanism requires FRIGIDA-LIKE 1 (FRL1), FRIGIDA-ESSENTIAL 1 (FES1), SUPPRESSOR OF FRIGIDA 4 (SUF4) and *FLC* EXPRESSOR (FLX), which associate in a FRI complex[8]. In laboratory conditions of constant cold, *FLC* is transcriptionally repressed in around two to three weeks[5,6,11]. However, in natural autumnal fluctuating temperatures in field conditions, *FLC* transcriptional shutdown took many months[3,4]. We therefore investigated whether and how FRI function might change in response to fluctuating temperature.

## FRI condensates accumulate in the cold

First, we analysed how cold influences FRI protein interactions. A line carrying a translational *FRI-GFP* fusion, expressed at the same level as endogenous *FRI*, and fully complementing the *fri* early flowering phenotype (Extended Data Fig. 1), was used for immunoprecipitation in combination with mass spectrometry (IP–MS)[10] (Supplementary Table 1). FRI–GFP did not enrich any of the FRI complex components from warm-grown plant extracts, but, counterintuitively, FRI–GFP accumulated in plants that underwent two weeks of cold exposure, and in these conditions FRL1 and FRL2—but not FLX, SUF4 or FES1—were enriched[8] (Extended Data Table 1). FRI interacted with subunits of the Mediator complex[12,13], WDR5a and ATX2 (which promote H3K4me3)[7,9,14], the PAF1 complex[15], general transcription factors[13], RNA-polymerase-II-associated proteins, and many RNA splicing factors and uridine-rich small nuclear ribonucleoproteins (snRNPs)[16–18], which suggests that FRI has a role in co-transcriptional regulation (Extended Data Table 1). The higher prevalence of these interactors

in extracts from cold-grown plants may reflect the cold-induced accumulation of FRI (Extended Data Table 1).

Second, we analysed the localization of FRI in vivo. Like many other co-transcriptional regulators[19,20] we found that FRI–GFP forms nuclear condensates, which were increased in size and number after cold exposure (Fig. 1a, Extended Data Fig. 2a–d). A FRI–Myc fusion showed similar condensate formation (Extended Data Fig. 2e, f). Fluorescence recovery after photobleaching (FRAP) revealed relatively slow FRI–GFP dynamics (Fig. 1b, c), like other biomolecular condensates[20], but different to FCA–GFP, which have previously been shown to form liquid-like foci (condensates) with a FRAP recovery time of seconds[21]. Formation of the FCA–GFP condensates was not enhanced by cold (Extended Data Fig. 2g–i); thus, FRI and FCA condensates have different biophysical properties.

We then analysed whether the FRI interactors co-associate with FRI in the condensates. FRL1–mScarlet I colocalized with FRI–GFP in nuclear condensates after transient co-expression in tobacco leaves (Fig. 1d) but did not form nuclear condensates when transfected alone (Extended Data Fig. 3a–c). Consistently, loss of FRL1 reduced the cold-induced enhancement of the size and number of FRI condensates (Extended Data Fig. 3d–f). By contrast, FRI condensates were less affected in *flx-2* and *suf4* mutants (Extended Data Fig. 3d–f), consistent with FRL1, but not FLX or SUF4, immunoprecipitating with FRI (Extended Data Table 1). Other FRI interactors, GFP–ELF7 and TAF15b–GFP, colocalized with FRI–mScarlet I in condensates (Fig. 1d, Extended Data Fig. 3g) and condensates containing the Cajal body marker protein U2B′′ (ref. [18]) frequently colocalized with FRI–GFP condensates after cold (Extended Data Fig. 3h–j). FRI condensates may therefore share components with

[1]John Innes Centre, Norwich Research Park, Norwich, UK. ✉e-mail: caroline.dean@jic.ac.uk

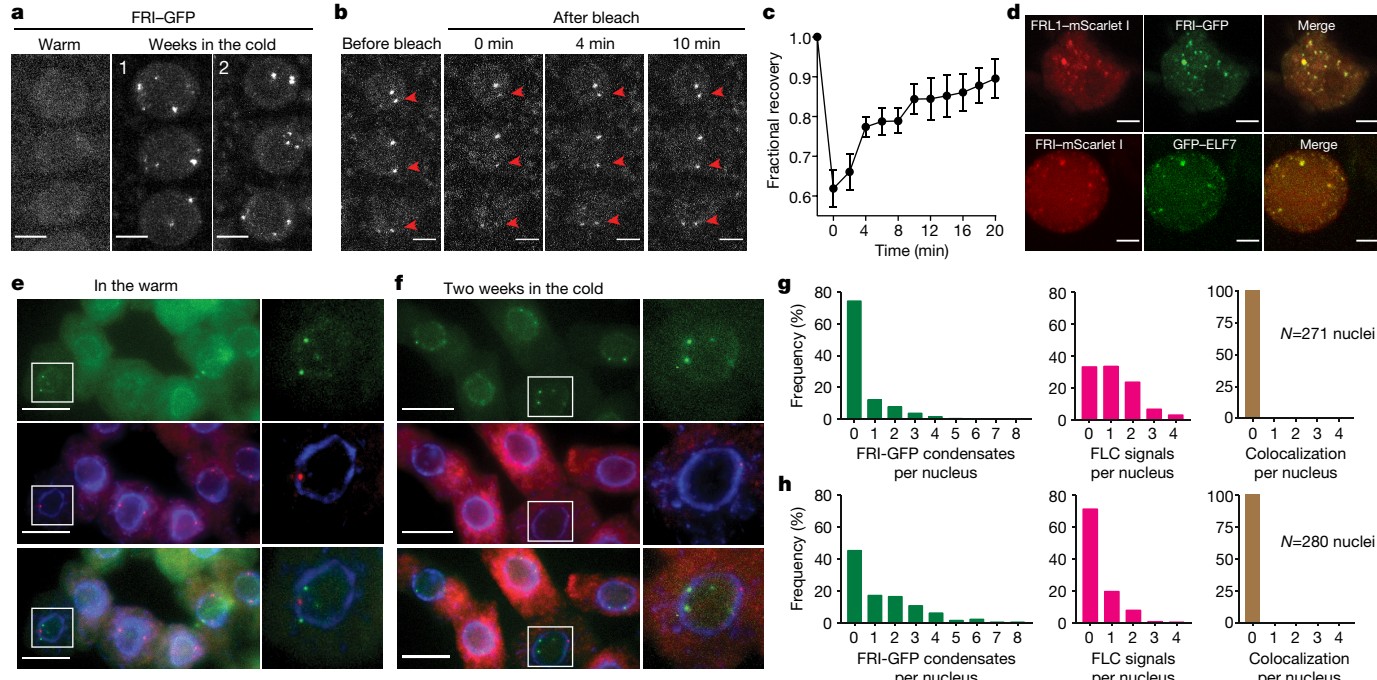

**Fig. 1 | Cold-promoted FRI nuclear condensates are linked to *FLC* transcriptional shutdown. a**, Confocal microscopic images of FRI–GFP nuclear condensates in root cells in the indicated conditions. For quantitative analysis, see Extended Data Fig. 2c, d. **b**, **c**, Images (**b**) and quantification (**c**) of FRAP of FRI–GFP nuclear condensates. Time 0 indicates the time of the photobleaching pulse. Red arrows indicate the bleached condensates. Mean ± s.e.m.; *n* = 10 condensates in 10 cells. **d**, Confocal analysis of subnuclear colocalization of FRI with the co-expressed proteins in tobacco leaf nuclei.

Data represent three independent experiments. **e**, **f**, Representative images (**e**, in the warm; **f**, in the cold) of nuclei expressing FRI–GFP (green) sequentially hybridized with intronic smFISH probes for *FLC* (red). DNA was labelled with DAPI (blue). Three independent experiments gave the same conclusion. **g**, **h**, Frequency distribution of FRI–GFP condensates (left), non-spliced *FLC* transcript signals (middle) and their colocalization per nucleus (right) in root cells in the warm (**g**) and in the cold (**h**). Numbers of analysed nuclei are as indicated. Scale bars, 5 μm (**a**, **b**, **d**); 10 μm (**e**, **f**).

Cajal bodies, similar to PML bodies in mammalian cells[22]. Overall, these data support the notion that multiple protein interactions enhance the formation of FRI condensates.

We further investigated the domains of FRI that are required for nuclear condensate formation. Consistent with disordered domains having important roles in biomolecular condensation[19,21], a GFP fusion protein with a version of FRI in which the C-terminal disordered domain was deleted no longer formed nuclear condensates (Extended Data Fig. 4). This supports the C terminus being required for FRI function[23]. Deletion of the FRI coiled-coil domains[10] also led to loss of condensate formation (Extended Data Fig. 4b, c). A naturally occurring 16-base-pair deletion that is found in the loss-of-function FRI allele in Col-0[10,23], which would produce a protein without the C-terminal disordered and coiled-coil domains, prevented condensate formation (Extended Data Fig. 4b, c). Thus, both the C-terminal disordered domain and two coiled-coil domains are required for FRI condensation in vivo.

## FRI is more stable in the cold

Increased protein concentration is known to promote biomolecular condensation[19,20] so we further examined the cold-induced accumulation of FRI. FRI–GFP accumulated after two to four weeks of cold (Extended Data Fig. 5a–c), with no corresponding change in the levels of *FRI* mRNA (Extended Data Fig. 1a). Analysis of nuclear FRI–TAP showed that this stability was not a consequence of the GFP fusion (Extended Data Fig. 5d). The FRI half-life was measured as less than 24 h in the warm, but more than 24 h in the cold (Extended Data Fig. 5e–h); thus, the cold enhancement of FRI nuclear condensates appears to be a consequence of increased FRI protein stability.

Overexpression using *35S:FRI-GFP* (Extended Data Fig. 6a, b) led to a small accumulation of FRI–GFP protein after two weeks of cold exposure and a decrease after four weeks (Extended Data Fig. 6c–e), although FRI is less stable in warm than in cold conditions (Extended Data Fig. 6f). A previous report showed that FRI abundance decreased in the cold[24], which may be a consequence of the medium as FRI protein is highly induced by glucose in warm conditions (Extended Data Fig. 6f). Overexpression of FRI–GFP is likely to influence nuclear condensate dynamics[25], but one to four weeks of cold still enhanced FRI–GFP condensation even though more FRI condensates were found in warm-grown *35S:FRI-GFP* lines compared to *FRI-GFP* (Extended Data Fig. 6g–i).

Levels of FRI–GFP protein decreased in *frl1-1* and *suf4* mutants, with no concomitant change in *FRI-GFP* mRNA[8] (Extended Data Fig. 6j–l). By contrast, FRI–GFP protein levels in *flx-2* did not change (Extended Data Fig. 6k, l), despite the disruption to FRI condensation (Extended Data Fig. 3d–f), supporting the notion that increased protein concentration is not sufficient for cold enhancement of FRI condensate formation. Together, these data show that cold stabilization of FRI and increased interaction with multiple factors reciprocally enhance the formation of FRI condensates.

## Condensates sequester FRI away from *FLC*

We further addressed the functional consequences of FRI nuclear condensate formation. FRI expanded the zone of *FLC* expression in warm-grown plants and this was antagonized by cold exposure[26] (Extended Data Fig. 7a, b). Association of FRI–GFP with the *FLC* 5′ region in warm conditions[8,9] decreased after two weeks of cold exposure (Extended Data Fig. 7c–f), paralleling the decrease in *FLC* transcription[5,11,26] (Extended Data Fig. 7g) and raising the possibility that

condensates are linked to *FLC* transcriptional repression. We used single-molecule RNA fluorescence in situ hybridization (smRNA FISH) and *FLC* intron 1 probes[27] to identify nascent *FLC* transcripts that mark transcriptionally active *FLC* loci and investigated whether FRI nuclear condensates associated with an active *FLC* locus. The nascent *FLC* transcripts and the FRI–GFP condensates never colocalized with each other (Fig. 1e–h). Notably, FRI association with the *COOLAIR* promoter increased after two weeks of cold exposure (Extended Data Fig. 7e), consistent with *COOLAIR* upregulation[5,11,26]. However, despite more cells transcribing *COOLAIR* in the cold[26] and more FRI condensates, we did not detect any colocalization between non-spliced *COOLAIR* clouds and FRI–GFP condensates (Extended Data Fig. 7h). FRI and its associated factors may therefore be sequestered into nuclear condensates away from the *FLC* locus, with nascent transcription dissolving any condensates at the locus[28].

Supporting this hypothesis, in the *35S:FRI-GFP* line, in which more FRI nuclear condensates form (Extended Data Fig. 6g–i), *FLC* transcript levels are lower (Extended Data Fig. 7i, j) and a reduction of FRI nuclear condensates in the *frl1-1* mutant (Extended Data Fig. 3d–f) correlates with a slower *FLC* transcriptional shutdown (Extended Data Fig. 7k–m). The low levels of *FLC* transcript in *flx-2* and *suf4* suggest a different mechanism (Extended Data Fig. 7k–m). FLX and SUF4 appear to promote the ability of FRI to transcriptionally activate, but they have less of a role in cold-induced FRI condensation (Extended Data Fig. 3d–f).

## FRI condensates track temperature shifts

*FLC* transcriptional shutdown occurs during autumn as temperatures fluctuate widely over daily and weekly timescales[3,4], so we tested whether the cold-induced sequestration of FRI has an important role in fluctuating temperatures. FRI–GFP nuclear condensate dynamics were analysed during a 12-h transient cold exposure (mimicking a cool night in autumn)[3]. We found that FRI–GFP condensates gradually increased in size and number (Fig. 2a, Extended Data Fig. 8a, b), through a process requiring protein synthesis (Extended Data Fig. 5e–h). The accumulation of FRI condensates correlated with downregulation of *FLC* to its lowest expression 12 h after transfer to cold (Fig. 2b), which was further repressed after 2 weeks of cold exposure (Extended Data Fig. 8a–c). Thus, FRI condensation increases rapidly in response to decreasing temperature, correlating with *FLC* transcriptional shutdown.

It has previously been observed that a spike of high temperature in autumn slows *FLC* shutdown[3,4]; thus, we asked whether the cold-induced accumulation of FRI condensates is reversible by warmth. FRI–GFP condensates, monitored at 6-h intervals over the first 24 h after plants were returned to warm temperature, were significantly reduced within the first 6 h and did not recover in either root or leaf cells (Fig. 2c, Extended Data Fig. 8d–f). This process was not fully blocked by inhibiting proteasome-mediated protein degradation (Extended Data Fig. 8g–j). *FLC* transcription was upregulated in parallel (Fig. 2d) and showed further reactivation after transfer to warm conditions for 10 days (Extended Data Fig. 8k). Therefore, the cold-induced condensation of FRI is easily reversed by warmth, which suggests that the condensates serve as reservoirs allowing a rapid response to warm temperature spikes. These rapid dynamics would buffer the shutdown of *FLC* transcription in the fluctuating temperatures of autumn, contributing to a requirement for the absence of warmth for *FLC* silencing[3].

To gain further insight into the dynamics of FRI nuclear condensates in response to temperature shifts, we performed a time-lapse experiment using a temperature-controlled microscope stage. This showed that FRI–GFP nuclear condensates undergo dynamic changes in response to changing temperature, disappearing within five hours of a return to warm conditions (Extended Data Fig. 8l, Supplementary Video 1), but being rescued by cold after a three-hour warm spike (Extended Data Fig. 8m, Supplementary Video 2). In response to the

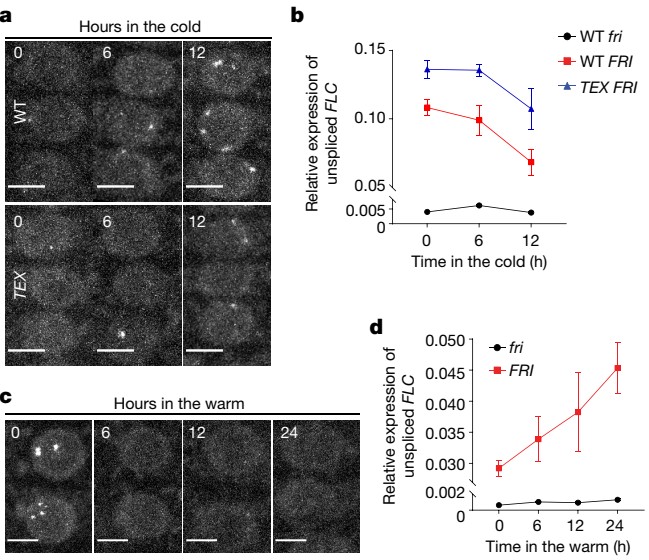

**Fig. 2 | Short-term temperature fluctuations influence the formation of FRI nuclear condensates and *FLC* transcription. a**, **c**, Confocal microscopic images of FRI–GFP nuclear condensates in wild-type (WT) (top) and *TEX* (bottom) root cells after 0, 6 and 12 h of cold treatment (**a**) and in wild-type plants after they were returned to the warm for 0, 6, 12 and 24 h after a 2-week cold treatment (**c**). Scale bars, 5 μm. For quantitative analysis, see Extended Data Fig. 8a, b, d, e. **b**, **d**, Relative transcript level of unspliced *FLC* in the indicated plants within the same time course of changed temperatures in **a**, **c** by quantitative PCR with reverse transcription (RT–qPCR). Mean ± s.e.m.; *n* = 4 (**b**) and 3 (**d**) biologically independent experiments.

temperature shifts, the FRI condensates fluctuated in number as they grew and fused (Extended Data Fig. 8n, o, Supplementary Video 2)—behaviour typical of biomolecular condensates[21,28,29]. This induction of FRI condensates did not occur in transient assays in tobacco leaves exposed to cold temperatures (Extended Data Fig. 8p–r).

## *COOLAIR* promotes FRI condensates in the cold

To investigate which other cold-specific factors might contribute to the heterotypic interactions required for condensate formation we tested a role for *COOLAIR*. One *COOLAIR* isoform is differentially induced by the FRI complex after two weeks of cold (Extended Data Fig. 9a–f): class II.ii, a distally polyadenylated transcript that includes an additional exon[5,11] (Fig. 3a). This change in splicing may involve FRI interacting with splicing factors (Extended Data Fig. 3h–j, Extended Data Table 1) that have previously been shown to have a role in cold-responsive gene regulation[16,17]. RNA immunoprecipitation (RNA-IP) showed that FRI–GFP specifically enriched *COOLAIR* class II.ii after two weeks of cold (Fig. 3b, Extended Data Fig. 9g, h). In *frl1-1*, in which the FRI condensation is severely attenuated (Extended Data Fig. 3d–f) but *COOLAIR* expression is still relatively high (Extended Data Fig. 9a–c), this enrichment was reduced (Extended Data Fig. 9i). Therefore, the FRI–class II.ii interaction is tightly connected with the cold induction of FRI condensation—a conclusion supported by the rapid changes in class II.ii in response to temperature shifts (Fig. 3c, d) and further induction after two weeks of cold (Extended Data Fig. 9f).

We next crossed FRI–GFP to a previously described *FLC* terminator exchange (*TEX*) line[5,26,27]. The *FLC* terminator–*COOLAIR* promoter is exchanged for an *RBCS3B* terminator, which prevents the cold induction of *COOLAIR* transcription[5,26] (Fig. 3c). There was no increase in the size of FRI condensates within the first 12 h of plants experiencing cold (Fig. 2a, Extended Data Fig. 8a, b), and both the size and the number of FRI condensates were significantly reduced after two weeks of cold

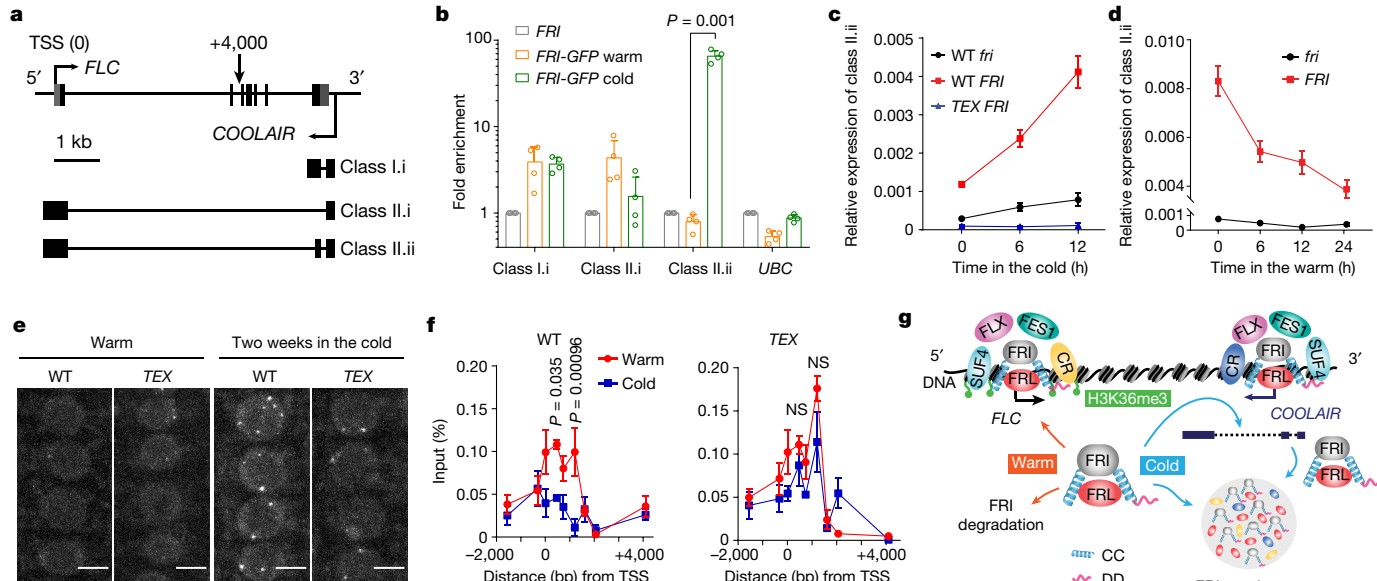

**Fig. 3 | *COOLAIR* promotes cold induction of FRI–GFP nuclear condensates and sequestration of FRI from the *FLC* promoter. a**, Schematic of *FLC* and *COOLAIR* transcripts at the *FLC* locus. Untranslated regions are indicated by grey boxes and exons by black boxes. kb, kilobase; TSS, transcription start site. **b**, RNA-IP assay of spliced *COOLAIR* enrichment by FRI–GFP with *UBC* as control. Mean ± s.d.; *n* = 4 replicates over 2 biologically independent experiments. Two-tailed *t*-test. **c**, **d**, Relative transcript level of *COOLAIR* class II.ii in the indicated plants within the same time course of changed temperatures as in Fig. 2 by RT–qPCR. Mean ± s.e.m.; *n* = 4 (**c**) and 3

(**d**) biologically independent experiments. **e**, Confocal images of wild-type and *TEX* root tip nuclei expressing FRI–GFP. Scale bars, 5 µm. For quantitative analysis, see Extended Data Fig. 9j, k. **f**, FRI–GFP occupancy on *FLC* promoter region in WT and *TEX* plants by CHIP. Mean ± s.e.m.; *n* = 3 biologically independent experiments. The exact distance from TSS referred to **a**. Two-way ANOVA adjusted by Sidak's multiple comparisons test. NS, no significance. **g**, A working model for temperature-controlled FRI nuclear condensation in *FLC* transcriptional regulation. CC, coiled-coil domain; CR, co-transcriptional regulators; DD, disordered domain.

(Fig. 3e, Extended Data Fig. 9j, k). This is linked to decreased levels of FRI protein in *TEX*, which might reflect negative feedback from reduction of *COOLAIR* expression (Extended Data Fig. 9l, m). Similarly, the lower cold induction of FRI condensation in *flx-2* and *suf4* mutants (Extended Data Fig. 3d–f) may relate to reduced *COOLAIR* expression (Extended Data Fig. 9a–f). In addition, chromatin immunoprecipitation (ChIP) experiments showed that the cold-induced reduction of FRI occupancy at the *FLC* promoter was less in the *TEX* line compared to the wild type (Fig. 3f), which could account for the inefficient cold-induced *FLC* transcriptional shutdown in *TEX*[5,26] (Fig. 2b). This mechanism could explain the *COOLAIR*-facilitated removal of H3K36me3 at the *FLC* locus in the early vernalization phase[5] (Fig. 3g).

## Conclusions

Together, our experiments point to a temperature-controlled condensation mechanism that modulates FRI activation of *FLC* transcription; such a mechanism would facilitate *FLC* shutdown in natural fluctuating temperatures. FRI associates with transcriptional co-activators and recruits histone modifiers to the *FLC* promoter, establishing an active transcriptional state at *FLC* in the warm (Fig. 3g). The instability of FRI in the warm enables a fast turnover of these transcriptional complexes (Fig. 3g). After transfer to the cold, FRI protein is stabilized and changes in protein or *COOLAIR* interactions result in the accumulation of nuclear condensates that sequester FRI away from the *FLC* promoter (Fig. 3g). This contrasts with mechanisms in which condensates promote the dwell time of transcriptional regulators at specific genomic loci[19]. The combinatorial effect of temperature-specific splicing, and splicing-specific nuclear condensation, gives a wide range of possibilities for *FLC* regulation in different temperature regimes. Given the recognized importance of biological condensates in gene regulation, this type of mechanism may have widespread roles in the plasticity of plant responses to varying environments[29–31].

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

## Methods

### Plant material and growth conditions

All the mutants and transgenic lines were in Columbia-0 (Col-0), except for *FCA-GFP* and *FLC-GUS*, which are in *Landsberg erecta* (Ler). *TEX FRI* (ref. [5]), *FCA-GFP* (ref. [21]), *frl1-1* (ref. [32]), *flx-2* (ref. [33]), *suf4* (ref. [34]) and *FLC-GUS* with or without *FRI^JU223^* (ref. [35]) have previously been described. *FRI^JU223^* is a functional transgenic *FRI* allele as described previously[10]. *FRI* is Col-0 with an introgressive active *Sf2 FRI* allele and was previously described[36]. *frl1-1*, *flx-2* and *suf4* were crossed into the *FRI* background to generate *frl1-1 FRI*, *flx-2 FRI* and *suf4 FRI*.

Seeds were surface-sterilized and sown on Murashige and Skoog (MS) agar plates without glucose or sucrose (unless otherwise stated) and kept at 4 °C in the dark for 3 days. For non-vernalization (NV), seedlings were grown in warm conditions (16 h light, 8 h darkness with constant 20 °C) for 10 days unless otherwise stated. For cold treatment, seedlings were pre-grown in warm conditions (16 h light, 8 h darkness with constant 20 °C) for 10 days unless otherwise stated, and then transferred to cold (8 h light, 16 h darkness with constant 5 °C) for a long-term vernalization, such as 2 weeks and 4 weeks (2WV and 4WV). As there is no difference between *FRI* and *fri* in response to photoperiod[34] and photoperiod influences flowering in parallel with *FLC*[37,38], we use this short-day photoperiod for long-term cold treatment to match natural conditions. For short-term cold treatment (0–12 h), the light conditions were kept the same as in warm conditions. For cold-to-warm transfer experiments, after 1 or 2 weeks of cold treatment, seedlings were moved to warm conditions (16 h light, 8 h darkness with constant 20 °C) for another specified duration, such as 6 h. Plants grow faster in the warm than in the cold (one day growth in the warm is approximately equivalent to seven days in the cold)[39], therefore for warm–cold comparisons plants were harvested at the same developmental stage rather than after the same time period.

### Plasmid construction and generation of transgenic lines

To generate *pFRI:FRI-GFP* and *pFRI:FRI-Myc* constructs, a 4.56-kb PmlI/HpaI fragment containing genomic *FRI* sequence from H51 (ref. [10]) was cloned into pBluescript KS (+). This clone spans 0.84 kb upstream of the *FRI* start codon to 1.4 kb downstream of the *FRI* stop codon. *GFP* sequence was amplified and inserted into the XbaI site over the *FRI* stop codon. The *FRI-GFP* fragment was then subcloned into SLJ75515[40] and transformed into *Agrobacterium tumefaciens* C58 (pGV2260) by triparental mating[36]. *Myc* sequence was inserted into the XbaI site 6 bp downstream of the *FRI* start codon. The *FRI-Myc* fragment was then subcloned into the EcoRI site of pGreenII 0229 and transformed into *A. tumefaciens* pGV3101 (pMP90). All of the XbaI sites were introduced by site-directed mutagenesis by the oligos listed in Supplementary Table 2. Transgenic *FRI-GFP* and *FRI-Myc* plants in the Col-0 background were generated by floral dipping with *A. tumefaciens*. Transgenic lines with a single insertion were identified in the T$_2$ generation and segregated 3:1 for Basta resistance. Independent *FRI-GFP* lines showed the same cold-induced condensation.

For the *pFRI:FRI-TAP* and *p35S:FRI-GFP* constructs, *FRI* cDNA was fused with the *TAP* and *GFP* sequences, respectively, and *FRI-TAP* was subcloned into pBluescript KS (+) with the *FRI* promoter, whereas *FRI-GFP* was driven by a CaMV *35S* promoter. *pFRI:FRI-TAP* and *p35S:FRI-GFP* in the binary vectors were then transferred into *A. tumefaciens*. Transgenic plants were also in the Col-0 background.

The *35S:GFP-ELF7* (ref. [41]), *pTAF15b:TAF15b-GFP* (ref. [42]) and *pCsV* (Cassava vein mosaic virus):*TAF15b-GFP* (ref. [42]) constructs were previously described.

To generate *pFRI:FRI-mScarlet I*, the same *FRI* genomic sequence from H51 (ref. [10]) was used as a template and mScarlet I sequence was inserted to the XbaI site over *FRI* stop codon in pBluescript KS (+). The *FRI-mScarlet I* fragment was then subcloned into SLJ6999[40]. For *pFRI:FRI-Col-0-GFP*, an equivalent *FRI* genomic sequence from Col-0

was amplified[10] and the GFP sequence was fused before the premature stop codon (after 1,335 bp downstream from ATG, equivalent to 314 amino acids). For *pFRI:FRI-DD-GFP*, the C-terminal disordered domain (DD) from 1,833 bp downstream from ATG to the stop codon (equivalent to 451 to 609 amino acids) was deleted. For *pFRI:FRI-CC-GFP*, two fragments encoding the coiled-coil domains (CC) are deleted: the first spans 118 bp to 300 bp downstream from ATG (equivalent to 40 to 100 amino acids) and the second is from 1,680 bp to 1,832 bp downstream from ATG (equivalent to 400 to 451 amino acids). For *pFRL1:FRL1-mScarlet I*, a 3,972-bp fragment containing *FRL1* genomic sequence from Col-0 was amplified. It includes 1,502 bp upstream from *FRL1* ATG and 1,057 bp downstream from *FRL1* TAG. mScarlet I sequence was inserted before the TAG translational stop codon. *FRI-Col-0-GFP*, *FRI-DD-GFP*, *FRI-CC-GFP* and *FRL1-mScarlet I* DNA fragments were then cloned into the SLJ75515[40] destination vector by Gateway Cloning. Plasmids were transformed into *A. tumefaciens* C58 (pGV2260) for infiltration of *Nicotiana benthamiana*[21]. All primers used for construction are listed in Supplementary Table 2.

### IP–MS analyses

The IP–MS analyses were performed as described previously[21,36]. For the first experiment (IP1) 2.5 g of seedlings was cross-linked in 1% formaldehyde as previously described[43]. For the second (IP2) and third experiment (IP3), the amount of plant material used was doubled. Cross-linked plants were ground into fine powder and lysed in 50 ml of cell lysis buffer (20 mM Tris HCl, pH 7.5, 250 mM sucrose, 25% glycerol, 20 mM KCl, 2.5 mM MgCl$_2$, 0.1% NP-40, 5 mM DTT). The lysate was filtered through two layers of Miracloth (Merck, D00172956) and pelleted by centrifugation. The pellets were washed twice with 10 ml of nuclear wash buffer (20 mM Tris HCl, pH 7.5, 2.5 mM MgCl$_2$, 25% glycerol, 0.3% Triton X-100, 5 mM DTT) and resuspended with RIPA buffer (1× PBS, 1% NP-40, 0.5% sodium deoxycholate, 0.1% SDS). All the buffers were supplemented with 0.1 U μl$^{-1}$ RNaseOUT (Invitrogen, 10777019), 1 mM PMSF and Roche Complete tablets to keep the integrity of any RNA–protein and protein–protein complex. Nuclear lysate was sonicated three times (Diagenode Bioruptor; M level, 30 s ON–30 s OFF, 5 min each time) and nuclear supernatant was obtained after centrifuge. GFP–Trap magnetic agarose beads (25 μl) (Chromotek, GTMA-20) were pre-blocked with 1 mg ml$^{-1}$ BSA and 1 mg ml$^{-1}$ yeast total RNA (Sigma) in RIPA buffer for 1 h at 4 °C and incubated with the nuclear supernatant for 2 h at 4 °C. Beads were sequentially washed twice with high-salt wash buffer and low-salt wash buffer (20 mM Tris HCl, pH 7.5, 500 mM (high-salt) or 150 mM NaCl (low-salt), 0.5 mM EDTA, 0.1% SDS and 1% Triton X-100). The beads were finally resuspended in 1× SDS loading buffer and boiled at 95 °C for 15 min to reverse cross-linking. The protein samples were purified from 10% SDS–PAGE gels followed by trypsin digestion. For LCMS analysis, an Orbitrap Fusion Tribrid mass spectrometer (Thermo Fisher Scientific) equipped with an Ulti-Mate 3000 RSLCnano LC system (Thermo Fisher Scientific) was used. Data were searched using Mascot server 2.7 (Matrix Science). Results were imported and evaluated in Scaffold 4.10.0 software (Proteome Software).

### RNA expression analysis

RNA was extracted with the hot phenol method[5,36]. Genomic DNA contamination was removed by TURBO DNA-free kit (Invitrogen, AM1907) following the manufacturer's guidelines. cDNA was synthesized by the SuperScript IV reverse transcriptase (Invitrogen, 18090050) using gene-specific reverse primers. All primers are listed in Supplementary Table 2. The TAIR (The Arabidopsis Information Resource) accession numbers for the genes analysed in this study are *FRI* (AT4G00650), *FLC* (AT5G10140) and *COOLAIR* (AT5G01675). Standard reference genes *PP2A* (AT1G13320) and *UBC* (AT5G25760) for gene expression were used for normalization[4]. Data were analysed using Microsoft Excel (v.2102, 64-bit) and GraphPad Prism 7.

## Microscopy and image quantification

The seedlings used for imaging were grown in the following conditions: NV, plants were grown in warm conditions for 7 days; 1WV, plants were grown in warm conditions for 6 days before being transferred to cold conditions for 1 week; 2WV or 4WV, plants were grown in cold conditions for 2 or 4 weeks with a 5-day pre-growth in warm conditions; before a short cold treatment (12 h or shorter), plants were grown in warm conditions for 7 days, so that all of the seedlings imaged are developmentally equivalent to a 7-day-old warm-grown seedling. The warm and cold conditions are the same as described in 'Plant material and growth conditions' unless otherwise stated.

The subcellular localization of FRI–GFP in root tips and first true leaves (Extended Data Figs. 2a, b, 8f) was imaged on a Zeiss LSM780 confocal microscope using a 40×/1.2 water objective and GAsP spectral detector array. The seedlings were kept intact during imaging. GFP was excited at 488 nm (Argon ion laser, laser power: 5.0 %) and detected at wavelengths of 491–695 nm in lambda mode. Linear unmixing projection in ZEN Black (2012) software was applied and the autofluorescence detected at 526–695 nm was unmixed and labelled as background (blue). Images were exported by ZEN Blue (2012).

For quantitative analysis of FRI–GFP and FCA–GFP nuclear condensates, imaging was performed on a Zeiss LSM880 confocal microscope using a 40×/1.1 water objective and GAsP spectral detector array. GFP was excited at 488 nm (Argon ion laser, laser power: 5.0 %) and detected at wavelengths of 499–525 nm in lambda mode. An optimal $z$-step size of 0.61 μm was used over a total depth of 10.37 μm from the upper surface (18 $z$-slices) and maximum intensity projection was applied for the $z$-stack. A binary image of just the spots was then created through threshold in Fiji (ImageJ) and the spot area was subsequently measured by Analyze Particles in Fiji (ImageJ). As all images were obtained with a 0.11 μm pixel size, the area of a single pixel is 0.0121 μm². To exclude single pixels, 0.02 μm² was set as the minimum size during the analysis. The spot number (with area larger than 0.02 μm²) per nucleus was gained through displaying the spot outline generated by Analyze Particles in the original image. Violin plots reflecting the data distribution were produced with PlotsOfData[44]. Statistical evaluations with multiple comparisons tests were performed using the GraphPad Prism 7 software.

To compare the fluorescence intensity between roots, imaging was performed with a Zeiss LSM780 confocal microscope using an EC Plan-Neofluar 20×/0.50 objective. To allow comparison between treatments, the same settings were used for all images. Measurement of the fluorescence intensity was conducted with Fiji (ImageJ) with the intensity normalized to the background for each image. To collect the total intensity of each root tip, sum slices projections from a $z$-stack of 16 steps with a step size of 2 μm were applied before analysis.

## FRAP

FRAP of FRI–GFP nuclear condensates was performed as described[45] with a Zeiss LSM780 confocal microscope coupled with a Linkam Heating and Cooling Stage (Meyer instruments). A chamber was created on slides using Grace Bio-Labs Secure Seal adhesive sheets (Sigma, GBL620001) and filled with MS medium. The 2WV *FRI-GFP* seedlings were carefully transferred into the chamber with stage temperature kept at 4 °C. Using a 40×/1.2 water objective, the region of an FRI–GFP nuclear condensate was bleached using a laser intensity of 100% at 488 nm. Recovery was recorded for every 2 min for a total of 20 min after bleaching. At each time point, maximum intensity projections from a $z$-stack of 12 steps with a step size of 1 μm were applied. Analysis of the recovery curves was carried out with Fiji (ImageJ), Microsoft Excel (v.2102, 64-bit) and GraphPad Prism 7.

## Time-lapse imaging

The time-lapse microscopy of FRI–GFP nuclear condensates was performed with a Zeiss LSM780 confocal microscope coupled with a Linkam Heating and Cooling Stage (Meyer instruments). For Supplementary Video 1, 1WV seedlings were transferred into a same chamber as in FRAP experiment and observed under a 40×/1.2 water objective. GFP was excited at 488 nm (Argon ion laser, laser power: 5.0 %) and detected at wavelengths of 490–551 nm. The temperature was first set at 4 °C and images were acquired every 15 or 20 min for 5 h after the temperature rose to 22 °C. At each time point, maximum intensity projections from a $z$-stack of 12 steps with a step size of 1 μm were applied. For Supplementary Video 2, 1WV seedlings were imaged after being grown in warm conditions (20 °C) for 3 h with the stage temperature kept at 4 °C, and images were acquired every 20 min for 6 h. The same imaging settings were used as in Supplementary Video 1. Images were processed and FRI–GFP nuclear condensates were quantified with Fiji (ImageJ).

## Cycloheximide and MG132 treatment

Cycloheximide (CHX) and MG132 treatment were according to a previous study[39]. Before treatment, plants were grown on MS agar plates as described above. For treatment, seedlings were carefully transferred to new MS agar plates supplemented with either 100 μM CHX (C1988, Sigma-Aldrich) or 100 μM MG132 (474787, Sigma-Aldrich) growing for 24 h. In parallel, the same seedlings were transferred to new MS agar plates supplemented with the same amount of solvent (ethanol for CHX and dimethyl sulfoxide for MG132) as a control. Root tips were then imaged with a Zeiss LSM780 confocal microscope using a 40×/1.2 water objective. GFP was excited at 488 nm (Argon ion laser, laser power: 20.0 %) and detected at wavelengths of 489–530 nm. An optimal $z$-step size of 0.435 μm was used over a total depth of 11.75 μm from the upper surface (28 $z$-slices). Images were processed and measurements were conducted with Fiji (ImageJ). Sum slices projection was applied for the $z$-stack. The whole nuclear intensity was measured and normalized to background in each image before comparison. All images were obtained with a 0.1-μm pixel size and 0.02 μm² was set as the minimum size during the analysis of FRI–GFP nuclear condensates. Statistical evaluations with multiple comparisons tests were performed using the GraphPad Prism software.

## smFISH and immunofluorescence

smFISH and immunofluorescence was performed as previously described[26,46]. Plants were grown in warm conditions for seven days (NV) or in cold conditions for two weeks with a five-day pre-growth in warm conditions (2WV). For FRI–GFP fluorescence microscopy with sequential smFISH, root tips were fixed with 4% paraformaldehyde (PFA) before being squashed. After permeabilization in 70% ethanol for 1 h, the subnuclear localization of FRI–GFP was imaged and the stage positions were saved at the microscope. Next, cover slips were carefully unmounted and *FLC* intron 1 or *COOLAIR* intron probes labelled with Quasar 570 were hybridized at 37 °C overnight. The probes were the same as previously described[26,27]. The probe signals were detected using the same stage positions as FRI–GFP. The microscope used was Zeiss Elyra PS with a 100×/1.46 oil-immersion objective and a cooled electron multiplying CCD (charge-coupled device) Andor iXon 897 camera. GFP was exited at 488 nm and detected at wavelengths of 495–550 nm; probes labelled with Quasar 570 were excited at 561 nm and detected at 570–620 nm; DAPI was excited at 405 nm and detected at 420–480 nm. An optimal $z$-step size of 0.2 μm was used over a total depth of 2.4 μm (23 $z$-slices) and maximum intensity projection was applied for the $z$-stack using Fiji (ImageJ).

For FRI–Myc immunofluorescence and colocalization of FRI–GFP and U2B″, squashed root cells were immersed in 70% ethanol overnight at 4 °C for permeabilization. Cell walls were digested with 0.2% Driselase (Sigma, D9515) and 0.15% Macerozyme R-10 (Duchefa Biochemie, M8002) in 1× PBS at 37 °C for 40 min. After being washed three times with 1× PBST, cells were blocked with 2.5% BSA (Thermo Fisher Scientific, AM2616). Primary antibodies: 1:125 diluted anti-c-Myc (Sigma, M5546),

1:500 diluted anti-GFP (Abcam, ab290) and 1:20 diluted anti-U2B″ (4G3, a gift from P. Shaw) were incubated at 37 °C for 3–4 h. Alexa Fluor 488 anti-rabbit secondary antibody (Thermo Fisher Scientific, A-11008, dilution: 1:200) and Alexa Fluor 555 anti-mouse secondary antibody (Thermo Fisher Scientific, A-21424, dilution: 1:200) were incubated for 1 h. After DAPI staining, slides were mounted in Vectashield (Vector Lab, H-1000). Images were acquired on a Zeiss LSM780 confocal microscope using a 63×/1.40 oil objective. For FRI–Myc immunofluorescence, DAPI was excited at 405 nm and detected at 410–513 nm; FRI–Myc-Alexa Fluor 555 was excited at 514 nm and detected at 545–697 nm. For FRI–GFP and U2B″ colocalization, DAPI was excited at 405 nm and detected at 407–489 nm; FRI–GFP–Alexa Fluor 488 was excited at 488 nm and detected at 491–561 nm; U2B″–Alexa Fluor 555 was excited at 561 nm and detected at 562–634 nm.

### GUS staining
FLC–GUS staining was performed as previously described[35]. Whole seedlings were vacuum-infiltrated with GUS staining buffer (1 mM X-gluc (5-bromo-4-chloro-3-indolyl glucuronide), 100 mM phosphate buffer, pH 7.0, 10 mM EDTA, 0.1% Triton X-100, 0.5 mM ferricyanide and ferrocyanide, pH was finally adjusted to 7.0) followed by an overnight or shorter incubation at 37 °C. Whole seedlings and roots were immediately imaged after de-staining with 95% ethanol.

### ChIP and quantitative PCR
FRI–GFP ChIP was performed as previously outlined with some modifications[1,36]. Five grams of NV seedlings and 2.5 g of 2WV seedlings were cross-linked in 1% formaldehyde and nuclei were purified and lysed as in the IP–MS experiment. Anti-GFP (Abcam, ab290, dilution: 1:400) and protein A agarose beads containing salmon sperm DNA (Millipore, 16-157) were used in the immunoprecipitation. The enriched DNA was purified using the ChIP DNA Clean & Concentrator Kit (Zymo Research, D5205)[43]. All ChIP experiments were quantified by quantitative PCR (qPCR) with appropriate primers (Supplementary Table 2). The enrichment levels of FRI–GFP at *ACTIN* (*ACT*) and *SHOOT MERISTEMLESS* (*STM*) were used as controls. A transgene carrying a wild-type *FLC* crossed with FRI–GFP was used as control for FRI–GFP ChIP in *TEX*.

### Western blot analysis
For checking FRI–GFP enrichment in the ChIP experiment, beads were directly boiled in 1×SDS loading buffer at 95 °C for 15 min after immunoprecipitation. For detecting FRI–GFP and FRI–TAP protein levels in NV, 2WV and 4WV plants, nuclei were purified with the same protocol of IP–MS but from 0.3 g of non-cross-linked seedlings. Whole seedlings were grounded and lysed to get total protein extraction for detecting FRI–GFP protein levels in *35S:FRI-GFP* plants. ChIP elution and the extracted nuclear and total proteins were separated on NuPAGE 4–12% Bis-Tris protein gel (Invitrogen, NP0321BOX) and transferred to a PVDF membrane (GE Healthcare Life Sciences). Antibodies against GFP (Roche, 11814460001, dilution: 1:2,000), TAP (Thermo Fisher Scientific, CAB1001, dilution: 1:1,000), H3 (Abcam, ab1791, dilution: 1:5,000) and tubulin (Merck Sigma-Aldrich, T5168, dilution: 1:4,000) were used as primary antibodies. Horseradish peroxidase (HRP)-conjugated secondary antibodies: anti-mouse IgG (GE, NA931, dilution: 1:20,000) and anti-rabbit IgG (GE, NA934, dilution: 1:20,000) were used for protein detection with chemiluminescent substrate (Thermo Fisher Scientific, 34095). Western blot signal was captured in ImageQuant LAS 500.

### RNA-IP assay
Five grams of NV seedlings and 2.5 g of 2WV seedlings were used for each immunoprecipitation. Nuclei were extracted and purified from cross-linked plant material with the same procedure as in the IP–MS experiment. The pellet was resuspended in 1 ml of nuclear lysis buffer (50 mM Tris HCl, pH 7.5, 100 mM NaCl, 1% Triton X-100, 1 mM MgCl₂

and 0.1 mM CaCl₂) and incubated at 37 °C for 10 min with Turbo DNase (Invitrogen, AM1907)[47]. Then SDS was added to a final concentration of 0.1% and NaCl to 150 mM. Following another hour incubation at 4 °C, nuclear lysate was cleared by centrifugation. A 100-µm quantity of the supernatant was saved as the input. Twenty-five microlitres of pre-blocked GFP–Trap magnetic agarose beads (Chromotek, GTMA-20) were incubated with the nuclear supernatant following the same immunoprecipitation and wash procedure as in IP–MS experiment. All the buffers used were supplemented with 0.1 U µl⁻¹ RNaseOUT (Invitrogen, 10777019), 1 mM PMSF and Roche Complete tablets. Then RNA was eluted and purified as previously described[21]. After another DNA digestion with Turbo DNase, reverse transcription with gene specific primers was performed by SuperScript IV reverse transcriptase. Primers for reverse transcription and qPCR are listed in Supplementary Table 2. RNA enrichment was analysed in Microsoft Excel (v.2102, 64-bit) and GraphPad Prism 7. RNA enriched in NV *FRI-GFP* was normalized to NV *FRI* whereas 2WV *FRI-GFP* was normalized to 2WV *FRI* to reduce any possible influence from *COOLAIR* expression variation.

### Reporting summary
Further information on research design is available in the Nature Research Reporting Summary linked to this paper.

### Data availability
Full lists of mass spectrometry are provided as Supplementary Table 1. Raw images of western blots are provided as Supplementary Fig. 1. Other raw images that support the findings of this study are available at https://doi.org/10.11922/sciencedb.01119. Source data are provided with this paper.

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

**Acknowledgements** We thank K. Torney for generating the *FRI-Myc* transgenic line; R. Ishikawa and J. Questa for generating the *frl1-1 FRI*, *flx-2 FRI* and *suf4 FRI* genotypes; P. Shaw for the gift of the U2B″ antibody; I. Lee and H. Kang for the *TAF15b-GFP* plasmids; L. Ma for the *GFP-ELF7* plasmid; Y. Li for initial observations with FRI–GFP microscopy; G. Calder for image analysis expertise; C. Xu for technical assistance on smRNA FISH and immunostaining; X. Fang for constant discussion; and the laboratories of C.D. and

M. Howard for critical input and reading of the manuscript. This work was supported by the UK Biotechnology and Biological Sciences Research Council Institute Strategic Program GEN (BB/P013511/1), a Royal Society Professorship to C.D., an EPSRC Physics of Life grant (EP/T00214X/1) and a European Research Council Advanced Investigator grant (EPISWITCH- 833254).

**Author contributions** P.Z. and C.D. conceived the study and wrote the manuscript. P.Z. performed most of the experiments and all of the data analysis. C.L. generated the *FRI-GFP*, *FRI-TAP* and *35S: FRI-GFP* transgenic lines and performed the FLC–GUS staining. C.D. obtained funding and supervised the work.

**Competing interests** The authors declare no competing interests.

**Additional information**
**Correspondence and requests for materials** should be addressed to Caroline Dean.

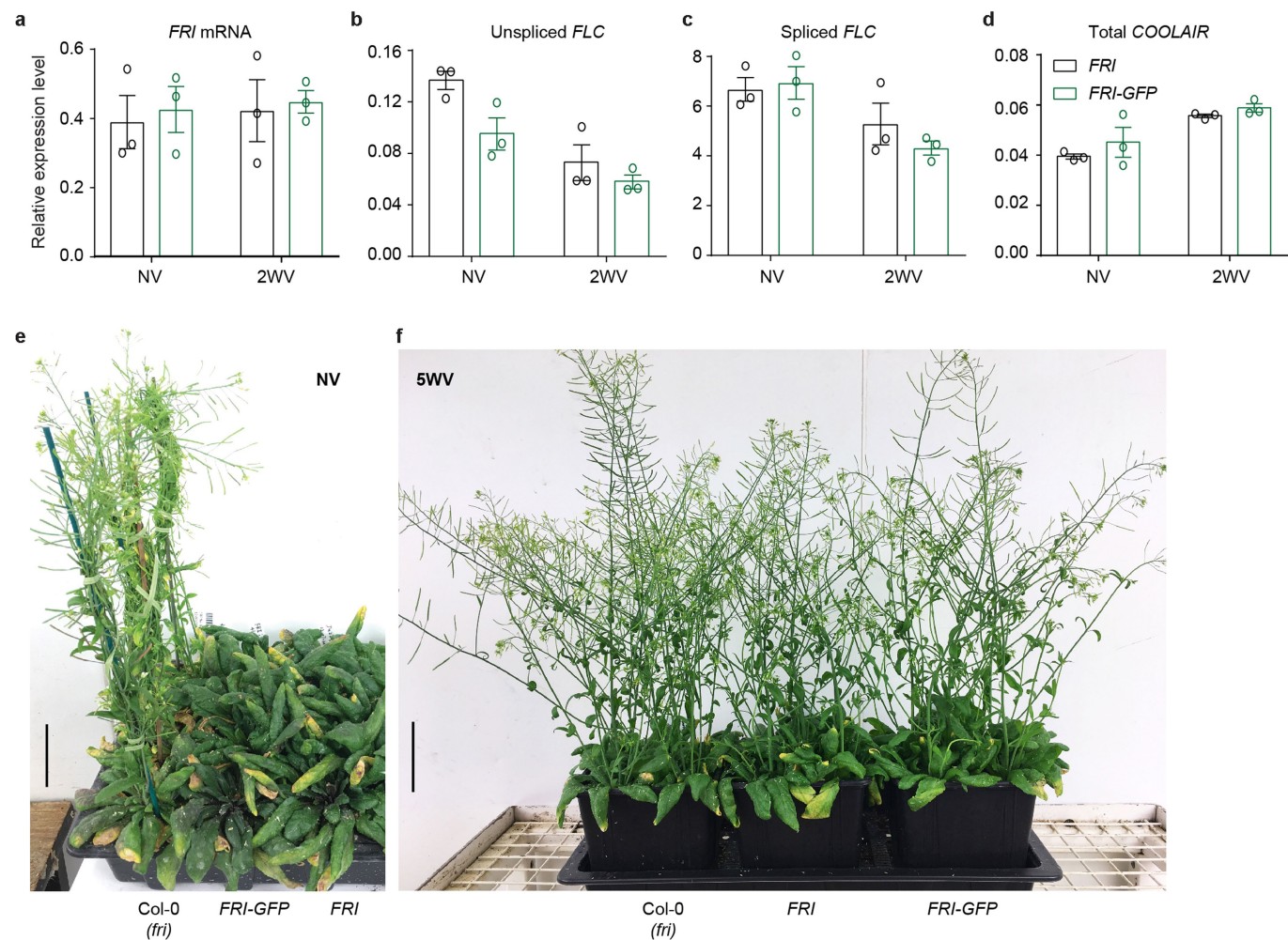

**Extended Data Fig. 1 | Transgenic *FRI-GFP* is functionally equivalent to endogenous *FRI*. a–d**, Relative expression level of *FRI* mRNA (**a**), unspliced *FLC* (**b**), spliced *FLC* (**c**) and total *COOLAIR* (**d**) in Col-0 plants with introgressive *FRI* (*FRI*) or transgenic *FRI-GFP* with its endogenous promoter, measured by RT–qPCR. Plants were given no cold (non-vernalization, NV) or 2 weeks of cold (2 weeks of vernalization, 2WV). Data are presented as mean ± s.e.m. of three independent biological replicates. **e**, **f**, Photographs showing flowering phenotype of Col-0 (*fri*), *FRI-GFP* and *FRI* plants in the warm (NV) or after 5 weeks of cold exposure (5WV). Scale bars, 5 cm.

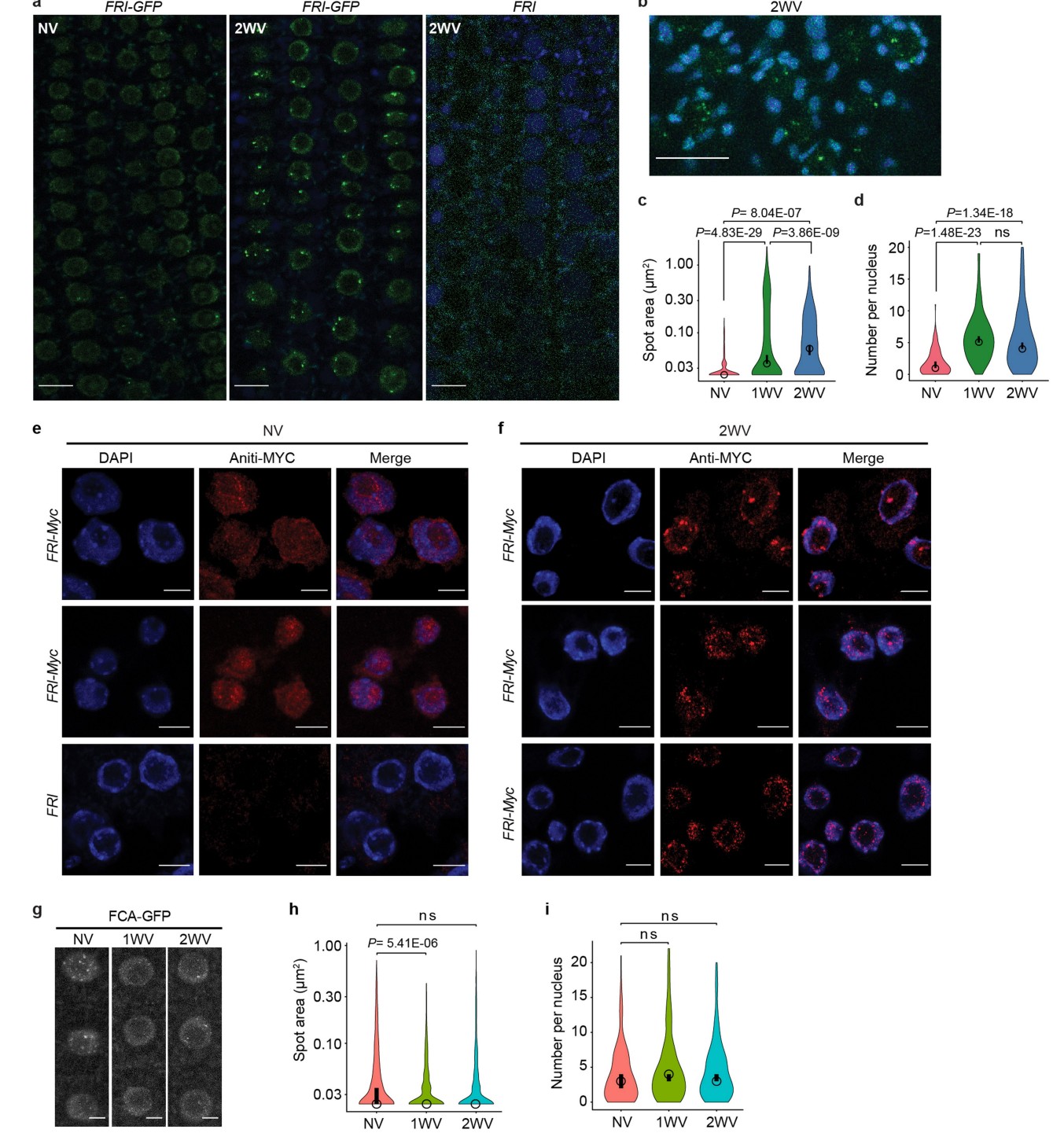

**Extended Data Fig. 2 | Cold promotes FRI nuclear condensates in both root and leaf cells. a**, Single confocal images of NV and 2WV *Arabidopsis* root tip nuclei expressing FRI–GFP (green) with 2WV non-tagged FRI as a negative control. Data are represented of three independent experiments. **b**, Confocal images of 2WV *Arabidopsis* leaf nuclei expressing FRI–GFP (green). Maximum intensity projections of Z-stacks spanning the entire width of the nucleus were applied. Autofluorescence was unmixed with lambda mode (blue) (see Methods). **a**, **b**, Scale bars, 10 μm. **c**, **d**, Quantification of FRI–GFP nuclear condensate area (**c**) and number per nucleus (**d**) in root cells in Fig. 1a. An open circle indicates the median of the data and a vertical bar indicates the 95% confidence interval (CI) determined by bootstrapping. n= 391 (NV), 904 (1WV) and 494 (2WV) condensates (**c**) and n=181 (NV), 144 (1WV) and 130 (2WV) nuclei (**d**). More than 10 plants were analysed. Comparison of mean by two-way ANOVA with adjustment (Sidak's multiple comparisons test). ns, no

significance. **e**, **f**, Representative confocal immunofluorescence images of subnuclear localization of FRI–Myc (red) in NV (**e**) and 2WV (**f**) *Arabidopsis* root cells. Non-tagged FRI was used as a negative control. DNA was stained by DAPI (blue). Scale bars, 5 μm. **g**, Confocal images of *Arabidopsis* root tip nuclei expressing FCA–GFP after 1 week (1WV) and 2 weeks (2WV) of cold treatment. Maximum intensity projections of Z-stacks spanning the entire width of the nucleus were applied. Scale bars, 5 μm. **h**, **i**, Quantification of FCA–GFP nuclear condensate area (**h**) and number per nucleus (**i**) in root cells. An open circle indicates the median of the data and a vertical bar indicates the 95% confidence interval (CI) determined by bootstrapping. Numbers of nuclear condensates measured in (**h**) were 655 (NV), 839 (1WV) and 613 (2WV). Numbers of nuclei analysed in (**i**) were 163 (NV), 148 (1WV) and 126 (2WV). At least 10 plants in each condition were analysed. Comparison of mean by one-way ANOVA with adjustment (Sidak's multiple comparisons test). ns, no significance.

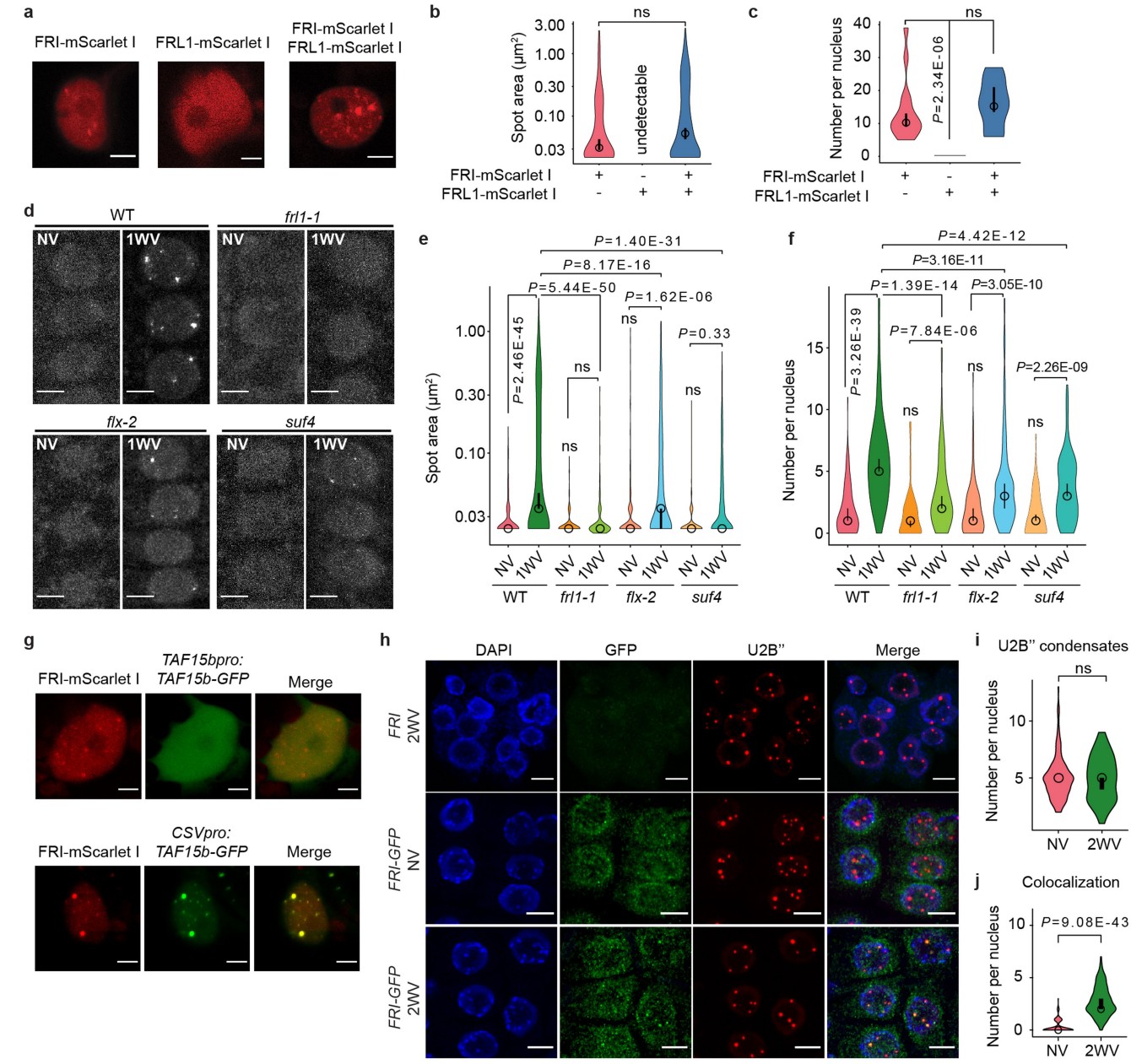

**Extended Data Fig. 3 | FRI associates with FRL1, TAF15b and U2B'' in nuclear condensates in vivo. a**, Confocal microscopic images of tobacco leaf nuclei with expression of FRI–mScarlet I or FRL1–mScarlet I alone or co-expression of both. Data are representative of 5 independent experiments. **b**, **c**, Quantification of FRI–mScarlet I nuclear condensate area (**b**) and number per nucleus (**c**) in tobacco leaf with or without FRL1–mScarlet I co-expressed. An open circle indicates the median of the data and a vertical bar indicates the 95% confidence interval (CI) determined by bootstrapping. Numbers of nuclear condensates measured in (**b**) were 204 (FRI), 0 (FRL1) and 320 (FRI+FRL1). Numbers of nuclei analysed in (**c**) were 15 (FRI), 16 (FRL1) and 20 (FRI+FRL1). Comparison of mean by one-way ANOVA with adjustment (Sidak's multiple comparisons test). ns, no significance. **d**, Confocal microscopic images of FRI–GFP nuclear condensates in the indicated mutants. **e**, **f**, Quantification of FRI–GFP nuclear condensate area (**e**) and number per nucleus (**f**) in the indicated genotypes. An open circle indicates the median of the data and a vertical bar indicates the 95% confidence interval (CI) determined by bootstrapping. Numbers of nuclear condensates measured in

(**e**) were (from left to right) 391, 904, 159, 437, 324, 735, 354 and 448. Numbers of nuclei analysed in (**f**) were (from left to right) 181, 144, 96, 105, 132, 177, 142 and 126. At least 10 plants were analysed. Comparison of mean by one-way ANOVA with adjustment (Sidak's multiple comparisons test). ns, no significance. **g**, Confocal microscopic images of tobacco leaf nuclei expressing FRI–mScarlet I and TAF15b–GFP. TAF15b–GFP was driven either by its endogenous promoter or overexpressed with CSV promoter. Data are representative of 3 independent experiments. **h**, Immunostaining images showing relative subnuclear localization of FRI–GFP (green) to U2B'' (red) in root cells in NV and 2WV conditions. DNA was labelled with DAPI (blue). Non-tagged FRI was used a negative control. **i**, **j**, Quantification of the total number of U2B'' condensates per nucleus (**i**) and those colocalized with FRI–GFP condensates (**j**) in NV and 2WV conditions. An open circle indicates the median of the data and a vertical bar indicates the 95% confidence interval (CI) determined by bootstrapping. n=166 nuclei (NV) and 130 nuclei (2WV). Comparison was via two-tailed *t* test with Welch's correction. ns, no significance. In all the images, scale bars, 5 µm.

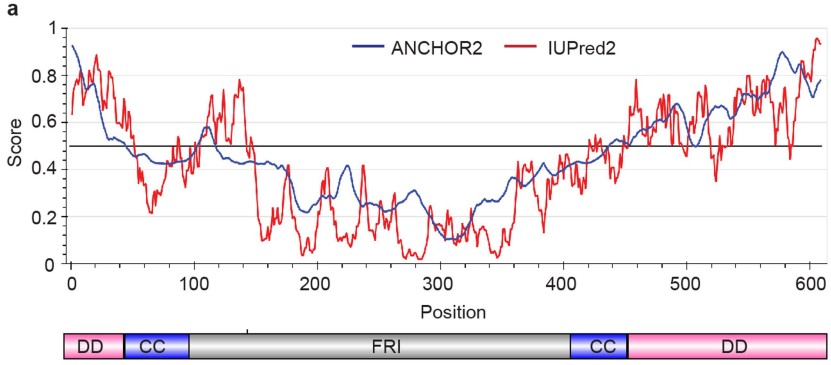

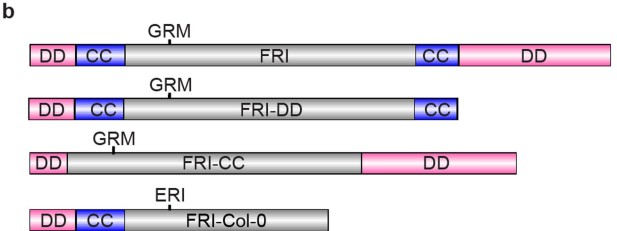

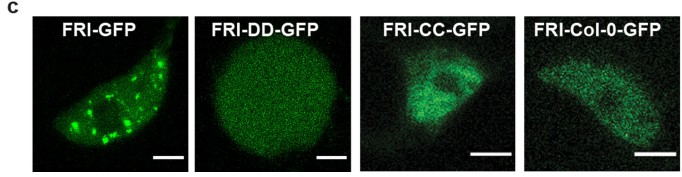

**Extended Data Fig. 4 | The protein domains required for in vivo FRI condensate formation. a**, Prediction of intrinsically disordered regions in FRI protein by IUPred2A. A schematic illustration of FRI domains was shown below. DD, disordered domain, CC, coiled-coil domain and the central domain was in grey. **b**, Schematic illustration of full-length FRI, C-terminal disordered domain deleted FRI (FRI-DD), coiled-coil domain deleted FRI (FRI-CC) and FRI encoded in Col-0 (FRI-Col-0). The mutated amino acids in FRI-Col-0 were indicated[10]. **c**, Subnuclear localization of full-length and truncated FRI–GFP in tobacco leaf nuclei. Images are representative of three independent experiments. Scale bars, 5 μm.

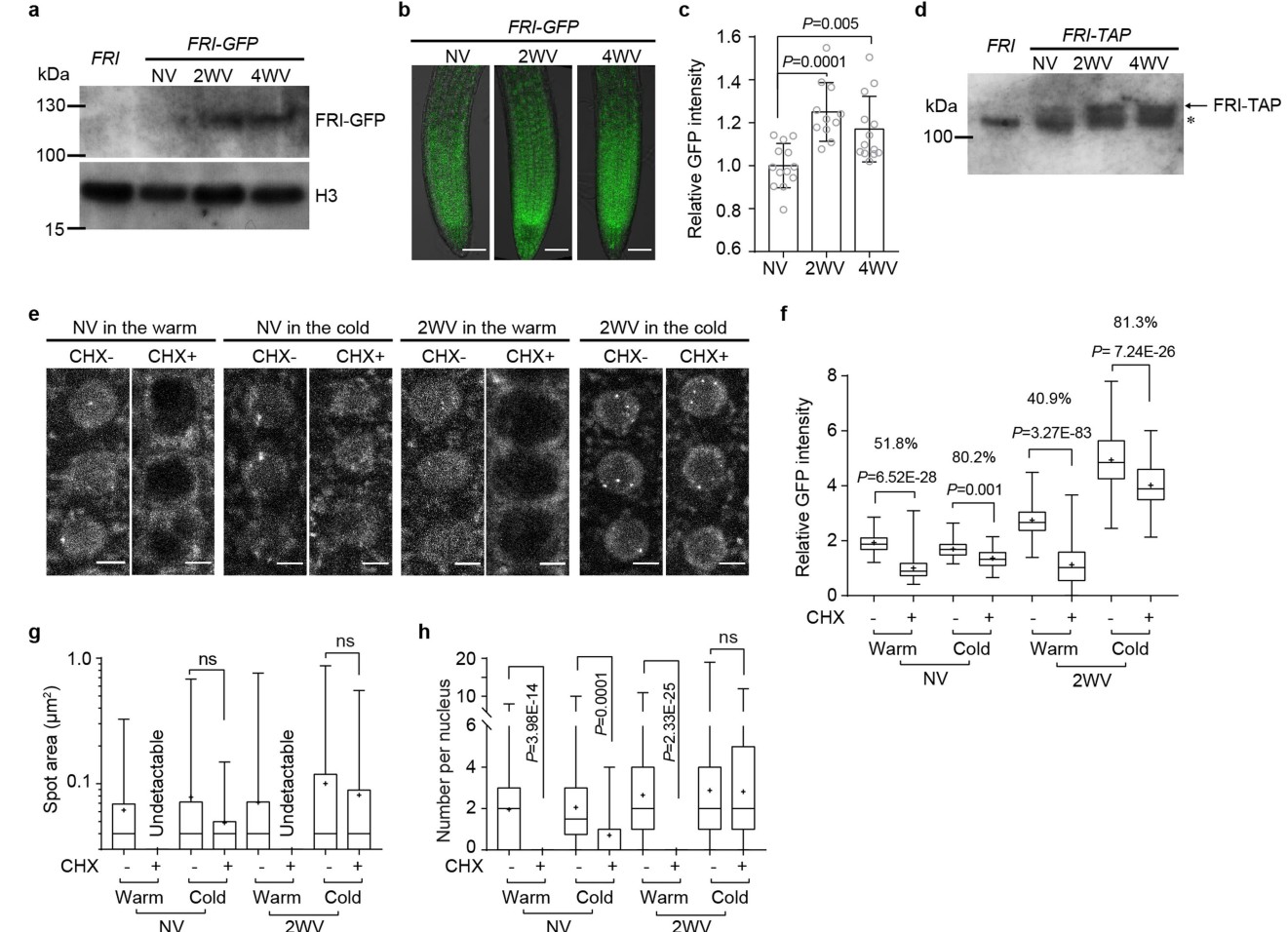

**Extended Data Fig. 5 | Stability of FRI–GFP is increased in the cold.**
**a**, Nuclear FRI–GFP protein level in NV, 2WV and 4WV FRI–GFP transgenic plants as determined by western blots. Non-tagged FRI was used as a negative control. H3 was used as nuclear protein loading control. Data are representative of two independent experiments. For gel source data, see Supplementary Fig. 1. **b**, **c**, Confocal microscopy images of FRI–GFP in root tips of NV, 2WV and 4WV FRI–GFP transgenic plants (**b**) and the quantification of the fluorescence signal (**c**). Scale bars, 50 μm. The fluorescence intensity in cold treated samples is normalized to NV samples. Data are presented as mean ± s.e.m.; n=15 (NV), 12 (2WV) and 13 (4WV) roots. Statistical analysis was via one-way ANOVA with adjustment (Sidak's multiple comparisons test). **d**, Nuclear FRI–TAP protein level in NV, 2WV and 4WV plants expressing FRI–TAP as determined by western blots. Non-tagged FRI was used as a negative control. Asterisks indicate non-specific signals. Data are representative of two independent experiments. For gel source data, see Supplementary Fig. 1.

**e**, Confocal microscopy of *Arabidopsis* root tip nuclei expressing FRI–GFP in NV or 2WV plants after treated with cycloheximide (CHX) in the indicated conditions for 24 h. For example, "NV in the cold" means plants grown in NV were kept in the cold for the 24h CHX treatment. Scale bars, 5 μm. **f**, Quantification of nuclear fluorescence intensity in (**e**). The relative intensity of CHX+ to CHX- in each treatment was indicated by percentage on top. n = 136, 122, 95, 107, 144, 153, 141 and 105 root nuclei (from left to right). **g**, **h**, Box plots showing the distribution of FRI–GFP nuclear condensate area and number in (**e**). Numbers of nuclear condensates measured in (**g**) were 226, 0, 138, 46, 270, 0, 282 and 238 (from left to right) and numbers of nuclei analysed in (**h**) were 113, 127, 66, 80, 94, 185, 95 and 85 (from left to right). **f**–**h**, At least 10 plants were analysed. Centre lines show median, box edges delineate 25th and 75th percentiles, bars extend to minimum and maximum values and '+' indicates the mean value. Mean was compared by one-way ANOVA with adjustment (Sidak's multiple comparisons test).

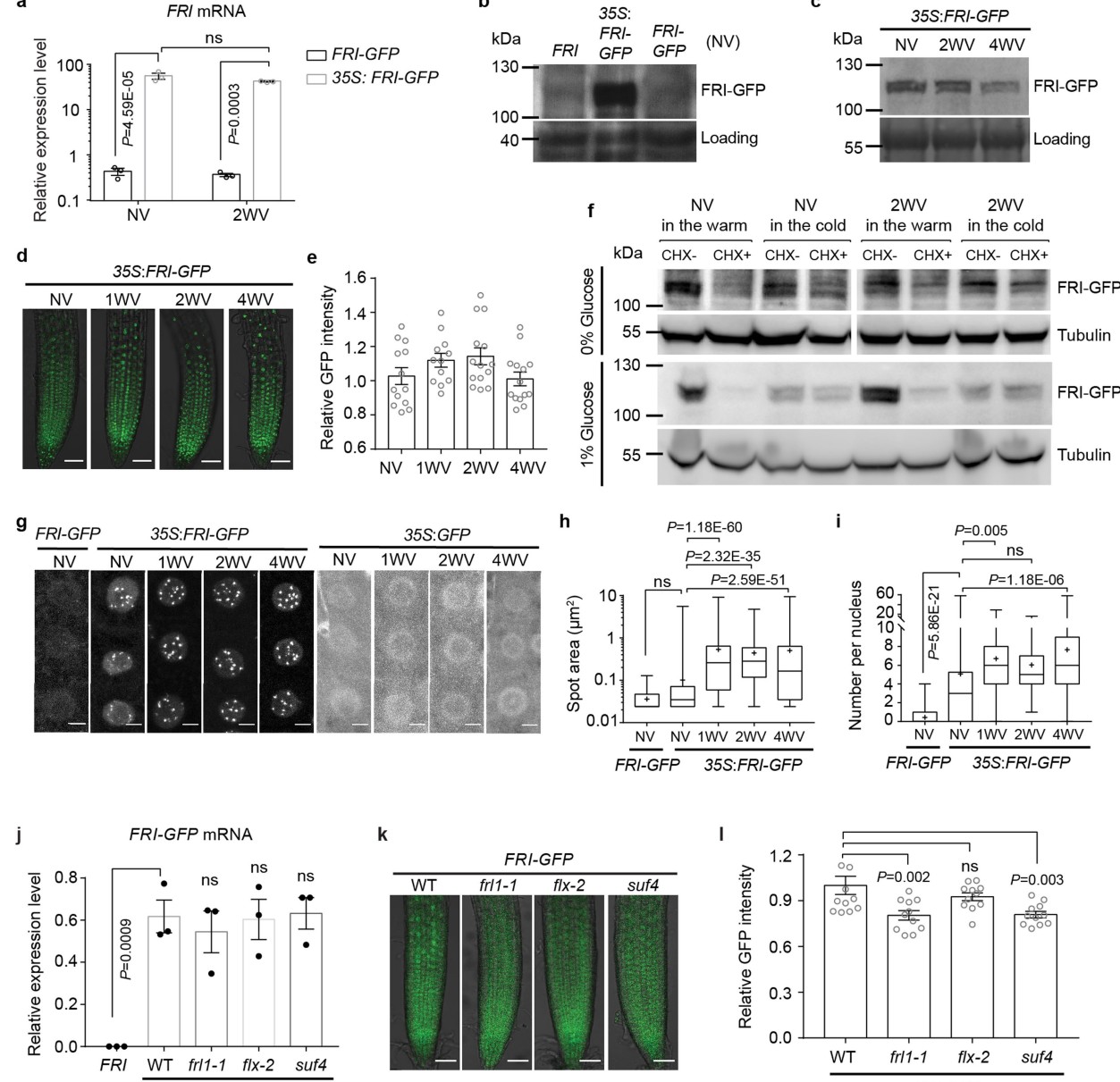

**Extended Data Fig. 6 | FRI–GFP in *35S:FRI-GFP, frl1-1, flx-2* and *suf4*.**
**a**, Relative transcript level of *FRI* mRNA in the indicated plants measured by RT–qPCR. Data are presented as mean ± s.e.m. of three independent biological replicates. Mean was compared by two-way ANOVA with adjustment (Sidak's multiple comparisons test). ns, no significance. **b, c**, Total FRI–GFP protein level in the indicated plants as determined by western blots. A non-specific band (**b**) or ponceau staining (**c**) was used as loading control. Data are representative of two independent experiments. For gel source data, see Supplementary Fig. 1. **d, e**, Confocal microscopic images (**d**) and quantification of GFP fluorescence signal (**e**) of root tips expressing *35S:FRI-GFP*. Scale bars, 50 μm. Data are presented as mean ± s.e.m., *n* = 13 (NV), 12 (1WV), 14 (2WV) and 14 (4WV) roots. **f**, Total FRI–GFP protein level in NV or 2WV *35S:FRI-GFP* plants after treated with cycloheximide (CHX) in the indicated conditions for 24 h as determined by western blots. For example, "2WV in the warm" means plants after 2 weeks of cold exposure were kept in the warm condition for the 24h CHX treatment. Plants were initially grown in growth medium without glucose (see Methods) then were transferred to medium without (0%) (top) or with 1% glucose (bottom) for the 24h CHX treatment. Tubulin was used as control. Data are representative of two independent experiments. For gel source data, see Supplementary Fig. 1. **g**, Confocal microscopy of root tip nuclei in *35S:FRI-GFP*

plants (middle) with *35S:GFP* (right) and NV *FRI-GFP* (left) as control. Maximum intensity projections of Z-stacks spanning the entire width of a nucleus were applied. Scale bars, 5 μm. Images represent 8 independent experiments. **h, i**, Box plots showing the distribution of FRI–GFP nuclear condensate area and number in (**g**). Centre lines show median, box edges delineate 25th and 75th percentiles, bars extend to minimum and maximum values and '+' indicates the mean value. *n* = 114, 1185, 1543, 1276 and 1412 nuclear condensates in (**h**) and 262, 222, 205, 207 and 216 root nuclei in (**i**) (from left to right) were analysed. Comparison of mean was via one-way ANOVA with adjustment (Sidak's multiple comparisons test). ns, no significance. **j**, Expression of *FRI-GFP* in NV plants with the indicated genotype, measured by RT–qPCR. Data are presented as mean ± s.e.m. of three independent biological replicates. One-way ANOVA was used for statistical analysis and *P* value was adjusted by Sidak's multiple comparisons test. ns, no significance. **k**, Representative confocal microscopic images of FRI–GFP root tips in 2WV plants with the indicated backgrounds. Scale bars, 50 μm. **l**, Quantification of the fluorescence intensity in (**k**). Data are represented as mean ± s.e.m., *n* = 14 (WT), 13 (*frl1-1*), 13 (*flx-2*) and 12 (*suf4*) roots. One-way ANOVA was used for statistical analysis and *P* value was adjusted by Dunnett's multiple comparisons test. ns, no significance.

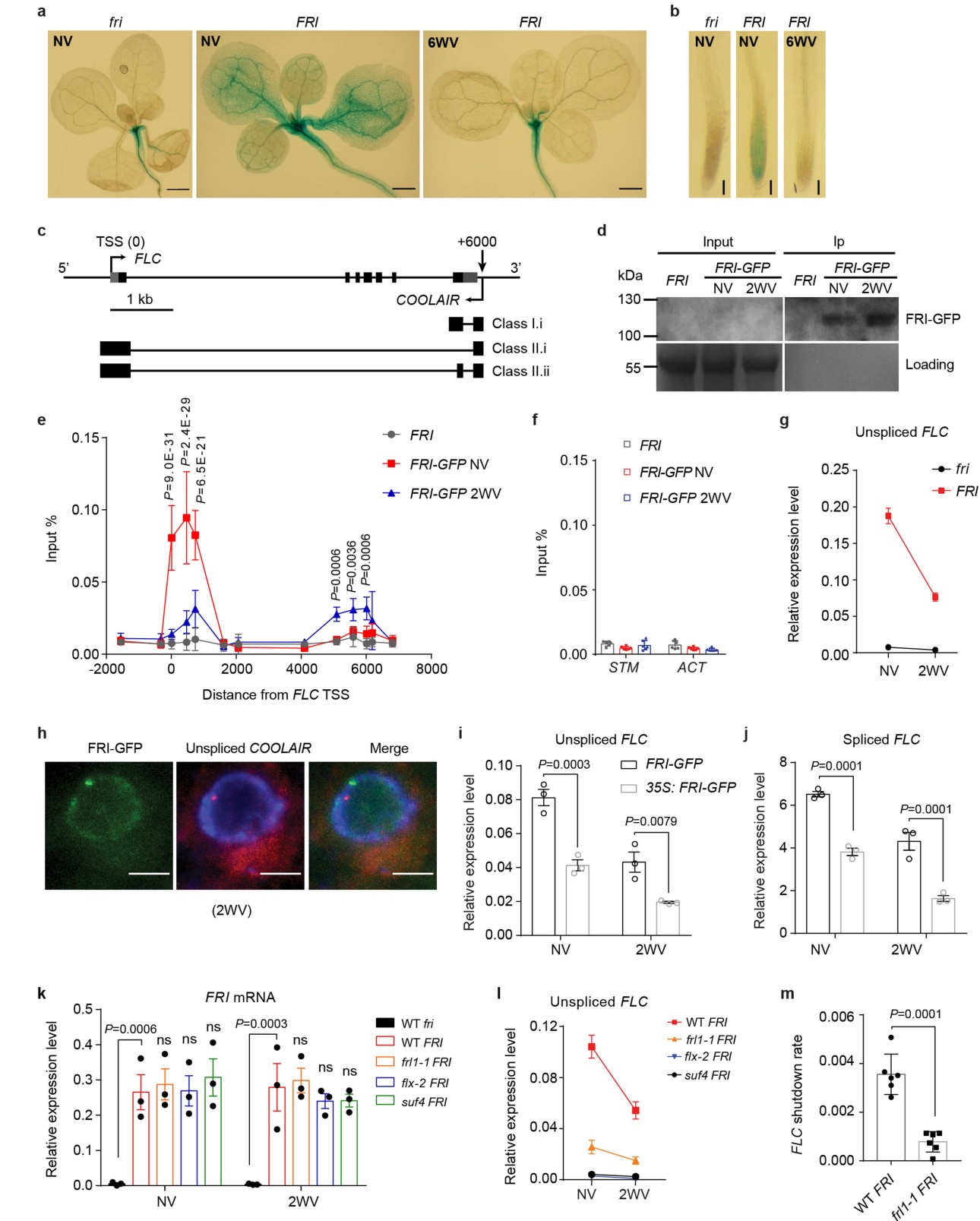

**Extended Data Fig. 7** | See next page for caption.

**Extended Data Fig. 7 | FRI occupancy on the *FLC* promoter is reduced in the cold and correlates with *FLC* transcriptional shutdown. a, b**, Spatial expression patterns of a translational FLC-GUS reporter in aerial parts (**a**) and root tips (**b**) of NV and 6WV plants in *fri* and *FRI* backgrounds. Scale bars, 1 mm (**a**) and 100 μm (**b**). Data represents two independent experiments. **c**, Schematic illustration of *FLC* and *COOLAIR* transcripts at the *FLC* locus. Untranslated regions (UTR) are indicated by grey boxes and exons by black boxes. kb, kilobase. **d**, FRI–GFP was detected after immunoprecipitation by western blots in ChIP experiments. Ponceau staining was used as the loading control. Data are representative of three independent experiments. For gel source data, see Supplementary Fig. 1. **e, f**, FRI–GFP ChIP across the *FLC* locus (**e**), *STM* and *ACT* locus (**f**) in plants expressing FRI–GFP. Non-tagged FRI was used as negative control. The exact distance from TSS referred to (**c**). Data are represented as mean ± s.d. of three independent biological experiments with two technical repeats. Two-way ANOVA was performed with *P* values adjusted by Sidak's multiple comparisons test. **g**, Unspliced *FLC* transcript level in NV and 2WV plants with the indicated backgrounds, measured by RT–qPCR. Data are presented as mean ± s.e.m. of three biologically independent experiments. **h**, Representative images of nuclei expressing FRI-GFP (green) sequentially hybridized with intronic smFISH probes for *COOLAIR* (red). DNA was labelled with DAPI (blue). n= 327 cells. Scale bars, 5 μm. **i, j**, Relative transcript level of unspliced *FLC* and spliced *FLC* in the indicated plants measured by RT–qPCR. Data were presented as mean ± s.e.m. of three independent biological replicates. Two-way ANOVA was performed with *P* values adjusted by Sidak's multiple comparisons test. **k, l**, Relative transcript level of *FRI* mRNA and unspliced *FLC* in NV and 2WV plants with the indicated genotype, measured by RT–qPCR. Data were presented as mean ± s.e.m. of three biologically independent experiments (**k**) and with two technical repeats (**l**). Two-way ANOVA was performed with *P* values adjusted by Sidak's multiple comparisons test. **m**, *FLC* transcriptional shutdown rate indicated by -Slope by Linear Regression of unspliced *FLC* in (**l**). Mean ± s.e.m., n=6 replicates over 3 biologically independent experiments. *P* value was through two-tailed *t* test with Welch's correction.

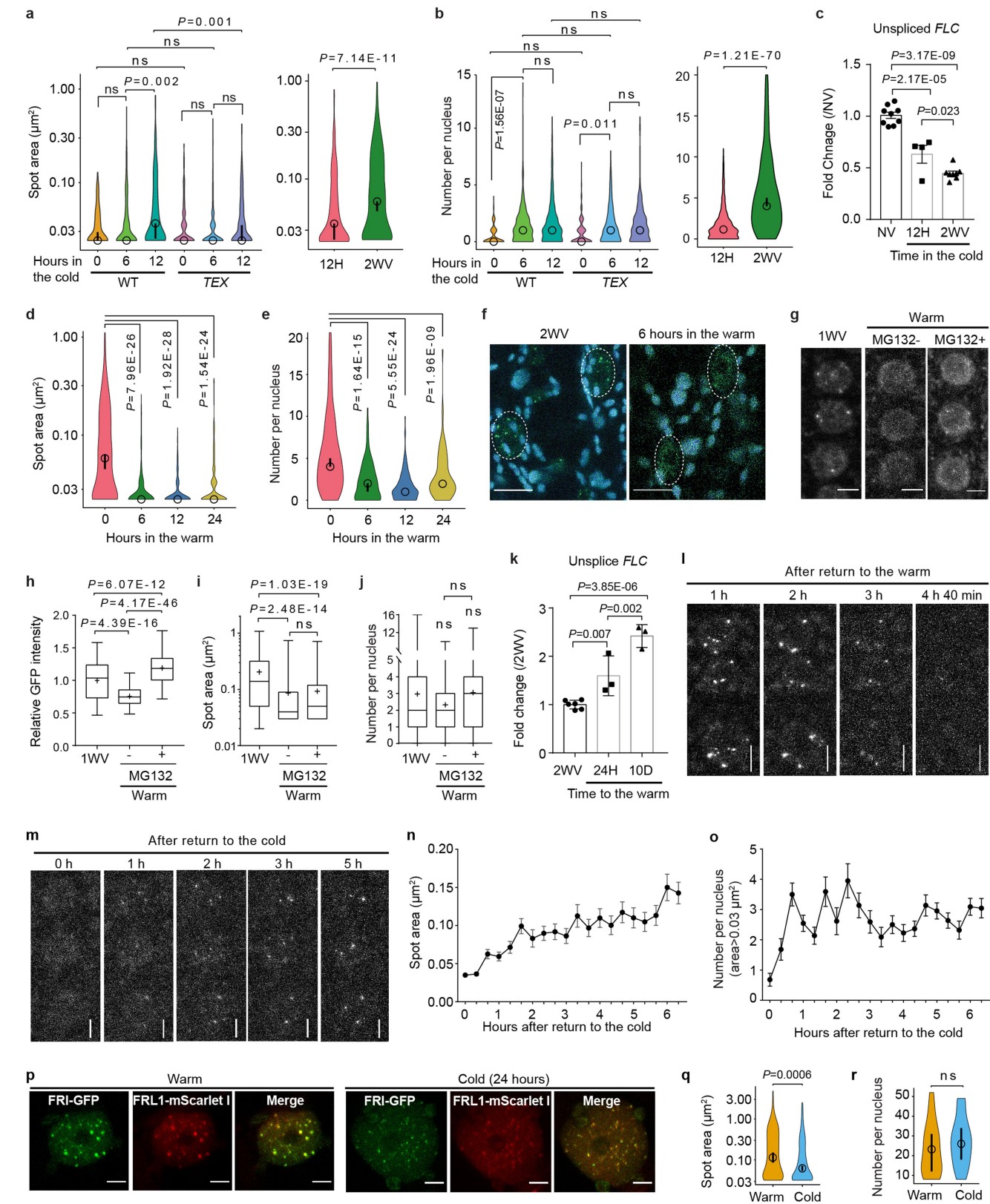

**Extended Data Fig. 8** | See next page for caption.

**Extended Data Fig. 8 | FRI–GFP nuclear condensate dynamics change in response to short-term temperature fluctuations. a**, **b**, Quantification of FRI–GFP nuclear condensate area (**a**) and number per nucleus (**b**) in WT and *TEX* root cells after 0, 6 and 12 h of cold treatment in Fig. 2a. Comparisons between 12 h of cold treatment (12H) and 2WV (same data shown in Extended Data Fig. 2c, d) were presented on the right. An open circle indicates the median of the data and a vertical bar indicates the 95% confidence interval (CI) determined by bootstrapping. $n$ = 114, 380, 505, 180, 218, 346, 505 and 494 nuclear condensates in (**a**) and 262, 291, 303, 214, 205, 292, 303 and 130 root nuclei in (**b**) (from left to right). **c**, Fold change on Unspliced *FLC* transcript level in NV plants after transferred in cold for 12 h (12H) and 2 weeks (2WV) compared to NV. **d**, **e**, Quantification of FRI–GFP nuclear condensate area (**d**) and number per nucleus (**e**) in 2WV wildtype root cells after they were returned to warm for 0, 6, 12 and 24 h in Fig. 2c. An open circle indicates the median of the data and a vertical bar indicates the 95% confidence interval (CI) determined by bootstrapping. Numbers of nuclear condensates measured in (**d**) were 494 (0), 223 (6), 228 (12) and 292 (24) and numbers of root nuclei analysed in (**e**) were 130 (0), 109 (6), 152 (12) and 95 (24). At least 10 plants were analysed for each treatment. **a-e**, One-way ANOVA was performed with *P* values adjusted by Sidak's multiple comparisons test. ns, no significance. **f**, Confocal images of 2WV *Arabidopsis* leaf petiole nuclei expressing FRI–GFP (green) (left) and after transferred to warm conditions for 6 h (right). Maximum intensity projections of Z-stacks spanning the entire width of the nucleus were applied. Autofluorescence was unmixed with lambda mode (blue) (see Methods). Scale bars, 10 μm. Data are representative of three independent experiments. **g**, Confocal microscopy of *Arabidopsis* root tip nuclei expressing FRI–GFP after treated with MG132 in the indicated conditions for 6 h. Plants were exposed to 1 week of cold before (left). Scale bars, 5 μm. **h**, Quantification of nuclear fluorescence intensity in (**g**). Fluorescence intensity was normalized to control. n=127, 122 and 166 root nuclei. **i**, **j**, Box plot showing the distribution of FRI–GFP nuclear condensate area and number in (**g**). n= 227, 139 and 355 nuclear condensates in (**i**) and 81, 60 and 116 root nuclei in (**j**) (from left to right). **h**–**j**, Centre lines show median, box edges delineate 25th and 75th percentiles, bars extend to minimum and maximum values and '+' indicates the mean value. **k**, Fold change of Unspliced *FLC* expression level in 2WV plants after transferred in warm for 24 h (24H) and 10 days (10D) compared to NV. **h**–**k**, One-way ANOVA was performed with *P* values adjusted by Sidak's multiple comparisons test. ns, no significance. **l**, **m**, Time-lapse microscopy of 1WV *Arabidopsis* root-tip nuclei expressing FRI–GFP after transfer to warm temperature (**l**) or return to cold temperature after a 3-hour warm spike (**m**). For each microscopic image maximum intensity projections of Z-stacks spanning the entire width of the nucleus were applied. Scale bars, 5 μm. Data are representative of two independent experiments. **n**, **o**, Quantitative measurement of the area (**n**) and number (**o**) of FRI–GFP nuclear condensates with an area ≥0.03 μm$^2$ at each time point for the time-lapse experiment related to (**m**) and Supplementary Video 2. Data are represented as mean ± s.e.m., $n$ = 22 nuclei. **p**, Confocal microscopic images of tobacco leaf nuclei expressing FRI–GFP and FRL1–mScarlet I before and after 24 h cold. Maximum intensity projections of Z-stacks spanning the entire width of the nucleus were applied. Scale bars, 5 μm. Three independent experiments gave similar results. **q**, **r**, Quantification of FRI–GFP nuclear condensate area (**q**) and number per nucleus (**r**) in tobacco leaf nuclei before and after 24 h of cold exposure. An open circle indicates the median of the data and a vertical bar indicates the 95% confidence interval (CI) determined by bootstrapping. $n$ = 398 (warm) and 353 (cold) nuclear condensates (**q**) and 17 (warm) and 13 (cold) tobacco leaf nuclei (**r**). Two-tailed *t* test with Welch's correction. ns, no significance.

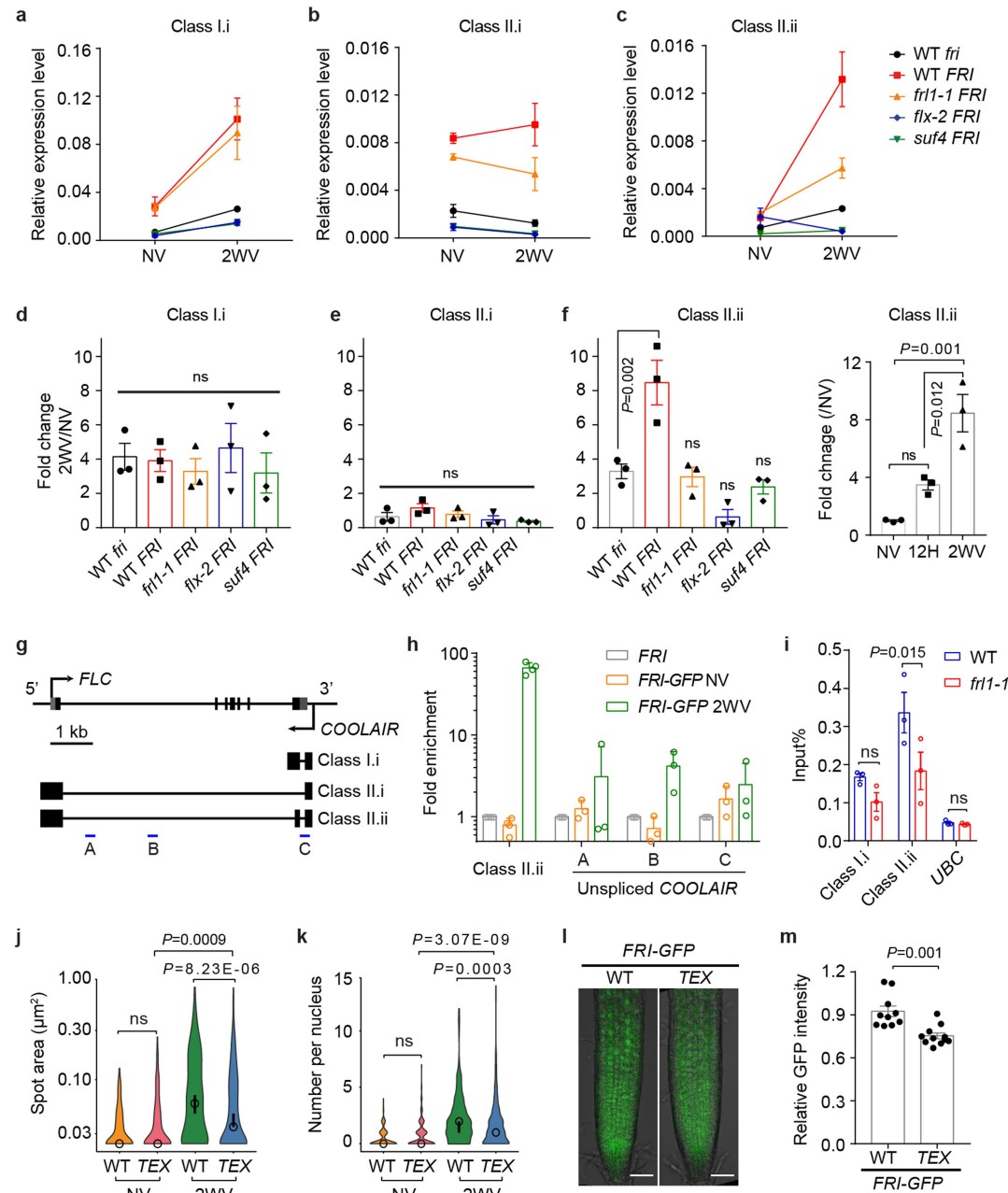

**Extended Data Fig. 9 | The association of FRI and *COOLAIR* is disturbed in FRI-complex mutants. a–c**, Relative transcript level of spliced *COOLAIR* isoforms in NV and 2WV plants with the indicated genotype, measured by RT–qPCR. Data are presented as mean ± s.e.m. of three independent biological replicates. **d–f**, Fold change of spliced *COOLAIR* expression in 2WV plants relative to NV plants in the indicated backgrounds. Fold change of Class II.ii in 12 h cold treated plants (12H) relative to NV was compared to 2WV on the right in (**f**). Data are presented as mean ± s.e.m. of three independent biological replicates. One-way ANOVA was performed with *P* values adjusted by Sidak's multiple comparisons test. ns, no significance. **g**, Schematic illustration of *FLC* and *COOLAIR* transcripts at the *FLC* locus. Untranslated regions (UTR) are indicated by grey boxes and exons by black boxes. kb, kilobase. **h**, RNA-IP assay of unspliced *COOLAIR* enriched by FRI–GFP in NV and 2WV plants. *COOLAIR* enrichment in FRI–GFP is normalized to non-tagged FRI. Data are presented as mean ± s.e.m. of three independent biological replicates. Although there are differences of the fold enrichment between NV and 2WV samples, no statistical significance was detected (two-way ANOVA). Class II.ii was shown as control.

Amplicons for unspliced *COOLAIR* are shown in (**g**) by blue bars. **i**, RNA-IP assay of spliced *COOLAIR* enriched by FRI–GFP in 2WV plants in WT and *frl1-1* backgrounds with *UBC* as control. Data are presented as mean ± s.e.m. of three independent biological replicates. Two-way ANOVA with adjustment by Sidak's multiple comparisons test. ns, no significance. **j, k**, Quantification of FRI–GFP nuclear condensate area (**j**) and number per nucleus (**k**) in root cells in Fig. 3e. An open circle indicates the median of the data and a vertical bar indicates the 95% confidence interval (CI) determined by bootstrapping. Numbers of nuclear condensates measured in (**j**) were 114 (NV-WT), 142 (NV-*TEX*), 435 (2WV-WT) and 500 (2WV-*TEX*). Numbers of nuclei analysed in (**k**) were 262 (NV-WT), 214 (NV-*TEX*), 199 (2WV-WT) and 293 (2WVV-*TEX*). At least 10 plants were analysed for each treatment. One-way ANOVA with adjustment by Sidak's multiple comparisons test. ns, no significance. **l**, Representative confocal microscopic images of FRI–GFP root tips in 2WV WT and *TEX* plants. Scale bars, 50 µm. **m**, Quantification of the fluorescence intensity in (**l**). Data are represented as mean ± s.e.m., n=10 (WT) and 11 (*TEX*) roots. Two-tailed *t* test with Welch's correction.

**Extended Data Table 1 | List of proteins identified by FRI–GFP affinity purification**

| Identified Protein | Accession No. | Mol. Mass (kDa) | No. of Matched Unique Peptides | | | | | |
|---|---|---|---|---|---|---|---|---|
| | | | NV | | | 2WV | | |
| | | | IP1 | IP2 | IP3 | IP1 | IP2 | IP3 |
| FRI | AT4G00650 | 69 | 1 | 13 | 7 | 17 | 22 | 95 |
| FRL1 | AT5G16320 | 53 | 0 | 0 | 0 | 1 | 3 | 11 |
| FRL2 | AT1G31814 | 53 | 0 | 0 | 0 | 0 | 0 | 4 |
| MED12 (Kinase) | AT4G00450 | 249 | 0 | 5 | 0 | 0 | 11 | 20 |
| MED13 (Kinase) | AT1G55325 | 217 | 0 | 8 | 0 | 0 | 9 | 35 |
| CDK8 (Kinase) | AT5G63610 | 53 | 0 | 0 | 0 | 0 | 0 | 5 |
| MED28 (Head) | AT3G52860 | 156 | 0 | 8 | 0 | 0 | 4 | 0 |
| MED16 (Tail) | AT4G04920 | 138 | 1 | 3 | 0 | 0 | 3 | 5 |
| MED23 (Tail) | AT1G23230 | 180 | 0 | 2 | 0 | 0 | 3 | 5 |
| MED14 (Tail) | AT3G04740 | 186 | 0 | 0 | 0 | 0 | 0 | 4 |
| MED3/27 (Tail) | AT3G09180 | 45 | 0 | 0 | 0 | 0 | 2 | 1 |
| MED33B (Tail) | AT2G48110 | 140 | 0 | 0 | 0 | 0 | 0 | 1 |
| TAF15b | AT5G58470 | 42 | 0 | 6 | 5 | 3 | 3 | 11 |
| TFIIB | AT2G41630 | 34 | 0 | 0 | 0 | 0 | 8 | 8 |
| TBP1 | AT3G13445 | 22 | 0 | 0 | 0 | 0 | 0 | 3 |
| TAF5 | AT5G25150 | 74 | 0 | 0 | 0 | 0 | 1 | 0 |
| TAF2 | AT1G73960 | 156 | 0 | 0 | 0 | 0 | 0 | 6 |
| TAF4 | AT5G43130 | 93 | 0 | 0 | 0 | 0 | 0 | 2 |
| TFIID subunit | AT5G65540 | 69 | 0 | 0 | 0 | 0 | 0 | 2 |
| TFIIE, alpha subunit | AT1G03280 | 54 | 0 | 0 | 0 | 0 | 1 | 2 |
| IWS1 | AT1G32130 | 57 | 0 | 3 | 1 | 0 | 4 | 3 |
| TFIIS | AT2G38560 | 42 | 0 | 0 | 0 | 1 | 2 | 2 |
| NRPB11 | AT3G52090 | 14 | 0 | 0 | 0 | 0 | 2 | 0 |
| NRPB7 | AT5G59180 | 19 | 0 | 0 | 0 | 0 | 2 | 0 |
| NRPB9B | AT4G16265 | 13 | 0 | 0 | 0 | 0 | 2 | 0 |
| NRPB8B | AT3G59600 | 17 | 0 | 0 | 0 | 0 | 0 | 4 |
| NRPB5 | AT3G22320 | 24 | 0 | 0 | 0 | 0 | 0 | 4 |
| NRPB12 | AT5G41010 | 6 | 0 | 0 | 0 | 0 | 0 | 1 |
| ELF8, Ctr9 (PAF1C) | AT2G06210 | 124 | 0 | 3 | 0 | 3 | 7 | 10 |
| ELF7, Paf1 (PAF1C) | AT1G79730 | 67 | 0 | 2 | 0 | 1 | 4 | 5 |
| VIP5, Rtf1 (PAF1C) | AT1G61040 | 71 | 0 | 3 | 0 | 0 | 2 | 11 |
| CDC73 (PAF1C) | AT3G22590 | 48 | 0 | 0 | 0 | 0 | 6 | 4 |
| VIP4, Leo1 (PAF1C) | AT5G61150 | 72 | 0 | 1 | 0 | 0 | 0 | 1 |
| WDR5a | AT3G49660 | 35 | 0 | 0 | 0 | 0 | 0 | 7 |
| REF6 | AT3G48430 | 153 | 0 | 2 | 0 | 0 | 2 | 9 |
| ATX2 | AT1G05830 | 123 | 0 | 0 | 0 | 0 | 0 | 1 |
| RCF1 | AT1G20920 | 133 | 0 | 3 | 0 | 0 | 8 | 8 |
| PRP31 | AT1G60170 | 53 | 0 | 0 | 0 | 0 | 3 | 4 |
| U2B" | AT2G30260 | 26 | 0 | 0 | 0 | 2 | 1 | 1 |
| PRP4 | AT2G41500 | 62 | 0 | 0 | 2 | 0 | 5 | 19 |
| U1-70K | AT3G50670 | 50 | 0 | 7 | 0 | 2 | 9 | 0 |
| Cwf21 | AT3G49601 | 68 | 0 | 0 | 0 | 0 | 4 | 13 |
| SMP2 | AT4G37120 | 62 | 0 | 0 | 0 | 0 | 1 | 2 |
| PRP18A | AT1G03140 | 48 | 0 | 0 | 0 | 0 | 2 | 3 |
| Splicing factor PWI domain-containing protein | AT2G29210 | 100 | 0 | 0 | 0 | 0 | 1 | 2 |
| U4/U6, U5 small nuclear ribonucleoprotein | AT5G57370 | 26 | 0 | 0 | 1 | 0 | 1 | 3 |
| RDM16 | AT1G28060 | 89 | 0 | 1 | 0 | 0 | 3 | 10 |

Data from three independent experiments are listed as IP1, IP2 and IP3. IP, immunoprecipitation. Subunits from the Mediator complex are listed with the modules they belong to[13]. NV (not vernalized), plants were grown in the warm; 2WV (2 weeks of vernalization), plants were given two weeks of cold treatment.

# Reporting Summary

## Statistics

For all statistical analyses, confirm that the following items are present in the figure legend, table legend, main text, or Methods section.

| n/a | Confirmed | |
|---|---|---|
| ☐ | ☒ | The exact sample size (*n*) for each experimental group/condition, given as a discrete number and unit of measurement |
| ☐ | ☒ | A statement on whether measurements were taken from distinct samples or whether the same sample was measured repeatedly |
| ☐ | ☒ | The statistical test(s) used AND whether they are one- or two-sided *Only common tests should be described solely by name; describe more complex techniques in the Methods section.* |
| ☒ | ☐ | A description of all covariates tested |
| ☐ | ☒ | A description of any assumptions or corrections, such as tests of normality and adjustment for multiple comparisons |
| ☐ | ☒ | A full description of the statistical parameters including central tendency (e.g. means) or other basic estimates (e.g. regression coefficient) AND variation (e.g. standard deviation) or associated estimates of uncertainty (e.g. confidence intervals) |
| ☐ | ☒ | For null hypothesis testing, the test statistic (e.g. *F*, *t*, *r*) with confidence intervals, effect sizes, degrees of freedom and *P* value noted *Give P values as exact values whenever suitable.* |
| ☒ | ☐ | For Bayesian analysis, information on the choice of priors and Markov chain Monte Carlo settings |
| ☒ | ☐ | For hierarchical and complex designs, identification of the appropriate level for tests and full reporting of outcomes |
| ☒ | ☐ | Estimates of effect sizes (e.g. Cohen's *d*, Pearson's *r*), indicating how they were calculated |

*Our web collection on statistics for biologists contains articles on many of the points above.*

## Software and code

Policy information about availability of computer code

| Data collection | For LCMS analysis, an Orbitrap Fusion™ Tribrid™ mass spectrometer (Thermo Fisher Scientific) equipped with an UltiMate™ 3000 RSLCnano LC system (Thermo Fisher Scientific) was used; all QPCR reactions were run with LightCycler® 480; all confocal images and movies were collected on Zeiss LSM780 confocal microscope, Zeiss LSM880 confocal microscope and Zeiss Elyra PS as indicated in the Methods; Western blot signal was captured in ImageQuant LAS 500. |
|---|---|
| Data analysis | Mascot server 2.7 (Matrix Science), Scaffold 4 (4.8.4), Microsoft Excel (version 2102, 64-bit), GraphPad Prism 7, Fiji-ImageJ 1.52i (Java 1.8.0_172, 64-bit) ), and Zen Black and Zen Blue (2012). Web service: PlotsOfData (https://huygens.science.uva.nl/PlotsOfData/) and IUPred2A (https://iupred2a.elte.hu/). |

For manuscripts utilizing custom algorithms or software that are central to the research but not yet described in published literature, software must be made available to editors and reviewers. We strongly encourage code deposition in a community repository (e.g. GitHub). See the Nature Portfolio guidelines for submitting code & software for further information.

## Data

Policy information about availability of data

All manuscripts must include a data availability statement. This statement should provide the following information, where applicable:

- Accession codes, unique identifiers, or web links for publicly available datasets
- A description of any restrictions on data availability
- For clinical datasets or third party data, please ensure that the statement adheres to our policy

Full lists of mass spectrometry are provided as Supplementary Table 1. Uncropped images of western blots are provided as Supplementary Figure 1. Other raw

# Field-specific reporting

Please select the one below that is the best fit for your research. If you are not sure, read the appropriate sections before making your selection.

☒ Life sciences ☐ Behavioural & social sciences ☐ Ecological, evolutionary & environmental sciences

For a reference copy of the document with all sections, see nature.com/documents/nr-reporting-summary-flat.pdf

# Life sciences study design

All studies must disclose on these points even when the disclosure is negative.

| | |
|---|---|
| Sample size | The sample size and the results of statistical analyses are described in the relevant figure legends. No statistical approach was used to predetermine sample size. Sample sizes were determined based on previous publications on similar experiments. The determined sample size was adequate as the differences between experimental groups was significant and reproducible.<br>Previous publications considered to determine sample size:<br>IP-MS analysis (https://doi.org/10.1038/s41586-019-1165-8)<br>RNA expression analysis (https://doi.org/10.1038/s41586-020-2485-4)<br>Microscopy and image quantification (https://doi.org/10.1038/s41586-019-1165-8; https://doi.org/10.1038/s41586-020-2485-4)<br>FRAP (Doi: 10.1007/978-1-4939-7318-7_26; https://doi.org/10.1038/s41586-019-1165-8)<br>Time-lapse imaging (https://doi.org/10.1038/s41586-019-1165-8)<br>Drug treatment (https://doi.org/10.1038/s41586-020-2485-4)<br>smFISH and immunofluorescence (DOI:10.21769/BioProtoc.2240; https://doi.org/10.1038/ncomms13031)<br>CHIP and qPCR (DOI: 10.1126/science.aan1121)<br>Western blot (https://doi.org/10.1038/s41586-019-1165-8; https://doi.org/10.1038/s41586-020-2485-4)<br>RNA-IP (https://doi.org/10.1038/s41586-019-1165-8) |
| Data exclusions | No data was excluded from analysis. |
| Replication | All key experimental findings were reproduced in more than three independent biological repeats with multiple technical replicates. All data except for the immunoblots are representative of at least three independent biological replicates. The immunoblot data are representative of two independent biological replicates. Similar results were obtained from independent biological replicates. Main conclusions were confirmed in different assays, including genetic assays in different mutant backgrounds or overexpression, bioimaging and immunoblots with transgenic lines carrying different tags. |
| Randomization | Plants of different genotypes were grown side by side to minimize unexpected environmental variations during growth and experimentation. Different treatments were carried out in parallel, with minimum covarying factors. Seedlings at the same developmental stage were collected and assessed randomly for each genotype/treatment.<br><br>For IP-MS, RNA-IP and CHIP, multiple seedlings were randomly collected from different plates for each replicate. For RNA expression/protein accumulation analysis and bio-imaging, multiple, randomly selected plants were collected from a plate for each replicate. |
| Blinding | Blinding was not applicable for this study because plants grown at different temperature conditions need to be collected at different growing time points and require specific handling temperatures. |

# Reporting for specific materials, systems and methods

We require information from authors about some types of materials, experimental systems and methods used in many studies. Here, indicate whether each material, system or method listed is relevant to your study. If you are not sure if a list item applies to your research, read the appropriate section before selecting a response.

## Materials & experimental systems

| n/a | Involved in the study |
|---|---|
| ☐ | ☒ Antibodies |
| ☒ | ☐ Eukaryotic cell lines |
| ☒ | ☐ Palaeontology and archaeology |
| ☒ | ☐ Animals and other organisms |
| ☒ | ☐ Human research participants |
| ☒ | ☐ Clinical data |
| ☒ | ☐ Dual use research of concern |

## Methods

| n/a | Involved in the study |
|---|---|
| ☒ | ☐ ChIP-seq |
| ☒ | ☐ Flow cytometry |
| ☒ | ☐ MRI-based neuroimaging |

# Antibodies

| | |
|---|---|
| Antibodies used | For Western blot:<br>Primary antibodies: anti-GFP (Roche, 11814460001, a mixture of clones 7.1 and 13.1, dilution 1:2,000), anti-TAP (Thermo Fisher, CAB1001, polyclonal, dilution 1:1,000), anti-H3 (Abcam, ab1791, polyclonal, dilution 1:5,000), anti-Tubulin (Merck Sigma-Aldrich, T5168, monoclonal, dilution 1:4,000); secondary antibodies: Mouse IgG HRP Linked Whole Antibody (GE, NXA931; 1:20,000), Rabbit IgG HRP Linked Whole Antibody (GE, NA934; 1:20,000).<br>For immunofluorescence:<br>Primary antibodies: anti-c-Myc (Sigma, M5546, clone 9E10, dilution: 1:125), anti-GFP (Abcam, ab290, dilution: 1:500), anti-U2B'' (4G3, a gift from Prof. Peter Shaw, originally obtained from Euro-diagnostica B.V., Apeldoorn, Netherlands, dilution: 1:20); secondary antibodies: Alexa Fluor 488 anti-Rabbit secondary antibody (Thermo Fisher, A-11008, dilution: 1:200) and Alexa Fluor 555 anti-mouse secondary antibody (Thermo Fisher, A-21424, dilution: 1:200).<br>For immunoprecipitation:<br>anti-GFP (Abcam, ab290, dilution: 1:400), GFP-Trap magnetic agarose beads (Chromotek, GTMA-20, dilution: 1:40). |
| Validation | anti-U2B'', 4G3, a gift from Prof. Peter Shaw, was validated by many previous studies for marking Cajal bodies.<br><br>Validation statements and relevant citation of all the other antibodies are available from the manufacturers and most of them are also validated in this manuscript:<br><br>anti-GFP (Roche, 11814460001)-https://www.sigmaaldrich.com/catalog/product/roche/11814460001?lang=en®ion=GB; also validated in Extended data Figs. 5a, 6b, and 7d where non-tagged FRI shows no bands.<br>anti-TAP (Thermo Fisher, CAB1001)-https://www.thermofisher.com/antibody/product/TAP-Tag-Antibody-Polyclonal/CAB1001; also validated in Extended data Fig.5d with non-tagged FRI showing no FRI-TAP band.<br>anti-H3 (Abcam, ab1791)-https://www.abcam.com/histone-h3-antibody-nuclear-marker-and-chip-grade-ab1791.html; anti-Mouse IgG (GE, NXA931)-https://www.sigmaaldrich.com/catalog/product/sigma/genxa9311ml?lang=en®ion=GB&gclid=CjwKCAiA9bmABhBbEiwASb35V0Lm5FrPvLOvJgcZsOjzs756hT2Pxl9x1HuHc_LyTQPZO_UXApnPWRoC5rMQAvD_BwE;<br>anti-Tubulin (Merk Sigma-Aldrich, T5168)-https://www.sigmaaldrich.com/GB/en/product/sigma/t5168?context=product; anti-Rabbit IgG (GE, NA934)-https://www.sigmaaldrich.com/catalog/product/sigma/gena9341ml?lang=en®ion=GB&gclid=CjwKCAiA9bmABhBbEiwASb35Vw9OIG58qJ0JS9_VrsKOXrp6JXUTUKV8iLDXc61BRdXi_CByayrNBRoCHjsQAvD_BwE;<br>anti-c-Myc (Sigma, M5546)-https://www.sigmaaldrich.com/catalog/product/sigma/m5546?lang=en®ion=GB; also validated in Extended data Fig. 2e with non-tagged FRI showing no signal.<br>anti-GFP (Abcam, ab290)-https://www.abcam.com/gfp-antibody-ab290.html; also validated in Extended data Fig.3h with negative control showing no signal and in Extended data Fig. 7d-f with no enrichment in negative control.<br>Alexa Fluor 488 anti-Rabbit secondary antibody (Thermo Fisher, A-11008)-https://www.thermofisher.com/antibody/product/Goat-anti-Rabbit-IgG-H-L-Cross-Adsorbed-Secondary-Antibody-Polyclonal/A-11008;<br>Alexa Fluor 555 anti-mouse secondary antibody (Thermo Fisher, A-21424)-https://www.thermofisher.com/antibody/product/Goat-anti-Mouse-IgG-H-L-Highly-Cross-Adsorbed-Secondary-Antibody-Polyclonal/A-21424;<br>GFP-Trap magnetic agarose beads (Chromotek, GTMA-20)-https://www.chromotek.com/products/detail/product-detail/gfp-trap-magnetic-agarose/; also validated in Fig. 3b and Extended data Fig. 9h with no enrichment detected in negative control. |

