## [Peer Review File · Nature]

Manuscript Title: Cold-induced *Arabidopsis* FRIGIDA nuclear condensates for *FLC* repression

Redactions – unpublished data

Reviewer Comments & Author Rebuttals

Reviewer Reports on the Initial Version:

Referee #1 (Remarks to the Author):

Zhu et al report data connecting the cold-induced formation of FRI-containing nuclear condensates to the repression of *FLC* in *Arabidopsis*. Using a line with FRI-GFP at endogenous expression levels they perform IP-MS +/- cold and find a number of interesting proteins related to transcription that all go up upon cold-induction. Surprisingly, FRI peptide counts also go up upon cold-induction even though the expression of its target gene (*FLC*) is reduced. They inspect these lines under a microscope and observe nuclear condensates which become more numerous and prominent upon cold-induction. They note a minute-scale recovery after photo-bleaching, which is significantly slower than a previous condensate they reported in 2019. Knockout of several FRI interaction partners (although some of these were not present in the IP) decreased cold-induced condensate accumulation. They find that FRI condensates colocalize with Cajal body constituents, which were present in the IP-MS. Deletion of a c-terminal domain or a coil-coil domain prevented FRI condensates. Using *FLC* intron probes, the authors find no colocalization between RNA FISH and GFP-FRI condensates. They then test cold-warm cycles and show that FRI condensates and *FLC* and *COOLAIR* expression are dynamic and reversible within 6-12 hours and suggest a correlation between these phenomena. They then focus on *COOLAIR*, a lncRNA induced by cold. RNA-IP showed FRI-GFP enriched on a specific splice form of *COOLAIR*. Finally using a line with defective cold-induction of *COOLAIR* they show that FRI condensates have a small reduction in cold-induced accumulation.

The paper presents interesting data, but is difficult to follow for a number of reasons. On a practical level, the data in the figures, text, and extended data are not presented in the same order making the reader go back and forth between threads of data and narrative and bouncing between multiple figures. This needs to be fixed. On the conceptual level, the authors do not seem to have their minds made up on what the emphasis of this paper should be. There are at least three themes being presented: environment sensing of biomolecular condensates, the requirement for heterotypic interactions leading to greater control of condensate formation, and lncRNA driven condensate formation. The paper bounces between these themes and the resulting paper is difficult to follow and piece together. Overall, the paper reports several interesting findings that do not form a cohesive whole that would be expected for publication in *Nature*. Major revisions would be required for publication.

Major comments:

- 1) The emphasis at the outset of the text is on solving the mystery of “how autumn is distinguished from winter” but the connection to this big question seems lost after the abstract. The title says that this sequestration is for “Autumn sensing” but that does not come through in the text and it is not clear which experiment tests “Autumn sensing”
 - a. Fig 3b and g. What are the expression levels after 2 weeks of cold? Is the *FLC* shutdown and *COOLAIR* induction complete after 12 hours in cold or is it just beginning?
 - b. Fig 3d and h. What are the expression levels after 2 weeks in warm? Is *FLC* reactivation and *COOLAIR* shutdown complete after 12 hours in warm or is it just beginning?
 - c. These expression timescales and the timescale of condensate accumulation and dissolution compared to the temperature fluctuations in field studies during autumn are relevant to the argument that these condensates are regulatory or act as buffers against transient spikes in

temperature. If 6-12 hours of cold or warm are sufficient to fully reactivate or silence FLC/COOLAIR it is difficult to imagine how this buffers from natural temperature fluctuations. Is the idea that you need long-term sustained suppression or activation of FLC to get vernalization? What is that time-window of sustained cold or warmth required?

- 2) There is a disconnect on what exactly is "temperature-dependent." The title and several sections in the text suggest that FRI condensates are temperature-dependent, but really it is COOLAIR expression and FRI-stability that are temperature-dependent. The authors demonstrate several times that FRI can form condensates at warm temperatures. This is more of a disconnect because there are many examples of protein condensation directly sensitive to environmental states (temperature, pH, redox state, etc.) without a secondary intermediate.
- 3) How does cold induce COOLAIR expression?
- 4) How does cold stabilize the FRI protein?
- 5) If FRI is sequestered into gel-like condensates sufficient to prevent activation of FLC, how does FRI activate COOLAIR in 2WV? What privileges COOLAIR isoform I.ii during 2WV? What prevents FRI from activation of COOLAIR in NV?
- 6) Is FRI forming its own condensate or is it becoming a constituent of cajal bodies or some other nuclear body?
- 7) The observation that *frl1*, *flx-2*, and *suf4* are required for cold enhancement is interesting, but it is unclear whether this is a direct effect of heterotypic interactions or an indirect effect. In particular, if these are so essential for heterotypic interactions forming a condensate why were these not picked up in the IP experiments? The explanation in the text of highly dynamic interactors does not make sense given that these IPs were performed in lysates of crosslinked cells, which should stabilize dynamic interactions.
- 8) Quantification of results in figure 2d is needed.
- 9) Quantification as reported in 1a-b for the mutants in 1c. The effect size of TEX on condensate accumulation is modest (fig 4e-f). Will be important to see how these compare to the mutants in 1c.
- 10) How does COOLAIR induce condensate accumulation? Is it a constituent of the FRI condensates? Do the authors expect that FISH probes against spliced COOLAIR would localize to the FRI-GFP condensates in 2WV? What about in the few FRI condensates observed in NV?
- 11) Is COOLAIR isoform expression sufficient to cause accumulated FRI condensates and does that lead to a decrease in FLC expression? Can you then show that either the FRI with CC or c-term deletion does not form condensates upon COOLAIR and FLC expression is not suppressed? Presumably FLC expression will be impacted by FRI mutations, but do you not get further repression upon 2 weeks of cold?
- 12) Line 144-145: The slower FLC shutdown due to *frl1-1* mutant is interesting, but it also suggests that you can get FLC shutdown without FRI sequestration into condensates. What would be the mechanism of FLC shutdown in the absence of FRI condensates? Is the accumulation of FRI condensates required for FLC shutdown or does it make it happen faster?
- 13) Would be more helpful to show extended figure 8c for the TEX line with the full set of primers and data plotted as %input.
- 14) How does this all interplay with Polycomb which the authors have previously proposed as the mechanism for FLC silencing and of COOLAIR function? What are the relative timescales of FRI-condensate accumulation and polycomb deposition?

Minor comments:

- 1) Line 43: The authors should provide a spreadsheet of the complete proteomics results as a supplemental item
- 2) Line 46: The images in extended figure 2 show different pattern of FRI in warm than is shown in Fig1. From the methods it seems as though very different objectives and microscopes were used to collect these data. LSM880 with 40x 1.2 objective versus an Elyra with 100x high NA objective seems to present different resolution. The 40x 1.2 NA objectives used for the majority of data

acquisition are not sufficient to detect some smaller more dynamic condensates. It is therefore inappropriate to say no condensates were observed but rather, condensates of this size were not observed when presenting data with the 40x 1.2NA objective.

3) Line 102: "known to be essential" is incorrect. "known to be involved"

4) Line 194: What is the significance of this particular isoform of COOLAIR?

5) Line 240: Why invoke a hub? It seems like complex is more appropriate here.

Referee #2 (Remarks to the Author):

The vernalization response (flowering in response to extended cold) of *Arabidopsis* has been studied in great detail, but a key question is how plants sense extended periods of cold to trigger the repression of the FLC gene, a repressor of flowering that is down regulated in cold and subsequently maintained in a repressed state by the plant's polycomb repression system. This paper presents evidence that FRI, an activator of FLC, is sequestered into nuclear condensates in cold temperatures, and that this sequestration is reversed in warm temperatures. The authors present a comprehensive dataset investigating domains of FRI required for condensate formation, requirement for other FRI complex components and a role for a splice variant of the COOLAIR antisense transcript in condensate formation.

The work presented is novel, the approach taken is valid and the data is of good quality. Where appropriate, statistics have been applied. The references cited credit previous work appropriately. The manuscript is well-written and reasonably accessible to a general reader.

I found this to be a good paper that presents new and exciting insights into the mechanism by which *Arabidopsis* responds to vernalization. I found that the data presented was convincing and the interpretation of the data was reasonable. The three areas where I would have liked to have seen more data were the definition of the regions of FRI required for condensate formation (the manuscript reported on two fairly coarse deletions that prevent condensate formation), a more concrete description of the protein composition of the condensates (for example by comparing the subcellular localisation of some of the proteins identified by affinity purification from Extended Data Table 1 with FRI-GFP) and more clarity on whether the processes of sequestration into condensates and their dissociation is purely temperature-driven or requiring cold specific factors.

Minor points:

Figure 4a. ChIP amplicon positions are indicated here but not referred to in this figure so could be removed.

Extended Figure 8c. This data would be better presented as a bar graph, joining the points with lines implies the the data can be interpolated, the amplicons are also not evenly spaced on the FLC gene as is implied by the spacing in the figure.

Referee #3 (Remarks to the Author):

Zhu et al. report in this manuscript on a cold-dependent mechanism of FRIGIDA sequestration into nuclear condensates which are shown to be unable of FLC promoter occupancy, hence facilitating FLC transcriptional repression. This process is shown to be reversed by warm temperature and is proposed to prevent premature flowering in response to fluctuating temperatures during autumn. Remarkably, FRI sequestration correlates with increased FRI interaction with several co-transcriptional regulators and RNA splicing ribonucleoproteins, and is associated with cold accumulation of a specific isoform of the antisense RNA COOLAIR. Thus, the work unveils a new mechanism for COOLAIR regulation of FLC transcription.

FRI accumulates in the cold and forms nuclear condensates that increase in size and number after cold exposure. Cold increases FRI half-life, with higher protein concentrations being necessary, but

not sufficient for FRI nuclear condensate formation. Cold-induced condensates are impaired in the *frl1-1* background and significantly disrupted in *flx-2* and *suf4*. Co-expression of FRL1 enhances FRI condensate formation, whereas FRL1 does not form condensates in its own. Each the disordered FRI C-terminal domain and coiled-coiled domains are shown to be required for condensate formation. FRI-GFP condensates do not co-localize with nascent FLC transcripts, showing that FRI is unable to activate FLC transcription when sequestered into nuclear condensates. Condensates increase in size and number on cold exposure and this process requires protein synthesis. Furthermore, nuclear condensates disappear within 5 hours of return to warm, are restored by cold after a 3 hour warm treatment, while are absent at warm temperatures independently of FRI protein levels, in support of FRI interaction with cold-specific factors being required for their formation. Authors show that the Class II.ii COOLAIR isoform, including an additional exon, is actually differentially expressed on cold exposure, in support of FRI interaction with RNA splicing factors, as observed by IP-MS, to be involved in accumulation of this isoform. COOLAIR Class II.ii RNA is in fact enriched in FRI-GFP pull-down fractions after 2 weeks of cold treatment, while formation of FRI-GFP condensates is significantly impaired in FLC Terminator Exchange (TEX) lines, devoid of COOLAIR cold induction. These lines are also defective in reduced FRI occupancy of the FLC promoter in the cold, which may account for their inefficient FLC silencing. Overall, these findings strongly support a role of FRI in specifically promoting COOLAIR Class II.ii production in the cold, at the same time that FRI –COOLAIR Class II.ii interaction facilitates FRI condensate assembly.

Taken together, this study reveals a prominent role of COOLAIR Class II.ii in temperature-dependent FRI condensation and in modulating FRI-mediated FLC transcription. Hence, rapid dynamics of FRI condensation are proposed to prevent premature FLC shut-down under fluctuating autumn temperatures, until prolonged cold exposure during winter months.

IP-MS studies identified FRL1 as the FRI partner with a strongest differential interaction between warm and cold conditions. Formation of FRI Nuclear condensates is in fact impaired in *frl1-1* mutant, correlating with a slower FLC transcriptional shutdown. By opposite, lack of Class II.ii isoform in the TEX background reduces size and number of nuclear condensates, but does not impair their formation. This seems to underscore a relevant role of FRL1 in nuclear condensate formation in addition to COOLAIR Class II.ii.

In Extended dataset 6, *frl1-1* effects on COOLAIR Class II.ii splicing were observed to be weaker than those of *flx2* and *suf4*. Actually, COOLAIR Class II.ii is reduced in *suf4* in response to cold treatment, although fold change is not statistically significant. On the other hand, impact of *frl1-1* on FRI condensation is stronger than that of *suf4*, and certainly *flx2*. This suggests that FRI likely associates with different protein partners in warm and cold conditions, with enhanced interaction with FRL1 in the cold playing a central role in triggering FRI condensation. Regretfully, FRI-GFP IP was much less efficient in NV as compared to 2WV conditions, which precludes robust identification of cold-specific partners. Additional protein interaction studies showing enhanced cold FRI interaction with FRL1 or the splicing factors identified by IP-MS, would strongly increase significance of the work. Alternatively, IP-MS studies using FRI-GFP *frl1-1* might be extremely informative with regard to identification of possible cold-responsive regulators mediating cold-induced FRI sequestration.

Minor points:

Extended data Fig.1: Provide the flowering phenotype of FRI-GFP and FRI lines after cold treatment.

Referee #4 (Remarks to the Author):

This is a manuscript on the fascinating topic of season sensing in plants. In this manuscript, the authors found that in cold stress the FLC activator FRIGIDA (FRI) is localized in nuclear condensates which overlap with Cajal bodies. The formation of the condensates is reversible since

they disappeared when the plants were moved back to warm. The dynamic formation of the FRI condensates is accompanied by decreased FRI occupancy at the FLC promoter in cold. The authors conclude that the formation of the condensates is regulated by the interaction of FRI with multiple co-transcriptional regulators and an isoform of FLC antisense RNA COOLAIR. This work is a great exploration for the molecular basis of the season sensing mechanism of higher plants.

Major comments (in the view of this reviewer, these need to be addressed before the manuscript can be published):

1. Line 45: “. . . we found that FRI-GFP forms nuclear condensates . . .” The nuclear-condensate localization of the FRI protein is not a novel discovery, and the dynamic of the localization of FRI in this manuscript is different from a previous report. In 2014, Hu, X. et al. (Plant Cell; reference 29) generated an FRI-GFP fusion protein driven by the native FRI promoter and found that it was localized in nuclear condensates and was colocalized with Cajal bodies by using transgenic Arabidopsis. However, different from the findings of the present manuscript, Hu, X. et al. found that cold treatment induced the degradation of FRI by using both confocal and Western blot. The authors should discuss these previous results and this difference in results.

2. The title and a number of prominent sentences imply that the work demonstrates that the sequestration of FRI is causing the reduction of FRI occupancy at the FLC promoter, which in turn facilitates FLC transcriptional repression. However, the work does not demonstrate such causality, it only demonstrates correlation of the changes. The data do not exclude the possibility that the causality runs completely in the reverse direction of the authors’ conclusions, with FLC transcriptional repression causing the reduction of FRI occupancy at the FLC promoter, which in turn promotes the formation of the FRI condensates. The latter scenario would be a very different story, where the condensates are not regulators of the cold program but rather a by-product of changes at the FLC locus.

The sentences in question are the one beginning Line 14 in the abstract: “Here, we show that cold temperature rapidly promotes localization of FRI into nuclear condensates, reducing FRI occupancy at the FLC promoter and facilitating FLC transcriptional repression.” and sentences line 235 “Taken together, our study has revealed a temperature-dependent biomolecular condensate mechanism that modulates FRI activation of FLC transcription, so facilitating FLC shut-down in natural fluctuating temperatures.” and line 241: “Upon transfer to cold, FRI protein is stabilized and changed protein/COOLAIR interactions result in accumulated nuclear condensates, sequestering FRI away from the FLC promoter and reducing FLC transcription (Fig. 4h).”

To test the causality implied by their model, the authors could exogenously drive formation of FRI condensates by promoting its oligomerization, e.g. by addition of a domain that oligomerizes upon addition to a drug. They could then see if FRI condensate formation (induced by addition of the drug) causes FLC transcriptional repression even in the absence of temperature changes, as would be predicted by their model.

3. The cold and warm conditions used in this study are not only different in temperature but also different in light and dark cycles (16 h light and 8 h dark in warm and 8 h light 16 h dark in cold) (lines 354-359). The sudden change to a different length of the daytime leads to the problem that everything in cold treatment could also be caused by the change of the light and dark cycle. The authors only attribute everything to the cold.

Additionally, plants in cold in this study actually have a longer living period than the control plants in warm. For example: NV = 10 day warm, 1WV = 10 day warm + 1 week cold. Controls with the same length of continuing warm treatment should also be provided, in the case above the additional control should be 10 day warm + 1 week warm. (see Fig. 3 of Hu et al 2014 Plant Cell for an example of this)

4. Line 19 of the abstract: “This sequestration is promoted by multivalent interactions including specific co-transcriptional regulators and a cold induced isoform of the antisense RNA COOLAIR”

and the model in Fig. 4h. If COOLAIR binds to the condensates and promotes their formation, it would be a great finding of a new regulation mechanism of FLC by COOLAIR. However, the authors do not present any direct evidence supporting the conclusion that COOLAIR is physically present in the FRI condensates. To test if COOLAIR (class II.ii) is in the condensates, an experiment such as in vivo co-localization of COOLAIR (class II.ii) and FRI within the FRI condensates, identification of class II.ii isoform in isolated FRI condensates or physical interaction of class II.ii isoform with FRI or any other verified FRI condensate component (such as FRL1) is needed.

5. GFP intensity in the nucleus was used alone for the quantification of protein abundance. This is not sufficient considering that proteins can move in and out of the nucleus. For example, Line 72/Extended Data Fig. 4 - FRI half-life was measured as <24 hours in the warm - it looks like FRI-GFP is getting exported from the nucleus upon CHX addition in warm conditions. To measure protein stability, protein levels in whole tissues should be measured by western blot.

6. Line 82: "Therefore, higher protein concentration is necessary, but not sufficient, for the cold enhancement of FRI nuclear condensate formation." I did not see data demonstrating the necessity of higher protein expression levels for cold enhancement of condensate formation. One way to support this point would be to show quantification of punctate abundance and intensity in NV cold vs NV warm, compared to 2WV in cold vs 2WV in warm.

Minor comments (suggested opportunities for improving the work, not essential for publication in the view of this reviewer):

1. Line 36: the authors said that the FRI-GFP line fully complemented all fri phenotypes, but in the figures only the early flowering phenotype was shown.

2. Line 75-78: I suggest removing "similar to the previous study" because the previous study used a line with "FRI-GFP fusion protein driven by the native FRI promoter" whereas the present work uses a 35S promoter, and from the Extended Data Fig. 5a, it seems that the FRI expression level in the present work's 35S line is much higher than in the FRI-GFP line.

3. Lines 19, 98-100, 191, 225: the term "multivalent interactions" appears to be used to refer to interactions with multiple proteins, which to this reviewer seems unconventional because typically in the context of biomolecular condensates, "multivalent interactions" normally means that at least one component has multiple binding sites for another. To improve clarity, all instances of "multivalent interactions" in this manuscript could be changed to "interactions with multiple proteins".

4. Line 141-145: FLC transcriptional shutdown rates are lower in fix-2 and suf4 mutants than in frl1-1 mutant (Extended Data Fig. 6e), but FRI nuclear condensates are observed in these two mutants (Fig. 1c). It would be helpful if the authors suggested an explanation for this observation.

5. Line 203-209: it would be good to assess FRI expression level in TEX line in the first 12 h cold condition and after 2 weeks of cold (as the cold condition tested in this manuscript). It is possible that the no increase and reduced condensates are caused by the decreased level of FRI, which might be due to a negative feedback of the reduction of COOLAIR.

6. Many different lengths of cold treatment were used in the manuscript (2WV for Fig.1a; 1WV in Fig.1b; 6WV for Fig.2a,b; 2WV for Fig.4b and 2WV and 4WV for Extended Data Fig. 2e). A standardized time point after cold treatment (better an early one like 1WV) would strengthen the manuscript significantly.

7. Fig.2b: why are two different images shown for FRI in panel b? Is this panel not fully labeled?

8. Extended Data Fig. 4a-e: to get the conclusion of "FRI half-life was measured as <24 hours in the warm, but >24 hours in the cold", maybe it's better to compare NV in warm with NV in cold, and 2WV in warm with 2WV in cold. So it would be better to add the statistical analysis for NV in cold and confocal images and statistical analysis for 2WV in warm.
9. Extended Data Fig. 9a-b: it would be helpful if the authors could suggest an explanation for the increased number of the condensates in the TEX line within the first 6 hours of plants experiencing cold; and why the number of condensates is almost the same as in WT.
10. Statistical analyses are needed for Fig.1c, Fig.3c, Extended Data Fig. 2c-d, Extended Data Fig. 3, Extended Data Fig. 4e and Extended Data Fig. 7a.
11. Better or more detailed figure legends are needed for Fig.1f,h,j, Fig.4a, Extended Data Fig. 1a-d, Extended Data Fig. 4a-e and Extended Data Fig. 8a.
12. Supplementary Table 1: in addition to the number of nuclei and nuclear condensates, can the number of plants analyzed for each experiment also be provided? In addition, 24 hours in cold are not actually shown in Extended Data Fig. 9a and b.
13. The organization of the Extended Data Figs with main Figures and the organization of panels within a figure could be improved (eg. Fig1 and Fig3, the panels were not introduced in the text in order).

Author Rebuttals to Initial Comments:

Referee #1 (Remarks to the Author):

Zhu et al report data connecting the cold-induced formation of FRI-containing nuclear condensates to the repression of FLC in Arabidopsis. Using a line with FRI-GFP at endogenous expression levels they perform IP-MS +/- cold and find a number of interesting proteins related to transcription that all go up upon cold-induction. Surprisingly, FRI peptide counts also go up upon cold-induction even though the expression of its target gene (FLC) is reduced. They inspect these lines under a microscope and observe nuclear condensates which become more numerous and prominent upon cold-induction. They note a minute-scale recovery after photo-bleaching, which is significantly slower than a previous condensate they reported in 2019. Knockout of several FRI interaction partners (although some of these were not present in the IP) decreased cold-induced condensate accumulation. They find that FRI condensates colocalize with Cajal body constituents, which were present in the IP-MS. Deletion of a c-terminal domain or a coil-coil domain prevented FRI condensates. Using FLC intron probes, the authors find no colocalization between RNA FISH and GFP-FRI condensates. They then test cold-warm cycles and show that FRI condensates and FLC and COOLAIR expression are dynamic and reversible within 6-12 hours and suggest a correlation between these phenomena. They then focus on COOLAIR, a lncRNA induced by cold. RNA-IP showed FRI-GFP enriched on a specific splice form of COOLAIR. Finally using a line with defective cold-induction of COOLAIR they show that FRI condensates have a small reduction in cold-induced accumulation.

The paper presents interesting data, but is difficult to follow for a number of reasons. On a practical level, the data in the figures, text, and extended data are not presented in the same order making the reader go back and forth between threads of data and narrative and bouncing between multiple figures. This needs to be fixed. On the conceptual level, the authors do not seem to have their minds made up on what the emphasis of this paper should be. There are at least three themes being presented: environment sensing of biomolecular condensates, the requirement for heterotypic interactions leading to greater control of condensate formation, and lncRNA driven condensate formation. The paper bounces between these themes and the resulting paper is difficult to follow and piece together. Overall, the paper reports several interesting findings that do not form a cohesive whole that would be expected for publication in Nature. Major revisions would be required for publication.

We have extensively edited the manuscript to address these concerns. The order of the figures has been reorganized to follow the text. The concept we focus on is the dynamic partitioning of a transcriptional activator conferring plasticity in response to natural temperature fluctuations.

Major comments:

1) The emphasis at the outset of the text is on solving the mystery of “how autumn is distinguished from winter” but the connection to this big question seems lost after the abstract. The title says that this sequestration is for “Autumn sensing” but that does not come through in the text and it is not clear which experiment tests “Autumn sensing”

We apologize for this ambiguity. Now the question has been defined as “how fluctuating cold induces *FLC* transcriptional repression”. The title has been changed to “Cold-induced *Arabidopsis* FRIGIDA nuclear condensates for *FLC* repression”.

a. Fig 3b and g. What are the expression levels after 2 weeks of cold? Is the *FLC* shutdown and *COOLAIR* induction complete after 12 hours in cold or is it just beginning?

The expression level of unspliced *FLC* after 2 weeks of cold is presented in Fig. 2c, Extended data Fig. 1b and Extended data Fig. 7f, l. The *COOLAIR* expression level and fold change after 2 weeks of cold are presented in Extended data Fig. 9c and f. By normalizing to the NV level, neither the *FLC* shutdown nor *COOLAIR* induction is complete after 12 hours in cold (Extended data Fig. 8c and Extended data Fig. 9f). These have also been added to the text line 176-177 and line 230-231.

b. Fig 3d and h. What are the expression levels after 2 weeks in warm? Is *FLC* reactivation and *COOLAIR* shutdown complete after 12 hours in warm or is it just beginning?

We have analyzed both *FLC* reactivation (now in Fig. 3d) and *COOLAIR* shutdown after 24 hours in warm (now in Fig. 4d). Neither is complete after 12 hours in warm. In addition, we found *FLC* expression level is further increased after 10 days in warm (Extended Data Fig. 8k) (has been described in line 200-201).

c. These expression timescales and the timescale of condensate accumulation and dissolution compared to the temperature fluctuations in field studies during autumn are relevant to the argument that these condensates are regulatory or act as buffers against transient spikes in temperature. If 6-12 hours of cold or warm are sufficient to fully reactivate or silence *FLC/COOLAIR* it is difficult to imagine how this buffers from natural temperature fluctuations. Is the idea that you need long-term sustained suppression or activation of *FLC* to get vernalization? What is that time-window of sustained cold or warmth required?

As mentioned above, 6-12 hours of cold or warm are not sufficient to fully silence or reactivate *FLC/COOLAIR*. The 6-12 hour time period was inspired by the temperature fluctuations in autumn (e. g. a cool night or a warm afternoon). In this study, we focused on the *FLC* transcriptional shutdown at the early vernalization stage, then followed by PRC2-mediated epigenetic silencing (Buzas et al., 2011). The transcriptional shut-down takes ~2-3 weeks in lab continuous cold, as we describe at the beginning of the paper (line 29-30). However, in the field where warm temperature spikes occur frequently, it takes months (Hepworth et al., 2018). After this initial transcriptional shutdown, it takes another 2-3 weeks of continuous cold exposure in the lab for the locus to become fully epigenetically silenced by the PRC2-mediated epigenetic mechanism (Questa et al., 2016 and Yang et al., 2017).

2) There is a disconnect on what exactly is “temperature-dependent.” The title and several sections in the text suggest that FRI condensates are temperature-dependent, but really it is *COOLAIR* expression and FRI-stability that are temperature-dependent. The authors demonstrate several times that FRI can form condensates at warm temperatures. This is more of a disconnect because there are many examples of protein condensation directly sensitive to environmental states (temperature, pH, redox state, etc.) without a secondary intermediate.

We have used “cold-induced” or “temperature controlled” to replace “temperature-dependent” to be more accurate. We indeed mean the “FRI sequestration” rather than condensation is temperature-dependent. FRI can form condensates in warm temperatures but these turn over faster and thus sequester less FRI.

3) How does cold induce *COOLAIR* expression?

We show that *COOLAIR* induction requires the integrity of the FRI complex (Extended data Fig. 9a-f) and the association between FRI and U2B” is enhanced in cold (Extended data Fig. 3e-g). Therefore, we envisage it is the changed FRI interactions in cold, including the cold specific association between FRI and *COOLAIR*, which switches FRI association to the *COOLAIR* promoter to enhance *COOLAIR* transcription.

C-Repeat Binding Factors (CBFs) are transcription factors in plants involved in response to low temperature. We find two CRT/DRE elements, with a CCGAC/GTCGG core sequence known to bind all three CBFs, downstream of the *FLC* translation stop codon (in the *COOLAIR* promoter region). In addition, the cold induction of *COOLAIR* is lost in *cbf1,2,3* (*cbfs*) triple mutant (Jade Doughty and Caroline Dean, unpublished data). The appearance of MED14, MED16, RCF1 and PRP31 in the FRI IP-MS (Extended Data Table 1), which are required for either transcription or pre-mRNA splicing of CBF-responsive cold-regulated genes (Hemsley et al., Guan et al. and Du et al.), suggests a possible role for these cold

responsive factors in recruiting FRIc to *COOLAIR* promoter. This will be part of the next steps in our investigation.

4) How does cold stabilize the FRI protein?

FRI protein is not stabilized in tobacco leaves (Extended Data Fig. 8n-p), suggesting this requires specific factors in *Arabidopsis*. We show FRL1 and cold induction of *COOLAIR* are both required (Extended data Fig. 6b, c and Extended data Fig. 9j, k). In addition, we have found both the coiled-coil domains and C-terminal disordered domain are required for the condensation, probably because they mediate the interactions with multiple factors. There are also studies demonstrating post translational modification influences protein stability. Our next step therefore is to investigate whether cold changes FRI protein post-translational modification.

5) If FRI is sequestered into gel-like condensates sufficient to prevent activation of FLC, how does FRI activate *COOLAIR* in 2WV? What privileges *COOLAIR* isoform I.ii during 2WV? What prevents FRI from activation of *COOLAIR* in NV?

This would link to comment 3). There is still non-sequestered FRI in the cold that could be recruited to *COOLAIR* promoter. There could also be a non-equilibrium feedback mechanism during transcription that high levels of RNA will dissolve the possibly formed condensates at the *FLC* locus (line 142-145).

As mentioned in line 222-224 we reason the cold induction of Class II.ii may involve FRI interaction with splicing factors, which have previously been shown to play a role in cold responsive gene regulation. We have verified a splicing factor interacting with FRI is required for the splicing of the additional exon of *COOLAIR* isoform II.ii in cold, but this part of the work is still ongoing and not appropriate to include here. In fact, FRI also interacts with these factors in NV (Extended data Table 1 and Extended data Fig. 3e-g) and the relative expression level of Class II.ii in FRI is higher than in *fri* even in NV (Extended data Fig.9 a-c). Therefore, the specific induction of *COOLAIR* isoform II.ii in cold is due to the increased interaction of FRI with cold specific transcriptional regulators and splicing factors.

6) Is FRI forming its own condensate or is it becoming a constituent of cajal bodies or some other nuclear body?

The condensates formed by U2B", a Cajal body marker, are not changed by cold and the FRI and U2B" condensates do not totally overlap (Extended data Fig.3 e-g). FRI condensates are therefore different from Cajal bodies but share components. We also confirmed the colocalization of FRI with GFP-ELF7 and overexpressed TAF15b-GFP. The FRI condensates have the interesting property where their association becomes stronger in the cold, due to increased FRI stability.

7) The observation that *fri1*, *flx-2*, and *suf4* are required for cold enhancement is interesting, but it is unclear whether this is a direct effect of heterotypic interactions or an indirect effect. In particular, if these are so essential for heterotypic interactions forming a condensate why were these not picked up in the IP experiments? The explanation in the text of highly dynamic interactors does not make sense given that these IPs were performed in lysates of crosslinked cells, which should stabilize dynamic interactions.

We co-expressed GFP-SUF4 and GFP-FLX with FRI-mScarlet I in tobacco leaves. In contrast to FRL1, which associates with FRI in nuclear condensates, we did not observe any condensate formation by SUF4 or FLX (please see the figure below). Therefore, in this system SUF4 and FLX indeed are less associated with FRI in condensates than FRL1. This is consistent with FRL1, but not FLX or SUF4 being immunoprecipitated in cold. FRI nuclear condensates are also less affected in *flx-2* and *suf4* (line 79-80). The disrupted FRI condensation in *flx-2* and *suf4* is more likely to be an indirect effect of disrupted *COOLAIR* transcription as we discuss in line 240-241.

[redacted]

FLX and SUF4 are genetically required for *FLC* upregulation in NV (line 164-167), however we did not detect them in the IP-MS, probably due to low recovery of proteins as a result of FRI instability in warm conditions (Extended Data Table 1). We have changed our description in line 38-41.

8) Quantification of results in figure 2d is needed.

Now the quantification data for figure 2d, e are shown in Fig. 2f, g.

9) Quantification as reported in 1a-b for the mutants in 1c. The effect size of TEX on condensate accumulation is modest (fig 4e-f). Will be important to see how these compare to the mutants in 1c.

The quantification data of FRI condensates in *frl1-1*, *flx-2* and *suf4* are displayed in Fig. 1h, i. The effect in *frl1-1* is the most severe while *flx-2* and *suf4* is less so. *flx-2* has the least effect on size, similar to the *TEX* line. One reason why the effect on condensate accumulation in *TEX* is modest is that *COOLAIR* is not expressed in all the cells in plants.

10) How does *COOLAIR* induce condensate accumulation? Is it a constituent of the FRI condensates? Do the authors expect that FISH probes against spliced *COOLAIR* would localize to the FRI-GFP condensates in 2WV? What about in the few FRI condensates observed in NV?

Unfortunately, we have not been able to show colocalization of the spliced isoform of *COOLAIR* due to technical limitations of current single molecule RNA FISH protocols. The additional exon of Class II.ii is too short to give a specific signal. However, we performed RIP in *frl1-1* where FRI condensation is severely affected (Fig. 1g-i), but *COOLAIR* expression is still relatively high (Extended Data Fig. 9a-c). We found enrichment of Class II.ii by FRI-GFP in cold is reduced (Extended Data Fig. 9i). This supports our view that *COOLAIR* is one of

the cold specific factors required for FRI condensate accumulation in cold. As the splicing of *COOLAIR* is co-transcriptional we think at least some proportion of *COOLAIR* will associate with the free FRI around the locus and this could be the “seed” for the condensate assembly.

The enrichment of Class II.ii by FRI-GFP in NV is at background level, while the enrichment of Class I.i and II.i is relatively high (Fig. 4b). So, either the few FRI condensates in NV do not associate with *COOLAIR* or there is only rare association with the other *COOLAIR* isoforms (I.i and II.i).

11) Is *COOLAIR* isoform expression sufficient to cause accumulated FRI condensates and does that lead to a decrease in *FLC* expression? Can you then show that either the FRI with CC or c-term deletion does not form condensates upon *COOLAIR* and *FLC* expression is not suppressed? Presumably *FLC* expression will be impacted by FRI mutations, but do you not get further repression upon 2 weeks of cold?

COOLAIR isoform expression is not sufficient to cause accumulated FRI condensation. In *fri1-1*, *COOLAIR* expression is higher than in *flx-2* and *suf4* (Extended Data Fig. 9 a-c), but the effect on FRI condensation accumulation is more severe (Fig. 1g, i). In addition, *COOLAIR* is not expressed in every cell. The physical association between FRI and FRL1 seems more important as we draw in the model (Fig. 4i).

The rest of this comment is linked to the next comment, please see the answer below.

12) Line 144-145: The slower *FLC* shutdown due to *fri1-1* mutant is interesting, but it also suggests that you can get *FLC* shutdown without FRI sequestration into condensates. What would be the mechanism of *FLC* shutdown in the absence of FRI condensates? Is the accumulation of FRI condensates required for *FLC* shutdown or does it make it happen faster?

The accumulation of FRI condensates indeed makes *FLC* shutdown happen faster. We see that sequestration of FRI from *FLC* promoter leads to reduction in the active histone marks H3K36me3 and H3K4me3 and this is one of several steps involved in *FLC* shut-down, a prerequisite for the epigenetic silencing (line 8-14). The accumulation of FRI condensates is one of the first steps in vernalization. Therefore, we write cold induced FRI condensation “facilitates *FLC* shutdown” (line 245 and line 266).

Linking to the last comment, in a *fri* mutant (FRI without the C terminal coiled-coil and disordered domain- FRI-Col-0 in Fig.1j) *FLC* expression is reduced in NV (Fig. 2a-c) but is further repressed after 2 weeks cold exposure (Fig. 2c). FRI-independent *FLC* repression involves induction of *COOLAIR* (Extended Data Fig. 9a-c), which is mutually transcribed with *FLC* (Rosa et al., 2016) and potentially other as yet unidentified processes. In *fri*, *FLC* is in a pre-silenced state with the whole locus covered by H3K27me3 and this also involves repression of transcription via an antisense-mediated PRC2 mediated mechanism (Menon et al., 2021).

13) Would be more helpful to show extended figure 8c for the TEX line with the full set of primers and data plotted as %input.

The data of FRI-GFP *TEX* CHIP is now plotted as input% in Fig. 4h with the WT transgenic line as control. *FLC-TEX* is in the *flc-2* background, a mutant produced by fast neutron that

still contains a residual fragment of the *FLC* 3' region. This prevents the analysis of FRI-GFP occupancy at the *FLC-TEX* 3' end, so we only analyzed the region covering 2000 to 4000 bp from the TSS (Csorba et al., 2014). In addition, we believe it is preferable to use a *FLC*-WT transgene in *flc-2* background as a control. FRI-GFP occupancy on transgenic *FLC*-WT is similar to the endogenous wildtype *FLC* locus (0-2000 bp from TSS) (now Extended Data Fig. 7c).

14) How does this all interplay with Polycomb which the authors have previously proposed as the mechanism for FLC silencing and of COOLAIR function? What are the relative timescales of FRI-condensate accumulation and polycomb deposition?

The cold induced FRI sequestration and thus *FLC* transcriptional shutdown is an early and more rapid vernalization stage than the Polycomb mediated *FLC* epigenetic silencing late vernalization stage. We now describe this at the beginning of the Abstract (line 8-12). In constant lab conditions, the first stage takes 2-3 weeks (line 29-30), while the later stage starts after ~ 3 weeks in cold with the induction of the PRC2 accessory protein VIN3. Full silencing (where the locus is covered by H3K27me3) takes ~ 6 weeks of cold exposure. In contrast, in the field conditions, *VIN3* level only starts to accumulate ~ 60 days after sowing due to the warm spikes in autumn (Hepworth et al., 2018).

Minor comments:

1) Line 43: The authors should provide a spreadsheet of the complete proteomics results as a supplemental item

Now the complete proteomics results from 3 independent experiments have been provided as Supplementary Table 2.

2) Line 46: The images in extended figure 2 show different pattern of FRI in warm than is shown in Fig1. From the methods it seems as though very different objectives and microscopes were used to collect these data. LSM880 with 40x 1.2 objective versus an Elyra with 100x high NA objective seems to present different resolution. The 40x 1.2 NA objectives used for the majority of data acquisition are not sufficient to detect some smaller more dynamic condensates. It is therefore inappropriate to say no condensates were observed but rather, condensates of this size were not observed when presenting data with the 40x 1.2NA objective.

We totally agree that different objectives give different resolution. That's why we have carefully described the details of objectives used in each experiment and indicated in the Methods the resolution of the LSM880 40x 1.1 NA objectives used for the majority quantitative data acquisition as well as the smallest area we can measure (line 491-493). We used the words "increased/promoted/enhanced in the cold" or "reduced/attenuated in the mutant" to describe the overall change in condensation (both size and number) and "less/more condensates formed" rather than "no condensates were observed". Extended figure 2 also looks different as they are single images (as indicated in the legend) while the

images in Fig.1 are Z-stack images with maximum intensity projection applied for purpose of the quantification and comparison.

3) Line 102: "known to be essential" is incorrect. "known to be involved"

Thanks for the correction. This has been corrected.

4) Line 194: What is the significance of this particular isoform of COOLAIR?

This is the first time that one specific isoform of *COOLAIR* has been found to be specifically regulated and involved in the cold induced sequestration of a major transcriptional activator. This represents a new mechanism of *COOLAIR* regulation of *FLC* transcription.

5) Line 240: Why invoke a hub? It seems like complex is more appropriate here.

Now "hub" has been replaced by "complex".

Referee #2 (Remarks to the Author):

The vernalization response (flowering in response to extended cold) of *Arabidopsis* has been studied in great detail, but a key question is how plants sense extended periods of cold to trigger the repression of the *FLC* gene, a repressor of flowering that is down regulated in cold and subsequently maintained in a repressed state by the plant's polycomb repression system.

This paper presents evidence that FRI, an activator of *FLC*, is sequestered into nuclear condensates in cold temperatures, and that this sequestration is reversed in warm temperatures. The authors present a comprehensive dataset investigating domains of FRI required for condensate formation, requirement for other FRI complex components and a role for a splice variant of the *COOLAIR* antisense transcript in condensate formation.

The work presented is novel, the approach taken is valid and the data is of good quality. Where appropriate, statistics have been applied. The references cited credit previous work appropriately. The manuscript is well-written and reasonably accessible to a general reader.

I found this to be a good paper that presents new and exciting insights into the mechanism by which *Arabidopsis* responds to vernalization. I found that the data presented was convincing and the interpretation of the data was reasonable. The three areas where I would have liked to have seen more data were the definition of the regions of FRI required for condensate formation (the manuscript reported on two fairly coarse deletions that prevent condensate formation), a more concrete description of the protein composition of the condensates (for example by comparing the subcellular localisation of some of the proteins identified by affinity purification from Extended Data Table 1 with FRI-GFP) and more clarity

on whether the processes of sequestration into condensates and their dissociation is purely temperature-driven or requiring cold specific factors.

We have hopefully improved the manuscript in the three specified areas:

1) We generated and analysed another FRI truncation with the C-terminal disordered domain (DD) deleted but without disrupting the coiled-coil domain (FRI-DD-GFP). This did not form nuclear condensates (Fig. 1j, k), suggesting the C-terminal disordered domain is required for FRI condensation.

2) We tested the colocalization of FRI with another two proteins identified by affinity purification from Extended Data Table 1. GFP-ELF7 and TAF15b-GFP associate with FRI-mScarlet I in nuclear condensates (Fig. 1f and Extended data Fig. 3d).

3) The processes of sequestration into condensates and their dissociation are not purely temperature driven as FRI condensation in tobacco leaves are not enhanced in cold (Extended Data Fig. 8n-p), indicating additional cold specific factors are required.

Minor points:

Figure 4a. ChIP amplicon positions are indicated here but not referred to in this figure so could be removed.

Now they have been removed.

Extended Figure 8c. This data would be better presented as a bar graph, joining the points with lines implies the the data can be interpolated, the amplicons are also not evenly spaced on the FLC gene as is implied by the spacing in the figure.

The X axis has now been changed to show the distance (bp) to *FLC* TSS (Extended data Fig. 7c). However, by joining the points with lines, we believe these data are more easily compared to the CHIP analysis of H3K4me3, H3K36me3 and H3K27me3 dynamics at *FLC* locus before and during vernalization (Yang et al., 2014).

Referee #3 (Remarks to the Author):

Zhu et al. report in this manuscript on a cold-dependent mechanism of FRIGIDA sequestration into nuclear condensates which are shown to be unable of FLC promoter occupancy, hence facilitating FLC transcriptional repression. This process is shown to be reversed by warm temperature and is proposed to prevent premature flowering in response to fluctuating temperatures during autumn. Remarkably, FRI sequestration correlates with increased FRI interaction with several co-transcriptional regulators and RNA splicing ribonucleoproteins, and is associated with cold accumulation of a specific isoform of the antisense RNA COOLAIR. Thus, the work unveils a new mechanism for COOLAIR regulation of FLC transcription.

FRI accumulates in the cold and forms nuclear condensates that increase in size and number after cold exposure. Cold increases FRI half-life, with higher protein concentrations being necessary, but not sufficient for FRI nuclear condensate formation. Cold-induced condensates are impaired in the *frl1-1* background and significantly disrupted in *flx-2* and *suf4*. Co-expression of FRL1 enhances FRI condensate formation, whereas FRL1 does not form condensates in its own. Each the disordered FRI C-terminal domain and coiled-coiled domains are shown to be required for condensate formation. FRI-GFP condensates do not co-localize with nascent FLC transcripts, showing that FRI is unable to activate FLC transcription when sequestered into nuclear condensates. Condensates increase in size and number on cold exposure and this process requires protein synthesis. Furthermore, nuclear condensates disappear within 5 hours of return to warm, are restored by cold after a 3 hour warm treatment, while are absent at warm temperatures independently of FRI protein levels, in support of FRI interaction with cold-specific factors being required for their formation. Authors show that the Class II.ii COOLAIR isoform, including an additional exon, is actually differentially expressed on cold exposure, in support of FRI interaction with RNA splicing factors, as observed by IP-MS, to be involved in accumulation of this isoform. COOLAIR Class II.ii RNA is in fact enriched in FRI-GFP pull-down fractions after 2 weeks of cold treatment, while formation of FRI-GFP condensates is significantly impaired in FLC Terminator Exchange (TEX) lines, devoid of COOLAIR cold induction. These lines are also defective in reduced FRI occupancy of the FLC promoter in the cold, which may account for their inefficient FLC silencing. Overall, these findings strongly support a role of FRI in specifically promoting COOLAIR Class II.ii production in the cold, at the same time that FRI –COOLAIR Class II.ii interaction facilitates FRI condensate assembly.

Taken together, this study reveals a prominent role of COOLAIR Class II.ii in temperature-dependent FRI condensation and in modulating FRI-mediated FLC transcription. Hence, rapid dynamics of FRI condensation are proposed to prevent premature FLC shut-down under fluctuating autumn temperatures, until prolonged cold exposure during winter months.

IP-MS studies identified FRL1 as the FRI partner with a strongest differential interaction between warm and cold conditions. Formation of FRI Nuclear condensates is in fact impaired in *frl1-1* mutant, correlating with a slower FLC transcriptional shutdown. By opposite, lack of Class II.ii isoform in the TEX background reduces size and number of nuclear condensates, but does not impair their formation. This seems to underscore a relevant role of FRL1 in nuclear condensate formation in addition to COOLAIR Class II.ii.

We agree, but COOLAIR is not expressed in every cell so the effect on FRI condensation in TEX is modest.

In Extended dataset 6, *frl1-1* effects on COOLAIR Class II.ii splicing were observed to be weaker than those of *flx2* and *suf4*. Actually, COOLAIR Class II.ii is reduced in *suf4* in response to cold treatment, although fold change is not statistically significant. On the other hand, impact of *frl1-1* on FRI condensation is stronger than that of *suf4*, and certainly *flx2*. This suggests that FRI likely associates with different protein partners in warm and cold conditions, with enhanced interaction with FRL1 in the cold playing a central role in triggering FRI condensation. Regrettably, FRI-GFP IP was much less efficient in NV as compared to 2WV conditions, which precludes robust identification of cold-specific partners. Additional protein interaction studies showing enhanced cold FRI interaction with FRL1 or the splicing factors identified by IP-MS, would strongly increase significance of the work. Alternatively,

IP-MS studies using FRI-GFP *frl1-1* might be extremely informative with regard to identification of possible cold-responsive regulators mediating cold-induced FRI sequestration.

Thanks for this suggestion. We agree that “FRI likely associates with different protein partners in warm and cold conditions, with enhanced interaction with FRL1 in the cold playing a central role in triggering FRI condensation”. FRL1 but not FLX2 or SUF4 is identified by FRI-GFP IP-MS, suggesting FLX2 and SUF4 are less physically associated with FRI in the cold. But they are required for the transcriptional activity of the non-sequestered FRI, thus *COOLAIR* Class II.ii induction is lost in the mutant, which may then negatively feedback to the cold promotion of FRI condensation. We have edited the text where relevant (line 38-41, line 79-80, line 165-167 and line 240-241).

We compared the colocalization of FRI and U2B” in NV and 2WV plants and found more colocalization between FRI-GFP condensates with U2B” speckles after cold treatment (Extended Data Fig. 3e-g) indicating their interactions are enhanced in cold. We have not performed IP-MS with FRI-GFP *frl1-1*; the low levels of FRI-GFP is significantly reduced in *frl1-1* making the IP less efficient (Extended Data Fig. 6b, c). However, *COOLAIR* enrichment by FRI-GFP in cold is indeed reduced in *frl1-1* (Extended Data Fig. 9i), supporting that *COOLAIR* is indeed one of the cold-responsive regulators mediating cold-induced FRI sequestration.

Minor points:

Extended data Fig.1: Provide the flowering phenotype of FRI-GFP and FRI lines after cold treatment.

Now the flowering phenotype of FRI-GFP and FRI lines after 5 weeks of cold treatment has been displayed in Extended data Fig.1f.

Referee #4 (Remarks to the Author):

This is a manuscript on the fascinating topic of season sensing in plants. In this manuscript, the authors found that in cold stress the FLC activator FRIGIDA (FRI) is localized in nuclear condensates which overlap with Cajal bodies. The formation of the condensates is reversible since they disappeared when the plants were moved back to warm. The dynamic formation of the FRI condensates is accompanied by decreased FRI occupancy at the FLC promoter in cold. The authors conclude that the formation of the condensates is regulated by the interaction of FRI with multiple co-transcriptional regulators and an isoform of FLC antisense RNA *COOLAIR*. This work is a great exploration for the molecular basis of the season sensing mechanism of higher plants.

Major comments (in the view of this reviewer, these need to be addressed before the manuscript can be published):

1. Line 45: “.. we found that FRI-GFP forms nuclear condensates ..” The nuclear-condensate localization of the FRI protein is not a novel discovery, and the dynamic of the localization of FRI in this manuscript is different from a previous report. In 2014, Hu, X. et al. (Plant Cell; reference 29) generated an FRI-GFP fusion protein driven by the native FRI promoter and found that it was localized in nuclear condensates and was colocalized with Cajal bodies by using transgenic Arabidopsis. However, different from the findings of the present manuscript, Hu, X. et al. found that cold treatment induced the degradation of FRI by using both confocal and Western blot. The authors should discuss these previous results and this difference in results.

The transgene used in Hu et al. (2014) resulted in very high levels of FRI protein in non-vernalized plants. Given the native FRI promoter was used it is likely the high expression is caused by multiple T-DNA insertions. This level of FRI is not seen in non-vernalized plants carrying an endogenous FRI with or without a GFP or TAP tag (Extended Data Fig.4 a-d). We also find high levels of FRI expression in a line carrying a 35S: *FRI-GFP* transgene (Extended Data Fig.5 a-e). However, this high level of FRI, which produces more FRI nuclear condensates (Extended Data Fig.5 g-i) does not produce more *FLC* compared to the endogenous FRI gene (Extended Data Fig. 7f, g). Endogenous FRI is low in NV but are high enough to fully activate *FLC*.

Consistent with Hu et al. (2014) we find the FRI protein levels are significantly lower after 4 weeks of cold in the 35S: *FRI-GFP* line (Extended Data Fig.5 c-e), however we still see increased stability in cold (Extended Data Fig. 5f). When CHX treatment was done in medium with 1% glucose there is an obvious accumulation of FRI protein in warm after 24 hours but not in cold (Extended Data Fig. 5f) though FRI is still more stable in the cold (Extended Data Fig. 5f). It thus seems that glucose induces the production of FRI in the warm. With increased protein production in the warm, but degradation unaffected, the protein accumulation will accumulate in NV. We have discussed this in line 111-113.

We performed all the other experiments with plants grown in medium without glucose/sucrose as we described in Methods (line 387-388).

2. The title and a number of prominent sentences imply that the work demonstrates that the sequestration of FRI is causing the reduction of FRI occupancy at the FLC promoter, which is in turn facilitating FLC transcriptional repression. However, the work does not demonstrate such causality, it only demonstrates correlation of the changes. The data do not exclude the possibility that the causality runs completely in the reverse direction of the authors' conclusions, with FLC transcriptional repression causing the reduction of FRI occupancy at the FLC promoter, which in turn promotes the formation of the FRI condensates. The latter scenario would be a very different story, where the condensates are not regulators of the cold program but rather a by-product of changes at the FLC locus.

The sentences in question are the one beginning Line 14 in the abstract: “Here, we show that cold temperature rapidly promotes localization of FRI into nuclear condensates, reducing FRI occupancy at the FLC promoter and facilitating FLC transcriptional repression.”

and sentences line 235 “Taken together, our study has revealed a temperature-dependent biomolecular condensate mechanism that modulates FRI activation of FLC transcription, so facilitating FLC shut-down in natural fluctuating temperatures.” and line 241: “Upon transfer to cold, FRI protein is stabilized and changed protein/COOLAIR interactions result in accumulated nuclear condensates, sequestering FRI away from the FLC promoter and reducing FLC transcription (Fig. 4h).”

To test the causality implied by their model, the authors could exogenously drive formation of FRI condensates by promoting its oligomerization, e.g. by addition of a domain that oligomerizes upon addition to a drug. They could then see if FRI condensate formation (induced by addition of the drug) causes FLC transcriptional repression even in the absence of temperature changes, as would be predicted by their model.

We have edited the text to be more cautious about our conclusions (such as line 15-16 and line 178-179). However, analysis of the *35S: FRI-GFP* transgene helps establish causality (Extended Data Fig.5 a-e). More FRI nuclear condensates are formed in *35S: FRI-GFP* in NV (Extended Data Fig.5 g-i) resulting in lower *FLC* transcription level compared to the endogenous FRI gene (Extended Data Fig. 7f, g). Thus increased FRI condensation does seem to cause *FLC* transcriptional repression even in the absence of temperature changes.

In addition, FRI forms condensates in tobacco leaves in warm where there is no *FLC* transcriptional repression (Fig. 1f and Extended Data Fig. 3a-d).

3. The cold and warm conditions used in this study are not only different in temperature but also different in light and dark cycles (16 h light and 8 h dark in warm and 8 h light 16 h dark in cold) (lines 354-359). The sudden change to a different length of the daytime leads to the problem that everything in cold treatment could also be caused by the change of the light and dark cycle. The authors only attribute everything to the cold.

We use the same light and dark cycles for the shorter cold treatments (0-24 hours) and we have now added this detail to the methods (line 393-394). There are no light/dark changes for the 6-hour treatments (Fig. 3), especially for production of the videos (Supplemental Video 1 and 2) taken on the temperature-controlled microscope stage.

We use short photoperiods for long-term vernalization to match natural conditions.

Photoperiod indeed influences flowering time, but mainly through the floral pathway integrators *FT* and *SOC1* (Crevillen and Dean, 2011). As a direct target of *FLC*, *FLC* repression induces *FT* thus enabling flowering (He et al., 2020). *FLC* and photoperiod are parallel pathways integrated by *FT* (Crevillen and Dean, 2011) and in short day photoperiods (8 h light 16 h dark we used in cold) *FT* expression is reduced resulting in later flowering. This is opposite to vernalization, indicating that temperature is the major factor promoting flowering in our standard vernalization conditions. More importantly, there is no difference between *FRI* and *fri* in response to photoperiod (Kim et al. 2006).

Additionally, plants in cold in this study actually have a longer living period than the control plants in warm. For example: NV = 10 day warm, 1WV = 10 day warm + 1 week cold. Controls with the same length of continuing warm treatment should also be provided, in the

case above the additional control should be 10 day warm + 1 week warm. (see Fig. 3 of Hu et al 2014 Plant Cell for an example of this)

Plants grow faster in warm than in cold (for *Arabidopsis*, growth in warm for 1 day equals growth in cold for 1 week), e. g. the cell cycle duration of root epidermal cells is about 24 hours (Rahni and Birnbaum 2019), but it takes about 1 week in cold. As gene expression is tightly related with plant development, we feel it is better to compare the young seedlings at the same developmental stage. However, *FLC* level is relatively constant between 10-12 days post germination, so we use a 10-day warm-grown sample as the NV control. We have edited the relevant sentences.

In addition, for the imaging we strictly controlled the developmental stage for the samples we analysed as described in the Methods line 470-476: “The seedlings used for imaging were grown in the following conditions: NV, plants were grown in warm condition for 7 days; 1WV, plants were grown in warm condition for 6 days before transferred to cold condition for 1 week; 2WV/ 4WV, plants were grown in cold condition for 2/4 weeks with a 5-day pre-growth in warm condition; before short cold treatment (12 hours or shorter), plants were grown in warm condition for 7 days.” so that all the seedlings imaged are developmentally equivalent to a 7-day old warm-grown seedling.

4. Line 19 of the abstract: “This sequestration is promoted by multivalent interactions including specific co-transcriptional regulators and a cold induced isoform of the antisense RNA COOLAIR” and the model in Fig. 4h. If COOLAIR binds to the condensates and promotes their formation, it would be a great finding of a new regulation mechanism of FLC by COOLAIR. However, the authors do not present any direct evidence supporting the conclusion that COOLAIR is physically present in the FRI condensates. To test if COOLAIR (class II.ii) is in the condensates, an experiment such as in vivo co-localization of COOLAIR (class II.ii) and FRI within the FRI condensates, identification of class II.ii isoform in isolated FRI condensates or physical interaction of class II.ii isoform with FRI or any other verified FRI condensate component (such as FRL1) is needed.

This is a similar question as comment 10) from Reviewer #1. Unfortunately, we have not been able to show colocalization of the spliced isoform of *COOLAIR* due to technical limitations of current single molecule RNA FISH protocols (numbers of probes required to generate a signal but the additional exon of Class II.ii is too short to give signal). However, we performed RIP in *frl1-1* where the FRI condensation is severely affected (Fig. 1g-i), but *COOLAIR* expression is still relatively high (Extended Data Fig. 9a-c). We found the enrichment of Class II.ii by FRI-GFP in cold is reduced (Extended Data Fig. 9i). This supports our view that *COOLAIR* is one of the cold specific factors required for cold induced FRI condensate accumulation. As the splicing of *COOLAIR* is co-transcriptional we think at least some proportion of *COOLAIR* will associate with the free FRI around the locus and this could be the “seed” for the condensate assembly.

5. GFP intensity in the nucleus was used alone for the quantification of protein abundance. This is not sufficient considering that proteins can move in and out of the nucleus. For example, Line 72/Extended Data Fig. 4 - FRI half-life was measured as <24 hours in the warm - it looks like FRI-GFP is getting exported from the nucleus upon CHX addition in

warm conditions. To measure protein stability, protein levels in whole tissues should be measured by western blot.

The endogenous FRI protein level from total protein cannot be detected by western blots unless nuclear protein is enriched (Extended Data Fig. 4a, d and Methods line 599-601). However, we performed a western blot analysis after CHX treatment in 35S: *FRI-GFP* using the whole tissues, which indeed shows *FRI-GFP* degrades faster in the warm (Extended Data Fig.5f).

6. Line 82: "Therefore, higher protein concentration is necessary, but not sufficient, for the cold enhancement of FRI nuclear condensate formation." I did not see data demonstrating the necessity of higher protein expression levels for cold enhancement of condensate formation. One way to support this point would be to show quantification of punctate abundance and intensity in NV cold vs NV warm, compared to 2WV in cold vs 2WV in warm.

Thanks for this suggestion. Now these data are presented in Extended Data Fig. 4e-h.

Minor comments (suggested opportunities for improving the work, not essential for publication in the view of this reviewer):

1. Line 36: the authors said that the *FRI-GFP* line fully complemented all *fri* phenotypes, but in the figures only the early flowering phenotype was shown.

Now this has been corrected as "fully complementing *fri* early flowering phenotype" (line 37).

2. Line 75-78: I suggest removing "similar to the previous study" because the previous study used a line with "FRI-GFP fusion protein driven by the native FRI promoter" whereas the present work uses a 35S promoter, and from the Extended Data Fig. 5a, it seems that the FRI expression level in the present work's 35S line is much higher than in the *FRI-GFP* line.

Now this has been removed.

3. Lines 19, 98-100, 191, 225: the term "multivalent interactions" appears to be used to refer to interactions with multiple proteins, which to this reviewer seems unconventional because typically in the context of biomolecular condensates, "multivalent interactions" normally means that at least one component has multiple binding sites for another. To improve clarity, all instances of "multivalent interactions" in this manuscript could be changed to "interactions with multiple proteins".

Thank you – we have edited as suggested.

4. Line 141-145: FLC transcriptional shutdown rates are lower in *fix-2* and *suf4* mutants than in *fri1-1* mutant (Extended Data Fig. 6e), but FRI nuclear condensates are observed in these two mutants (Fig. 1c). It would be helpful if the authors suggested an explanation for this observation.

This relates to the comment 7) from reviewer #1. The *FLC* transcript levels in NV conditions are very low in *fix-2* and *suf4* (Extended Data Fig. 7h-j) suggesting *FLX* and *SUF4* are

required for FRI transcriptional activity. But they are less associated with FRI condensates in the cold (Extended Data Table 1) consistent with FRI nuclear condensates being less influenced in *flx-2* and *suf4* mutants (Fig. 1g-i). However, the reduction of FRI condensation in these mutants may also relate to the reduced *COOLAIR* expression (Extended Data Fig. 9a-f) as in *TEX*. We have discussed these points in line 164-167 and line 240-241.

5. Line 203-209: it would be good to assess FRI expression level in *TEX* line in the first 12 h cold condition and after 2 weeks of cold (as the cold condition tested in this manuscript). It is possible that the no increase and reduced condensates are caused by the decreased level of FRI, which might be due to a negative feedback of the reduction of *COOLAIR*.

FRI expression level in *TEX* line after 2 weeks of cold has been assessed and displayed in Extended Data Fig. 9j, k and the related explanation is added in line 2380-239.

6. Many different lengths of cold treatment were used in the manuscript (2WV for Fig.1a; 1WV in Fig.1b; 6WV for Fig.2a,b; 2WV for Fig.4b and 2WV and 4WV for Extended Data Fig. 2e). A standardized time point after cold treatment (better an early one like 1WV) would strengthen the manuscript significantly.

We have now added 1WV data to Fig.1a-c. FRI condensation is further enhanced after 2 weeks cold, consistent with increased induction of *COOLAIR* expression level after 2 WV rather than 1WV (Csorba et al., 2014), with a peak at 3WV (graph below). We therefore chose 2WV as the cold treatment in most of the experiments, e. g. qPCR (Fig.2c, Extended data Fig.1 a-d and Extended data Fig. 7f-j), immunostaining (Extended Data Fig. 2c, d and Extended Data Fig. 3e-g), FISH (Fig. 2d-g), RIP (Fig. 4b and Extended data Fig. 9g-i), CHIP (Extended data Fig. 7a-d and Fig. 4h), and the comparison between WT and *TEX* (Fig. 4e-g).

However, we needed to add the additional 4WV data point for the FRI protein level analysis (Extended data Fig. 4a-d and Extended Data Fig. 5c, g-i) to make the comparison with Hu et al. (2014) - they showed reduction in FRI protein level at 4WV (Fig.3A, Hu et al. (2014). In addition, we used 6WV for Fig. 2a, b because after 2 WV the repression of *FLC* is not stable when returned to warm (Fig.3 c, d). *FLC* silencing after 6WV is stable so we used these conditions for the GUS staining as this requires incubation at 37 °C for several hours to overnight (see methods).

7. Fig.2b: why are two different images shown for FRI in panel b? Is this panel not fully labeled?

Now they are fully labeled.

8. Extended Data Fig. 4a-e: to get the conclusion of "FRI half-life was measured as <24 hours in the warm, but >24 hours in the cold", maybe it's better to compare NV in warm with

NV in cold, and 2WV in warm with 2WV in cold. So it would be better to add the statistical analysis for NV in cold and confocal images and statistical analysis for 2WV in warm.

The statistical analysis for NV in cold and confocal images and statistical analysis for 2WV in warm have been added and data are shown in Extended Data Fig. 4e-h.

9. Extended Data Fig. 9a-b: it would be helpful if the authors could suggest an explanation for the increased number of the condensates in the TEX line within the first 6 hours of plants experiencing cold; and why the number of condensates is almost the same as in WT.

From Supplementary Video 2 and Extended Data Fig. 8 j, k, it is clear that FRI condensate number fluctuates considerably during the first few hours of cold, but nuclear condensate size continuously increases, and we did detect a significant difference in condensate size between WT and TEX at 12h. Moreover, the violin plot shows a different distribution between WT and TEX. In addition, not every cell expresses *COOLAIR* so the effect is modest.

10. Statistical analyses are needed for Fig.1c, Fig.3c, Extended Data Fig. 2c-d, Extended Data Fig. 3, Extended Data Fig. 4e and Extended Data Fig. 7a.

Statistical analyses have been done for all the data mentioned above except for Extended Data Fig. 2c-d. Due to the amplification effect from the secondary antibody during immunostaining, the quantification of Extended Data Fig. 2c-d is likely less accurate. Instead of quantification, we display more images.

11. Better or more detailed figure legends are needed for Fig.1f,h,j, Fig.4a, Extended Data Fig. 1a-d, Extended Data Fig. 4a-e and Extended Data Fig. 8a.

We apologize for this due to the limited space from the Journal's guideline. But now they have been modified.

12. Supplementary Table 1: in addition to the number of nuclei and nuclear condensates, can the number of plants analyzed for each experiment also be provided? In addition, 24 hours in cold are not actually shown in Extended Data Fig. 9a and b.

Thanks for the suggestion and correction. At least 10 plants were analyzed for each experiment/treatment and this is now indicated in Supplementary Table 1.

13. The organization of the Extended Data Figs with main Figures and the organization of panels within a figure could be improved (eg. Fig1 and Fig3, the panels were not introduced in the text in order).

Now this has been fixed.

References

1. Buzas, D. M., Robertson, M., Finnegan, E. J. & Helliwell, C. A. Transcription-dependence of histone H3 lysine 27 trimethylation at the *Arabidopsis* polycomb target gene *FLC*. *Plant J* **65**, 872-881 (2011).
2. Hepworth, J. et al. Absence of warmth permits epigenetic memory of winter in *Arabidopsis*. *Nat Commun* **9**, 639-639 (2018).

3. Questa, J. I., Song, J., Geraldo, N., An, H. & Dean, C. *Arabidopsis* transcriptional repressor VAL1 triggers Polycomb silencing at *FLC* during vernalization. *Science* **353**, 485-488 (2016).
4. Yang, H. et al. Distinct phases of Polycomb silencing to hold epigenetic memory of cold in *Arabidopsis*. *Science* **357**, 1142-1145 (2017).
5. Hemsley, P. A. et al. The *Arabidopsis* mediator complex subunits MED16, MED14, and MED2 regulate mediator and RNA polymerase II recruitment to CBF-responsive cold-regulated genes. *Plant Cell* **26**, 465-484 (2014).
6. Guan, Q. et al. A DEAD box RNA helicase is critical for pre-mRNA splicing, cold-responsive gene regulation, and cold tolerance in *Arabidopsis*. *Plant Cell* **25**, 342-356 (2013).
7. Du, J. L. et al. The splicing factor PRP31 is involved in transcriptional gene silencing and stress response in *Arabidopsis*. *Mol Plant* **8**, 1053-1068 (2015).
8. Rosa, S., Duncan, S. & Dean, C. Mutually exclusive sense-antisense transcription at *FLC* facilitates environmentally induced gene repression. *Nat Commun* **7**, 13031-13031 (2016).
9. Menon, G. et al. Digital paradigm for Polycomb epigenetic switching and memory. *Curr Opin Plant Biol* **61**, 102012 (2021).
10. Csorba, T., Questa, J. I., Sun, Q. & Dean, C. Antisense *COOLAIR* mediates the coordinated switching of chromatin states at *FLC* during vernalization. *Proc Natl Acad Sci USA* **111**, 16160-16165 (2014).
11. Yang, H., Howard, M. & Dean, C. Antagonistic roles for H3K36me3 and H3K27me3 in the cold-induced epigenetic switch at *Arabidopsis FLC*. *Curr Biol* **24**, 1793-1797 (2014).
12. Crevillén, P. and Dean, C. Regulation of the floral repressor gene *FLC*: the complexity of transcription in a chromatin context. *Curr Opin Plant Biol* **14**, 38-44 (2011).
13. He, Y., Chen, T. and Zeng, X. Genetic and epigenetic understanding of the seasonal timing of flowering. *Plant Commun* **1**, 100008 (2019).
14. Kim, S., Choi, K., Park, C., Hwang, H. J. & Lee, I. *SUPPRESSOR OF FRIGIDA4*, encoding a C2H2-Type zinc finger protein, represses flowering by transcriptional activation of *Arabidopsis FLOWERING LOCUS C*. *Plant Cell* **18**, 2985-2998 (2006).
15. R.Rahni & K.D. Birnbaum, Week-long imaging of cell divisions in the *Arabidopsis* root meristem. *Plant Methods* **15**, 30 (2019)

Reviewer Reports on the First Revision:

Referee #1 (Remarks to the Author):

The revised manuscript is significantly improved. I am satisfied by the authors's response to review comments. It is now suitable for publication.

Referee #2 (Remarks to the Author):

This revised manuscript has addressed the comments raised in my initial review as well as other reviewer comments. I found the manuscript was better focused and had a more logical flow. I have no further comments to raise on the manuscript.

Referee #3 (Remarks to the Author):

Epigenetic silencing of FLC during vernalization occurs after its transcriptional down-regulation in autumn. In this manuscript Zhu et al. show that cold promotes FRI localization into nuclear condensates, whose formation associates with decreased FLC promoter occupancy, and FLC transcriptional repression. Formation of these nuclear condensates is reversed by warm spikes, and is shown to depend on FRI stabilization in the cold, and on interaction with FRL1 and specific co-transcriptional regulators, in addition to the cold-induced COOLAIR II.ii isoform. Dynamic partitioning of the FRI activator in response to temperature fluctuations is thus proposed to allow plants monitor seasonal progression and adjust accordingly flowering transition.

In this revised version of the MS authors provide direct RNA-IP experimental evidence for association of COOLAIR II.ii with the cold-induced FRI condensates, therefore linking alternative COOLAIR splicing with formation of these nuclear condensates. The work shows that FRI condensates have a prevalent role in sequestering this activator for FLC transcriptional shutdown. Warm temperature transients moreover reverse this process, thus revealing a novel mechanism to the dynamic control of FRI activity that presumably prevents premature flowering during autumn months.

Main message of the MS is now easier to follow and studies were carefully performed. There are however a number of remaining observations that merit to be discussed in terms of their possible biological significance:

- 1) FRI is observed to be strongly induced by glucose in warm conditions. Does sucrose has a similar effect? Notably, sucrose accumulates as an osmoprotectant in the cold, and this might have a relevant role in cold-dependent FRI stabilization.
- 2) FRI interaction with the FLC 5' region decreases with cold exposure, similar to FLC transcription. Concomitantly with this, FRI association with the COOLAIR promoter is increased. This suggests that FRI may form alternative transcriptional complexes on cold-exposure, in addition to be sequestered into nuclear condensates. Are the FRI recognition motifs in the FLC and COOLAIR promoters identical? What does mediate this switch? Are FLX and SUF4 involved? (COOLAIR expression is reduced in *flx-2* and *suf4*). This is an interesting observation that deserves to be further discussed.
- 3) CHX application seems to promote in Extended Data Fig 4e the nuclear exclusion of FRI in warm conditions. Overall fluorescence does not seem to be reduced in these images, but depleted from the nucleus. Is this an artefact of the treatment or it indicates that there is a warm/cold-dependent mechanism for FRI nuclear shuttling? This might be relevant as nuclei were obtained for western blot analyses.
- 4) Statistical significance for the area and number of condensates is sometimes confusing. It might be easier for readers if only statistically significant changes normalized to time 0 are included. In extended data Fig.8e, for instance, differences between 12 and 24 h are shown to be significant (**), but not those between 6 and 24 h which looks odd. Does this reflect that condensates recover after several hours of transfer to warm? Same for extended data Fig. 2f.

Referee #4 (Remarks to the Author):

The authors have made some improvements and have addressed my major comments 1, 5, and 6, as well as all of my minor comments. However, major comments 2, 3, and 4 have not been addressed satisfactorily, and in the view of this reviewer, these remaining points should be addressed before the manuscript is suitable for publication. They are pasted below for clarity:

2.

My original comment:

The title and a number of prominent sentences imply that the work demonstrates that the sequestration of FRI is causing the reduction of FRI occupancy at the FLC promoter, which is in turn facilitating FLC transcriptional repression. However, the work does not demonstrate such causality, it only demonstrates correlation of the changes. The data do not exclude the possibility that the causality runs completely in the reverse direction of the authors' conclusions, with FLC transcriptional repression causing the reduction of FRI occupancy at the FLC promoter, which in turn promotes the formation of the FRI condensates. The latter scenario would be a very different story, where the condensates are not regulators of the cold program but rather a by-product of changes at the FLC locus.

The sentences in question are the one beginning Line 14 in the abstract: "Here, we show that cold temperature rapidly promotes localization of FRI into nuclear condensates, reducing FRI occupancy at the FLC promoter and facilitating FLC transcriptional repression." and sentences line 235 "Taken together, our study has revealed a temperature-dependent biomolecular condensate mechanism that modulates FRI activation of FLC transcription, so facilitating FLC shut-down in natural fluctuating temperatures." and line 241: "Upon transfer to cold, FRI protein is stabilized and changed protein/COOLAIR interactions result in accumulated nuclear condensates, sequestering FRI away from the FLC promoter and reducing FLC transcription (Fig. 4h)."

To test the causality implied by their model, the authors could exogenously drive formation of FRI condensates by promoting its oligomerization, e.g. by addition of a domain that oligomerizes upon addition to a drug. They could then see if FRI condensate formation (induced by addition of the drug) causes FLC transcriptional repression even in the absence of temperature changes, as would be predicted by their model.

Authors' response:

We have edited the text to be more cautious about our conclusions (such as line 15-16 and line 178-179). However, analysis of the 35S: FRI-GFP transgene helps establish causality (Extended Data Fig.5 a-e). More FRI nuclear condensates are formed in 35S: FRI-GFP in NV (Extended Data Fig.5 g-i) resulting in lower FLC transcription level compared to the endogenous FRI gene (Extended Data Fig. 7f, g). Thus increased FRI condensation does seem to cause FLC transcriptional repression even in the absence of temperature changes. In addition, FRI forms condensates in tobacco leaves in warm where there is no FLC transcriptional repression (Fig. 1f and Extended Data Fig. 3a-d).

My response to the authors:

The authors have not addressed this concern. The observation that increased levels of FRI-GFP (in the 35S: FRI-GFP line) lead to decreased FLC transcript seems inconsistent with the authors' proposed model where the condensates would titrate FRI away from the FLC promoter, facilitating FLC transcriptional repression. Additionally, the formation of FRI nuclear condensates in 35S: FRI-GFP in NV and the tobacco leaves in warm suggests that the formation of the FRI condensates is independent of cold stress. It seems that to support the authors' model, additional experiments specifically testing the causality relationships are needed, such as the one proposed by this reviewer above (involving drug-induced oligomerization, which would not involve possibly confounding effects of changing protein levels).

3.

My original comment:

The cold and warm conditions used in this study are not only different in temperature but also different in light and dark cycles (16 h light and 8 h dark in warm and 8 h light 16 h dark in cold) (lines 354-359). The sudden change to a different length of the daytime leads to the problem that everything in cold treatment could also be caused by the change of the light and dark cycle. The authors only attribute everything to the cold.

Authors' response:

We use the same light and dark cycles for the shorter cold treatments (0-24 hours) and we have now added this detail to the methods (line 393-394). There are no light/dark changes for the 6-hour treatments (Fig. 3), especially for production of the videos (Supplemental Video 1 and 2) taken on the temperature-controlled microscope stage.

We use short photoperiods for long-term vernalization to match natural conditions.

Photoperiod indeed influences flowering time, but mainly through the floral pathway integrators FT and SOC1 (Crevillen and Dean, 2011). As a direct target of FLC, FLC repression induces FT thus enabling flowering (He et al., 2020). FLC and photoperiod are parallel pathways integrated by FT (Crevillen and Dean, 2011) and in short day photoperiods (8 h light 16 h dark we used in cold) FT expression is reduced resulting in later flowering. This is opposite to vernalization, indicating that temperature is the major factor promoting flowering in our standard vernalization conditions. More importantly, there is no difference between FRI and fri in response to photoperiod (Kim et al. 2006).

My response to the authors:

The authors should include these explanations (about why they don't think the light conditions affect their conclusions) in the main text.

My original comment:

Additionally, plants in cold in this study actually have a longer living period than the control plants in warm. For example: NV = 10 day warm, 1WV = 10 day warm + 1 week cold. Controls with the same length of continuing warm treatment should also be provided, in the case above the additional control should be 10 day warm + 1 week warm. (see Fig. 3 of Hu et al 2014 Plant Cell for an example of this)

Authors' response:

Plants grow faster in warm than in cold (for Arabidopsis, growth in warm for 1 day equals growth in cold for 1 week), e. g. the cell cycle duration of root epidermal cells is about 24 hours (Rahni and Birnbaum 2019), but it takes about 1 week in cold. As gene expression is tightly related with plant development, we feel it is better to compare the young seedlings at the same developmental stage. However, FLC level is relatively constant between 10-12 days post germination, so we use a 10-day warm-grown sample as the NV control. We have edited the relevant sentences.

In addition, for the imaging we strictly controlled the developmental stage for the samples we analysed as described in the Methods line 470-476: "The seedlings used for imaging were grown in the following conditions: NV, plants were grown in warm condition for 7 days; 1WV, plants were grown in warm condition for 6 days before transferred to cold condition for 1 week; 2WV/ 4WV, plants were grown in cold condition for 2/4 weeks with a 5-day pre-growth in warm condition; before short cold treatment (12 hours or shorter), plants were grown in warm condition for 7 days." so that all the seedlings imaged are developmentally equivalent to a 7-day old warm-grown seedling.

My response to the authors:

The authors should include this rationale for matching developmental stage instead of days of growth in the methods.

4.

My original comment:

Line 19 of the abstract: "This sequestration is promoted by multivalent interactions including specific co-transcriptional regulators and a cold induced isoform of the antisense RNA COOLAIR" and the model in Fig. 4h. If COOLAIR binds to the condensates and promotes their formation, it would be a great finding of a new regulation mechanism of FLC by COOLAIR. However, the authors do not present any direct evidence supporting the conclusion that COOLAIR is physically present in the FRI condensates. To test if COOLAIR

(class II.ii) is in the condensates, an experiment such as in vivo co-localization of COOLAIR (class II.ii) and FRI within the FRI condensates, identification of class II.ii isoform in isolated FRI condensates or physical interaction of class II.ii isoform with FRI or any other verified FRI condensate component (such as FRL1) is needed.

Authors' response:

This is a similar question as comment 10) from Reviewer #1. Unfortunately, we have not been able to show colocalization of the spliced isoform of COOLAIR due to technical limitations of current single molecule RNA FISH protocols (numbers of probes required to generate a signal but the additional exon of Class II.ii is too short to give signal). However, we performed RIP in *frl1-1* where the FRI condensation is severely affected (Fig. 1g-i), but COOLAIR expression is still relatively high (Extended Data Fig. 9a-c). We found the enrichment of Class II.ii by FRI-GFP in cold is reduced (Extended Data Fig. 9i). This supports our view that COOLAIR is one of the cold specific factors required for cold induced FRI condensate accumulation. As the splicing of COOLAIR is co-transcriptional we think at least some proportion of COOLAIR will associate with the free FRI around the locus and this could be the "seed" for the condensate assembly.

My response to the authors:

The authors have not addressed this concern. If the authors want to get the conclusion that "antisense RNA COOLAIR physically associates with FRI" (line 21 in abstract), evidence for the physical interaction of COOLAIR with FRI is required. If the authors have problems with colocalization, they should try the other experiments suggested, or other methods. Without solid data, they should not conclude the physical association as is currently written in the abstract, and they should remove the physical association from the model in Fig. 4i.

Author Rebuttals to First Revision:

Referee #1 (Remarks to the Author):

The revised manuscript is significantly improved. I am satisfied by the authors's response to review comments. It is now suitable for publication.

Referee #2 (Remarks to the Author):

This revised manuscript has addressed the comments raised in my initial review as well as other reviewer comments. I found the manuscript was better focused and had a more logical flow. I have no further comments to raise on the manuscript.

We thank the referee #1 and referee #2 for their positive feedback.

Referee #3 (Remarks to the Author):

Epigenetic silencing of FLC during vernalization occurs after its transcriptional down-regulation in autumn. In this manuscript Zhu et al. show that cold promotes FRI localization into nuclear condensates, whose formation associates with decreased FLC promoter occupancy, and FLC transcriptional repression. Formation of these nuclear condensates is reversed by warm spikes, and is shown to depend on FRI stabilization in the cold, and on interaction with FRL1 and specific co-transcriptional regulators, in addition to the cold-

induced COOLAIR II.ii isoform. Dynamic partitioning of the FRI activator in response to temperature fluctuations is thus proposed to allow plants monitor seasonal progression and adjust accordingly flowering transition.

In this revised version of the MS authors provide direct RNA-IP experimental evidence for association of COOLAIR II.ii with the cold-induced FRI condensates, therefore linking alternative COOLAIR splicing with formation of these nuclear condensates. The work shows that FRI condensates have a prevalent role in sequestering this activator for FLC transcriptional shutdown. Warm temperature transients moreover reverse this process, thus revealing a novel mechanism to the dynamic control of FRI activity that presumably prevents premature flowering during autumn months.

Main message of the MS is now easier to follow and studies were carefully performed. There are however a number of remaining observations that merit to be discussed in terms of their possible biological significance:

1) FRI is observed to be strongly induced by glucose in warm conditions. Does sucrose has a similar effect? Notably, sucrose accumulates as an osmoprotectant in the cold, and this might have a relevant role in cold-dependent FRI stabilization.

How sugar signalling and temperature cues are integrated to regulate FRI production is a complex question. Sucrose, glucose, fructose and raffinose were all found to accumulate in plants exposed to cold treatment (Klotke et al., 2004). However, the temperature-dependent stability of FRI did not change when 1% glucose was added to the medium (Extended data Fig. 6f). Clearly, glucose has different effects on FRI protein production in warm and cold conditions (Extended data Fig. 6f). Indeed, the induction of FRI by glucose in warm conditions may be linked to the glucose-TOR pathway. Target of rapamycin (TOR), an evolutionarily conserved protein kinase, is a major glucose signalling mediator (Shi et al., 2018). Sucrose also triggers the TOR signalling pathway so may therefore have a similar effect to glucose (Shi et al., 2018). Thus, we feel the integration of sugar and temperature signalling would need considerable further investigation and is out of the scope of this paper.

2) FRI interaction with the FLC 5' region decreases with cold exposure, similar to FLC transcription. Concomitantly with this, FRI association with the COOLAIR promoter is increased. This suggests that FRI may form alternative transcriptional complexes on cold-exposure, in addition to be sequestered into nuclear condensates. Are the FRI recognition motifs in the FLC and COOLAIR promoters identical? What does mediate this switch? Are FLX and SUF4 involved? (COOLAIR expression is reduced in flx-2 and suf4). This is an interesting observation that deserves to be further discussed.

Thank you for finding this observation interesting. There was a similar discussion in the last revision in response to the comment c.3) from referee #1.

The FRI recognition motifs in the *FLC* and *COOLAIR* promoters are not identical. It has been demonstrated that FRIc binding at *FLC* 5' end in warm depends on the DNA binding ability of SUF4, where a minimal 15-bp sequence is required (Choi et al. 2011). However, the same *cis* element is not found at the 3' end of *FLC* (*COOLAIR* promoter) suggesting a different mechanism at play. This mechanism is likely to involve C-Repeat Binding Factors (CBFs) - transcription factors in plants involved in response to low temperature. We have found two CRT/DRE elements, with a CCGAC/GTCGG core sequence known to bind all three CBFs, downstream of the *FLC* translation stop codon (in the *COOLAIR* promoter region). In addition, the cold induction of *COOLAIR* is lost in *cbf1,2,3* triple mutant (Jade Doughty and Caroline Dean, unpublished data). The appearance of MED14, MED16, RCF1 and PRP31 in the FRI IP-MS (Extended Data Table 1), which are required for either transcription or pre-mRNA splicing of CBF-responsive cold-regulated genes (Hemsley et al., Guan et al. and Du et al.), suggests a possible role for these cold responsive factors in recruiting FRIc to *COOLAIR* promoter. This will be part of the next steps in our investigation.

We show that *COOLAIR* induction requires the integrity of the FRI complex (both FLX and SUF4 are involved) (Extended data Fig. 9a-f) and we have discussed this in the main text (line 169-171 and line 130-132, the formatted manuscript). In addition, the association between FRI and U2B" is enhanced in cold (Extended data Fig. 3h-j). Therefore, we envisage it is the changed FRI interactions in cold, including the cold specific association between FRI and *COOLAIR*, which switches FRI association to the *COOLAIR* promoter to enhance *COOLAIR* transcription.

3) CHX application seems to promote in Extended Data Fig 4e the nuclear exclusion of FRI in warm conditions. Overall fluorescence does not seem to be reduced in these images, but depleted from the nucleus. Is this an artefact of the treatment or it indicates that there is a warm/cold-dependent mechanism for FRI nuclear shuttling? This might be relevant as nuclei were obtained for western blot analyses.

Autofluorescence increases when plant cells are stressed so changes in nuclear/cytoplasmic fluorescence do not necessarily imply regulated nuclear shuttling. However, to try and address this comment we did measure the fluorescence intensity outside the nucleus and only detected an increase in the 2WV sample compared to the warm sample after CHX treatment (see graph below). The reduction in the nuclear FRI-GFP upon transfer from cold to warm temperature is recovered by inhibiting proteasome-mediated protein degradation with MG132 (Extended data Fig. 8g, h). Thus, we feel the data do not support a warm/cold-dependent nuclear shuttling mechanism regulating FRI.

Endogenous FRI cannot be detected in total protein extracts by western blots, necessitating nuclear enrichment. However, we performed a western blot analysis after CHX treatment in 35S: *FRI-GFP* using whole tissue extracts, which indeed shows FRI-GFP degrades faster in the warm (Extended Data Fig. 6f).

4) Statistical significance for the area and number of condensates is sometimes confusing. It might be easier for readers if only statistically significant changes normalized to time 0 are included. In extended data Fig.8e, for instance, differences between 12 and 24 h are shown to be significant (**), but not those between 6 and 24 h which looks odd. Does this reflect that condensates recover after several hours of transfer to warm? Same for extended data Fig. 2f.

Thanks for this suggestion. We have removed the statistical significance between time points after time 0 in Extended data Fig. 8d,e and Extended data Fig. 2f. FRI-GFP nuclear condensates undergo dynamic changes at the first 5 hours transfer to warm (Supplementary Video 1). Both size and number of FRI condensates are tightly related to the protein level (Extended data Fig. 6) which is determined by protein production and degradation. FRI stability changes a lot from cold to warm, thus in a time period before the protein reaching a stable level, FRI condensates may change - the number of condensate may fluctuate (similar phenomena was also observed when plants returned to cold (Extended data Fig. 8m)).

Referee #4 (Remarks to the Author):

The authors have made some improvements and have addressed my major comments 1, 5, and 6, as well as all of my minor comments. However, major comments 2, 3, and 4 have not been addressed satisfactorily, and in the view of this reviewer, these remaining points should be addressed before the manuscript is suitable for publication. They are pasted below for clarity:

2.

My original comment:

The title and a number of prominent sentences imply that the work demonstrates that the sequestration of FRI is causing the reduction of FRI occupancy at the FLC promoter, which is in turn facilitating FLC transcriptional repression. However, the work does not demonstrate such causality, it only demonstrates correlation of the changes. The data do not exclude the possibility that the causality runs completely in the reverse direction of the authors' conclusions, with FLC transcriptional repression causing the reduction of FRI occupancy at the FLC promoter, which in turn promotes the formation of the FRI condensates. The latter scenario would be a very different story, where the condensates are not regulators of the cold program but rather a by-product of changes at the FLC locus.

The sentences in question are the one beginning Line 14 in the abstract: “Here, we show that cold temperature rapidly promotes localization of FRI into nuclear condensates, reducing FRI occupancy at the FLC promoter and facilitating FLC transcriptional repression.” and sentences line 235 “Taken together, our study has revealed a temperature-dependent biomolecular condensate mechanism that modulates FRI activation of FLC transcription, so facilitating FLC shut-down in natural fluctuating temperatures.” and line 241: “Upon transfer to cold, FRI protein is stabilized and changed protein/COOLAIR interactions result in accumulated nuclear condensates, sequestering FRI away from the FLC promoter and reducing FLC transcription (Fig. 4h).”

To test the causality implied by their model, the authors could exogenously drive formation of FRI condensates by promoting its oligomerization, e.g. by addition of a domain that oligomerizes upon addition to a drug. They could then see if FRI condensate formation (induced by addition of the drug) causes FLC transcriptional repression even in the absence of temperature changes, as would be predicted by their model.

Authors’ response:

We have edited the text to be more cautious about our conclusions (such as line 15-16 and line 178-179). However, analysis of the 35S: FRI-GFP transgene helps establish causality (Extended Data Fig.5 a-e). More FRI nuclear condensates are formed in 35S: FRI-GFP in NV (Extended Data Fig.5 g-i) resulting in lower FLC transcription level compared to the endogenous FRI gene (Extended Data Fig. 7f, g). Thus increased FRI condensation does seem to cause FLC transcriptional repression even in the absence of temperature changes. In addition, FRI forms condensates in tobacco leaves in warm where there is no FLC transcriptional repression (Fig. 1f and Extended Data Fig. 3a-d).

My response to the authors:

The authors have not addressed this concern. The observation that increased levels of FRI-GFP (in the 35S: FRI-GFP line) lead to decreased FLC transcript seems inconsistent with the authors’ proposed model where the condensates would titrate FRI away from the FLC promoter, facilitating FLC transcriptional repression. Additionally, the formation of FRI nuclear condensates in 35S: FRI-GFP in NV and the tobacco leaves in warm suggests that the formation of the FRI condensates is independent of cold stress. It seems that to support the authors’ model, additional experiments specifically testing the causality relationships are needed, such as the one proposed by this reviewer above (involving drug-induced oligomerization, which would not involve possibly confounding effects of changing protein levels).

We thank the referee for the proposed experiment, but we feel drug-induced oligomerization would influence many proteins that directly or indirectly regulate *FLC* transcription and so no clear conclusion on the role of FRI condensation would be reached. We reiterate below the evidence we have that supports our model and we have re-edited the text (line15-17, 197-199 and 202-204, the formatted manuscript) to be more cautious in our conclusions.

Evidence we have that supports our model:

- a) In *flx-2* and *suf4*, *FLC* expression is extremely low in NV and relatively unchanged in the cold (Extended data Fig. 7i), but FRI condensates still form in NV and increase in the cold (Extended data Fig. 3d-f) excluding the possibility that *FLC* repression causes and promotes the formation of the FRI condensates.
- b) The FRI condensate behaviour in tobacco leaves where no *FLC* repression occurs (Fig. 1f, Extended data Fig. 3a-c, g) agrees with condensates formation not being dependent on *FLC* repression.
- c) Even though *FLC* expression level is higher in *TEX* but lower in *frl1-1*, both show disrupted FRI condensate formation and slower *FLC* shutdown rate (Fig. 2a, b and Extended data Fig.3d-f, Extended data Fig. 7i, j).
- d) In response to the reviewer comment “increased levels of FRI-GFP (in the 35S: FRI-GFP line) lead to decreased FLC transcript seems inconsistent with the authors’ proposed model where the condensates would titrate FRI away from the FLC promoter, facilitating FLC transcriptional repression” we disagree- increased FRI in the 35S: *FRI-GFP* line gives more FRI nuclear condensates and reduced *FLC* transcription – this is consistent with our model (Extended Data Fig. 6g-i and Extended Data Fig. 7f, g). This is not only in the cold but also in the NV conditions, which supports the idea that increased FRI condensate formation facilitates *FLC* transcriptional repression even in the absence of temperature changes.

We agree that the formation of the FRI condensates is independent of cold stress. We have started with the description of FRI condensates in NV conditions (line 47-49, the formatted manuscript) and use ‘induce’, ‘enhance’, ‘promote’, ‘increase’ and so on to describe the effect of cold on FRI condensate formation in the title, abstract and text. The instability of FRI in the warm results in faster turnover and thus less accumulation of FRI condensates. In 35S: *FRI-GFP* in NV and in tobacco leaves the production of FRI-GFP protein is strongly increased, and increased protein levels result in accumulation of FRI nuclear condensates.

3.

My original comment:

The cold and warm conditions used in this study are not only different in temperature but also different in light and dark cycles (16 h light and 8 h dark in warm and 8 h light 16 h dark in cold) (lines 354-359). The sudden change to a different length of the daytime leads to the problem that everything in cold treatment could also be caused by the change of the light and dark cycle. The authors only attribute everything to the cold.

Authors’ response:

We use the same light and dark cycles for the shorter cold treatments (0-24 hours) and we have now added this detail to the methods (line 393-394). There are no light/dark changes for the 6-hour treatments (Fig. 3), especially for production of the videos (Supplemental Video 1 and 2) taken on the temperature-controlled microscope stage.

We use short photoperiods for long-term vernalization to match natural conditions.

Photoperiod indeed influences flowering time, but mainly through the floral pathway

integrators FT and SOC1 (Crevillen and Dean, 2011). As a direct target of FLC, FLC repression induces FT thus enabling flowering (He et al., 2020). FLC and photoperiod are parallel pathways integrated by FT (Crevillen and Dean, 2011) and in short day photoperiods (8 h light 16 h dark we used in cold) FT expression is reduced resulting in later flowering. This is opposite to vernalization, indicating that temperature is the major factor promoting flowering in our standard vernalization conditions. More importantly, there is no difference between FRI and fri in response to photoperiod (Kim et al. 2006).

My response to the authors:

The authors should include these explanations (about why they don't think the light conditions affect their conclusions) in the main text.

Thanks for this suggestion. Due to the journal's limitation of the length for the main text, we now have added these explanations to the Methods (line 346-348, the formatted manuscript).

My original comment:

Additionally, plants in cold in this study actually have a longer living period than the control plants in warm. For example: NV = 10 day warm, 1WV = 10 day warm + 1 week cold. Controls with the same length of continuing warm treatment should also be provided, in the case above the additional control should be 10 day warm + 1 week warm. (see Fig. 3 of Hu et al 2014 Plant Cell for an example of this)

Authors' response:

Plants grow faster in warm than in cold (for Arabidopsis, growth in warm for 1 day equals growth in cold for 1 week), e. g. the cell cycle duration of root epidermal cells is about 24 hours (Rahni and Birnbaum 2019), but it takes about 1 week in cold. As gene expression is tightly related with plant development, we feel it is better to compare the young seedlings at the same developmental stage. However, FLC level is relatively constant between 10-12 days post germination, so we use a 10-day warm-grown sample as the NV control. We have edited the relevant sentences.

In addition, for the imaging we strictly controlled the developmental stage for the samples we analysed as described in the Methods line 470-476: "The seedlings used for imaging were grown in the following conditions: NV, plants were grown in warm condition for 7 days; 1WV, plants were grown in warm condition for 6 days before transferred to cold condition for 1 week; 2WV/ 4WV, plants were grown in cold condition for 2/4 weeks with a 5-day pre-growth in warm condition; before short cold treatment (12 hours or shorter), plants were grown in warm condition for 7 days." so that all the seedlings imaged are developmentally equivalent to a 7-day old warm-grown seedling.

My response to the authors:

The authors should include this rationale for matching developmental stage instead of days of growth in the methods.

Thanks for this suggestion. We have now added this to the Methods (line 352-354 and line 432, the formatted manuscript).

4.

My original comment:

Line 19 of the abstract: “This sequestration is promoted by multivalent interactions including specific co-transcriptional regulators and a cold induced isoform of the antisense RNA COOLAIR“ and the model in Fig. 4h. If COOLAIR binds to the condensates and promotes their formation, it would be a great finding of a new regulation mechanism of FLC by COOLAIR. However, the authors do not present any direct evidence supporting the conclusion that COOLAIR is physically present in the FRI condensates. To test if COOLAIR (class II.ii) is in the condensates, an experiment such as in vivo co-localization of COOLAIR (class II.ii) and FRI within the FRI condensates, identification of class II.ii isoform in isolated FRI condensates or physical interaction of class II.ii isoform with FRI or any other verified FRI condensate component (such as FRL1) is needed.

Authors’ response:

This is a similar question as comment 10) from Reviewer #1. Unfortunately, we have not been able to show colocalization of the spliced isoform of COOLAIR due to technical limitations of current single molecule RNA FISH protocols (numbers of probes required to generate a signal but the additional exon of Class II.ii is too short to give signal). However, we performed RIP in *frl1-1* where the FRI condensation is severely affected (Fig. 1g-i), but COOLAIR expression is still relatively high (Extended Data Fig. 9a-c). We found the enrichment of Class II.ii by FRI-GFP in cold is reduced (Extended Data Fig. 9i). This supports our view that COOLAIR is one of the cold specific factors required for cold induced FRI condensate accumulation. As the splicing of COOLAIR is co-transcriptional we think at least some proportion of COOLAIR will associate with the free FRI around the locus and this could be the “seed” for the condensate assembly.

My response to the authors:

The authors have not addressed this concern. If the authors want to get the conclusion that “antisense RNA COOLAIR physically associates with FRI” (line 21 in abstract), evidence for the physical interaction of COOLAIR with FRI is required. If the authors have problems with colocalization, they should try the other experiments suggested, or other methods. Without solid data, they should not conclude the physical association as is currently written in the abstract, and they should remove the physical association from the model in Fig. 4i.

Thanks for pointing this out. We now have deleted the sentence “physically associates with FRI” from the abstract (line 20, the formatted manuscript) and changed the model in Fig. 4i (now is Fig. 3i).

References

Klotke, J. K., J., Gatzke, N., Heyer, A.G. Impact of soluble sugar concentrations on the acquisition of freezing tolerance in accessions of *Arabidopsis thaliana* with contrasting cold

adaptation - evidence for a role of raffinose in cold acclimation. *Plant Cell Environ* 27, 1395-1404 (2004).

Shi, L., Wu, Y. & Sheen, J. TOR signalling in plants: conservation and innovation. *Development* 145, 160887 (2018).

K. Choi et al., The FRIGIDA complex activates transcription of *FLC*, a strong flowering repressor in *Arabidopsis*, by recruiting chromatin modification factors. *The Plant cell* 23, 289-303 (2011).

Hemsley, P. A. et al. The *Arabidopsis* mediator complex subunits MED16, MED14, and MED2 regulate mediator and RNA polymerase II recruitment to CBF-responsive cold-regulated genes. *The Plant Cell* 26, 465-484 (2014).

Guan, Q. et al. A DEAD box RNA helicase is critical for pre-mRNA splicing, cold-responsive gene regulation, and cold tolerance in *Arabidopsis*. *The Plant Cell* 25, 342-356 (2013).

Du, J. L. et al. The splicing factor PRP31 is involved in transcriptional gene silencing and stress response in *Arabidopsis*. *Mol Plant* 8, 1053-1068 (2015).

Reviewer Reports on the Second Revision:

Referee #3 (Remarks to the Author):

Authors have addressed in this revised form all issued concerns. I had suggested to expand the model to provide some more details on the switch of FRI occupancy from the FLC into the COOLAIR promoters. I can see that authors are still working on this mechanism and space constrains did not enable them further discussing on this process. This is however not essential to the current MS. Therefore, I believe it is now acceptable for publication.

Referee #4 (Remarks to the Author):

The authors have addressed all of my comments.